# Local Entropy Search over Descent Sequences for Bayesian Optimization

**David Stenger**[1], **Armin Lindicke**[1] , **Alexander von Rohr**[2] , **Sebastian Trimpe**[1]
[1]Institute for Data Science in Mechanical Engineering, RWTH Aachen University, Germany
[2]Learning Systems and Robotics Lab, Technical University of Munich, Germany
[1]`{david.stenger,trimpe}@dsme.rwth-aachen.de`, [2]`{alex.von.rohr}@tum.de`

## Abstract

Searching large and complex design spaces for a global optimum can be infeasible and unnecessary. A practical alternative is to iteratively refine the neighborhood of an initial design using local optimization methods such as gradient descent. We propose local entropy search (LES), a Bayesian optimization paradigm that explicitly targets the solutions reachable by the descent sequences of iterative optimizers. The algorithm propagates the posterior belief over the objective through the optimizer, resulting in a probability distribution over descent sequences. It then selects the next evaluation by maximizing mutual information with that distribution, using a combination of analytic entropy calculations and Monte-Carlo sampling of descent sequences. Empirical results on high-complexity synthetic objectives and benchmark problems show that LES achieves strong sample efficiency compared to existing local and global Bayesian optimization methods.

## 1 Introduction

Many practical optimization problems can be solved to the desired accuracy by relying solely on iterative search strategies such as gradient descent, quasi-Newton methods, or evolutionary algorithms. These methods do not necessarily discover a global minimizer, but refine the current solution. Local optimization has repeatedly demonstrated its effectiveness in finding good solutions particularly in high-dimensional and complex search spaces. Indeed, gradient-based methods remain state-of-the-art for solving extreme-scale problems such as training deep neural networks with billions of parameters (Chowdhery et al., 2023). However, those local optimization methods cannot directly be applied to expensive-to-evaluate black-box functions due to their poor sample-efficiency.

Bayesian optimization (BO) (Garnett, 2023) methods are popular for expensive-to-evaluate black-box functions, yet they typically aim to minimize regret relative to the global optimum. Global search requires reducing uncertainty across the entire domain, which can be intractable in large and high-dimensional spaces (Hvarfner et al., 2024; Xu et al., 2025). Thus, the emphasis on global search in BO stands in contrast to the demonstrated effectiveness of local optimization for complex problems.

We introduce local entropy search (LES)[1], an information-theoretic framework for local BO that transfers the idea of iterative local optimizers to the expensive-to-evaluate black-box setting. LES explicitly targets the solution obtainable by an iterative optimizer starting from an initial design. LES propagates the uncertainty of a Gaussian process (GP) surrogate through the local optimizer, yielding a distribution over the descent sequence and the local optimum (see Fig. 1). At each iteration, LES chooses the next evaluation to maximize the mutual information with the distribution over the descent sequence, thereby reducing its uncertainty.

While several recent efforts (see Sec. 2.1) have brought local strategies to BO framework, acquisition functions have so far provided only partial use of the iterative descent structure or have focused on special cases. Our approach builds directly on entropy search principles (Hennig & Schuler, 2012) but shifts the focus from the global optimum to the reachable local optimum as defined by the optimizer's descent trajectory.

---

[1]Code available at https://github.com/Data-Science-in-Mechanical-Engineering/local-entropy-search.

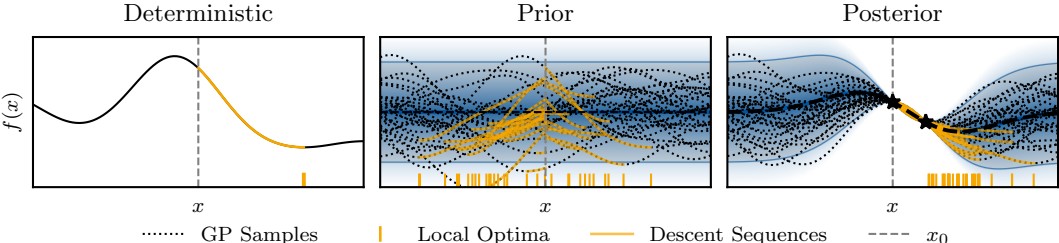

Figure 1: **Distribution over Descent Sequences:** *Left:* Local optimization on the unknown objective function. *Middle:* The prior over the objective function induces a belief over descent sequences. *Right:* After sampling data points the distributions over descent sequences and local optimum approach the deterministic ones. In LES the next query minimizes the entropy of the descent sequences.

To ensure tractability, LES combines analytic predictive entropy calculations with Monte-Carlo conditioning over sampled descent sequences. We evaluate LES against state-of-the-art local and global BO methods on higher-dimensional synthetic objectives with varying complexity and policy search tasks. LES achieves lower simple and cumulative regret with fewer evaluations especially in high-complexity tasks. In summary, our contributions are:

- We formulate *local entropy search* (LES) as an information-theoretic Bayesian optimization paradigm that targets the terminal iterate of an arbitrary optimizer, making LES the first entropy-based approach explicitly focusing on local optima rather than the global optimum.
- We present a computationally lightweight instantiation of LES as an active learning problem over descent sequences by propagating the posterior belief through the optimizer via Monte-Carlo approximation based on efficient GP sampling (Wilson et al., 2020).
- We provide empirical evidence that this LES instantiation surpasses both global entropy-search baselines and existing local BO methods on high-complexity benchmarks.

In addition, we adapt a recent stopping criterion for BO (Wilson, 2024) to the LES setting, guaranteeing a probabilistic *local regret* bound (see Appx. E).

## 2 PRELIMINARIES

In this section, we discuss the related work on local BO and entropy search and briefly introduce GPs and an analytical approximation of their sampling paths.

### 2.1 RELATED WORK

Bayesian optimization is a popular method for many challenging real-world applications such as AutoML (Barbudo et al., 2023), drug discovery (Colliandre & Muller, 2023), and policy search (Paulson et al., 2023). For a recent introduction and overview see (Garnett, 2023). Below we present related work in entropy search and local BO – the two BO subfields most relevant to LES.

**Entropy Search** Global entropy search acquisition functions use an information-theoretic perspective to select the next BO query point (See Appx. A for a detailed introduction). They find a point that maximizes the expected information gain about properties of the global optimum. The original entropy search (ES) (Hennig & Schuler, 2012) and predictive entropy search (PES) (Hernández-Lobato et al., 2014) maximize the information gain about the *location* of the optimum. In contrast, max-value entropy search (MES) (Wang & Jegelka, 2017) maximize the information gain about the *function value* of the optimum. MES is computationally more efficient than both ES and PES. Joint entropy search (Hvarfner et al., 2022; Tu et al., 2022) maximizes the information gain about the joint distribution of *location* and *function value* of the optimum. Entropy search has been extended to constrained (Perrone et al., 2019), multi-objective (Belakaria et al., 2020), and multi-fidelity (Marco et al., 2017) optimization. With LES we propose the first local version of entropy search by targeting a local optimum instead of a global one.

**Local BO** There are two main approaches commonly used for local BO, *trust regions* (Akrour et al., 2017; Fröhlich Lukas P. et al., 2019; Eriksson et al., 2019) and line search (Kirschner et al., 2019; Müller et al., 2021; Nguyen et al., 2022; Wu et al., 2023; Fan et al., 2024). Trust-region BO such as TuRBO (Eriksson et al., 2019) explicitly maintain a subset of the search space $\mathcal{X}$ and restricts queries to be within this subset. In contrast, BO methods based on line search maintain an incumbent solution and iteratively choose a search direction and step size to improve the candidate. For example, gradient information BO (GIBO) (Müller et al., 2021) leverages the GP model of the objective's gradient $\nabla f$ to find its search direction and step size. Similar to these methods, LES starts its search from an incumbent solution but in contrast to GIBO and its variants does not use the learned gradient to update it but instead learns about the entire descent sequence. The local BO methods above have been shown to outperform global BO methods in high-complexity tasks.

Recent works showed that vanilla Bayesian optimization can solve high-dimensional problems if model complexity, specifically length scales of the GP kernel, is chosen appropriately (Hvarfner et al., 2024; Xu et al., 2025). While assumed model complexity is an important consideration for BO, the design of suitable prior assumption can benefit local and global BO methods and the acquisition strategy is mostly orthogonal. In Section 6, we show empirically that local BO and especially LES outperforms global BO for complex and higher-dimensional tasks.

## 2.2 GPs and Efficient Posterior GP Sampling

In this paper, BO uses a Gaussian Process (GP) (Rasmussen & Williams, 2006) as a fast-to-evaluate probabilistic surrogate for the unknown scalar function $f(\boldsymbol{x})$. We denote a GP as

$$p(f) = \mathcal{GP}(f; \mu, k), \tag{1}$$

where $\mu : \mathcal{X} \to \mathbb{R}$ is the prior mean and $k : \mathcal{X} \times \mathcal{X} \to \mathbb{R}$ is the prior covariance function, with

$$\mu(\boldsymbol{x}) = \mathbb{E}[f(\boldsymbol{x}) \mid \boldsymbol{x}], \quad k(\boldsymbol{x}, \boldsymbol{x}') = \mathbb{E}[(f(\boldsymbol{x}) - \mu(\boldsymbol{x}))(f(\boldsymbol{x}') - \mu(\boldsymbol{x}'))], \tag{2}$$

where $\mathbb{E}$ is the expectation. Without loss of generality, we assume $\mu(\boldsymbol{x}) = 0$. Given a dataset of noisy observations $y(\boldsymbol{x}) = f(\boldsymbol{x}) + \epsilon_t$, where the noise realization $\epsilon_t$ is modeled as a draw from a normal distribution $\mathcal{N}(0, \sigma_n^2)$, denoted as $\mathcal{D}_t = \{(\boldsymbol{x}_1, y_1), ..., (\boldsymbol{x}_t, y_t)\}$ the posterior belief over a function value $f(\boldsymbol{x})$ is a normal distribution denoted as

$$p(f(\boldsymbol{x}) \mid \mathcal{D}_t) = \mathcal{N}\left(f(\boldsymbol{x}); \mu(\boldsymbol{x} \mid \mathcal{D}_t), k(\boldsymbol{x}, \boldsymbol{x} \mid \mathcal{D}_t)\right). \tag{3}$$

Here, we denote $\mu(\boldsymbol{x} \mid \mathcal{D}_t)$ as the posterior mean and $k(\boldsymbol{x}, \boldsymbol{x}' \mid \mathcal{D}_t)$ as the posterior covariance between two points $\boldsymbol{x}$ and $\boldsymbol{x}'$. The predictive variance for a noisy observation is $\sigma_y^2(\boldsymbol{x} \mid \mathcal{D}_t) = k(\boldsymbol{x}, \boldsymbol{x} \mid \mathcal{D}_t) + \sigma_n^2$. See (Rasmussen & Williams, 2006) for details on GP regression.

To draw samples from the posterior distribution of objective functions $p(f \mid \mathcal{D}_k)$ we follow the analytical approximation proposed by (Wilson et al., 2020) that relies on Matheron's rule (Journel & Huijbregts, 1978) (see Appx. C.1). We denote $f_t^l$ as a sample from $p(f \mid \mathcal{D}_t)$:

$$f_t^l(\cdot) = \underbrace{\sum_{i=1}^{I} w_i^l \phi_i(\cdot)}_{\text{weight-space prior}} + \underbrace{\sum_{j=1}^{t} v_j k\left(\cdot, \boldsymbol{x}_j\right)}_{\text{function-space update}}, \tag{4}$$

where $w_i^l$ are randomly drawn weights, $\phi_i(\cdot)$ are basis functions and $v_j$ is calculated from the training-data covariance matrix. The sample approximations of (4) are analytic which means we can easily differentiate with respect to $\boldsymbol{x}$ and apply iterative optimizers such as gradient descent. LES can be applied to other probabilistic models from which we can efficiently draw posterior samples, e.g., variational Bayesian last layer models (Brunzema et al., 2024).

## 3 Problem Statement

The local black-box optimization problem is to find the best reachable solution from a given initial design $\boldsymbol{x}_0$

$$\text{given } \boldsymbol{x}_0 \in \mathcal{X}, \text{ find } \boldsymbol{x}^* = \underset{\boldsymbol{x} \in \mathcal{X}(\boldsymbol{x}_0) \subseteq \mathcal{X}}{\arg\min} f(\boldsymbol{x}), \tag{5}$$

where $f : \mathcal{X} \rightarrow \mathbb{R}$ is the black-box objective function and $\mathcal{X}(\boldsymbol{x}_0)$ is some neighborhood around $\boldsymbol{x}_0$.

We consider local optimization where an iterative (and possibly stochastic) optimization routine $\mathcal{O}$ generates a sequence of iterates – the *descent sequence* – converging to a local optimum $\boldsymbol{x}^*$ as

$$\mathcal{O}_{\boldsymbol{x}_0} : \mathcal{F}(\mathcal{X}) \longrightarrow \mathcal{X}^N, \qquad \mathcal{O}_{\boldsymbol{x}_0}(f) := (\boldsymbol{z}_0 = \boldsymbol{x}_0, \boldsymbol{z}_1, \dots) \subset \mathcal{X} \tag{6}$$

where

$$\boldsymbol{x}^* = \lim_{n \to \infty} \boldsymbol{z}_n, \tag{7}$$

or $\boldsymbol{x}^* = \boldsymbol{z}_N$ for finite descent sequences. We assume that the chosen optimizer converges for all sample paths of the GP prior (1). For instance, under suitable assumptions, gradient descent with an appropriate step size $\eta$ produces the convergent sequence

$$\mathrm{GD}_{\boldsymbol{x}_0}(f) := \left\{ \boldsymbol{z}_0 = \boldsymbol{x}_0, \, \boldsymbol{z}_1 = \boldsymbol{z}_0 - \eta \nabla f(\boldsymbol{z}_0), \, \boldsymbol{z}_2 = \boldsymbol{z}_1 - \eta \nabla f(\boldsymbol{z}_1), \dots \right\}. \tag{8}$$

When $f$ is known we can directly apply $\mathcal{O}$. However, we are in the standard BO black-box setting, where we cannot directly create this sequence. We instead consider the bandit setting with sequential queries to an expensive-to-evaluate black-box function with noisy zero-order evaluations $y_t = f(\boldsymbol{x}_t) + \epsilon_t$ at locations $\boldsymbol{x}_t$. Therefore, the objective of this work is to find a practical strategy that best approximates the solution to (5) in a data-efficient manner by learning about the descent sequence.

## 4 ENTROPY MINIMIZATION OF UNCERTAIN DESCENT SEQUENCES

**Overview.** We aim to apply the entropy search principle not to the global optimum but to the *descent sequence* started from $\boldsymbol{x}_0$. **Goal:** Choose the next query $\boldsymbol{x}$ that most reduces uncertainty about the entire descent sequence starting from $\boldsymbol{x}_0$, i.e., the sequence generated by an iterative optimizer $\mathcal{O}$. Once we know the descent sequence, we also know the local optimum $\boldsymbol{x}^*$. **Idea:** Treat the descent sequence itself as the object of interest and apply the entropy-search principle to it: select $\boldsymbol{x}$ to maximize information gain about that sequence. We target the mutual information between $(\boldsymbol{x}, y(\boldsymbol{x}))$ and the random descent sequence induced by the GP posterior and $\mathcal{O}$. The remainder of this section formalizes this idea. After the formalization we give a practical algorithm in Sect. 5.

Given $p(f)$ as a distribution over functions, applying an optimizer $\mathcal{O}$ from $\boldsymbol{x}_0$ induces a random descent sequence with observations $R$, as

$$Q_{\boldsymbol{x}_0} = ((\boldsymbol{z}_0, R_0), (\boldsymbol{z}_1, R_1), \dots). \tag{9}$$

where $\boldsymbol{z}_n$ are iterates and $R_n$ are some observation of the objective function $f$ at $\boldsymbol{x}$ depending on $\mathcal{O}$. Examples are function values $R_n := f(\boldsymbol{z}_n)$ (e.g., hill climbing) or gradient information $R_n := \nabla f(\boldsymbol{z}_n)$ (e.g. gradient descent). Reducing the uncertainty about $Q_{\boldsymbol{x}_0}$ also reduces uncertainty about the local optimum $\boldsymbol{x}^*$ (see Fig. 1).

To apply the entropy search framework (Hennig & Schuler, 2012) to this problem, we minimize the entropy $\mathrm{H}(Q_{\boldsymbol{x}_0})$ of the descent sequence. Minimizing the entropy with a new observation is equivalent to maximizing the mutual information between the new observation $(\boldsymbol{x}, y(\boldsymbol{x}))$ and the descent sequence $Q_{\boldsymbol{x}_0}$ conditioned on the current data $\mathcal{D}_t$. Therefore, in LES we reduce entropy over $Q_{\boldsymbol{x}_0}$ by selecting queries that maximize mutual information with this distribution, as

$$\alpha_{\mathrm{LES}}(\boldsymbol{x}) = I\left((\boldsymbol{x}, y(\boldsymbol{x})); Q_{\boldsymbol{x}_0} \mid \mathcal{D}_t\right), \tag{10}$$

which we reformulate into a tractable form in Sect. 5. Note, that the observations $R$ needs to contain the information the iterative optimizer requires to determine the next iterate. Therefore, (10) requires knowledge about the inner workings of the iterative optimizer to determine $Q$. We will discuss the case of gradient descent sequences as an example at the end of this section.

*Remark* 1. An alternative formulation for LES is to define the random variable $O_{\boldsymbol{x}_0}^*$ directly over the *local optimum* $\boldsymbol{x}^*$ of $\mathcal{O}_{\boldsymbol{x}_0}(f)$. The distribution of $O_{\boldsymbol{x}_0}^*$ is the push-forward of $p(f)$ under $\mathcal{O}_{\boldsymbol{x}_0}$ and taking the limit as in (7). Unfortunately, the mutual information

$$I\left((\boldsymbol{x}, y(\boldsymbol{x})); O_{\boldsymbol{x}_0}^* \mid \mathcal{D}_t\right). \tag{11}$$

is not tractable in the general case since we would need to condition a GP on the event $O_{\boldsymbol{x}_0}^* = \boldsymbol{x}^*$ which involves the entire descent sequences (see Appx. B.1). We propose and evaluate an alternative approximation in Appx. F.2.

A tractable special case is gradient descent that terminates after the first step. The resulting acquisition function is GIBO (Müller et al., 2021) with entropy instead of the trace of the covariance matrix (see Appx. B.2).

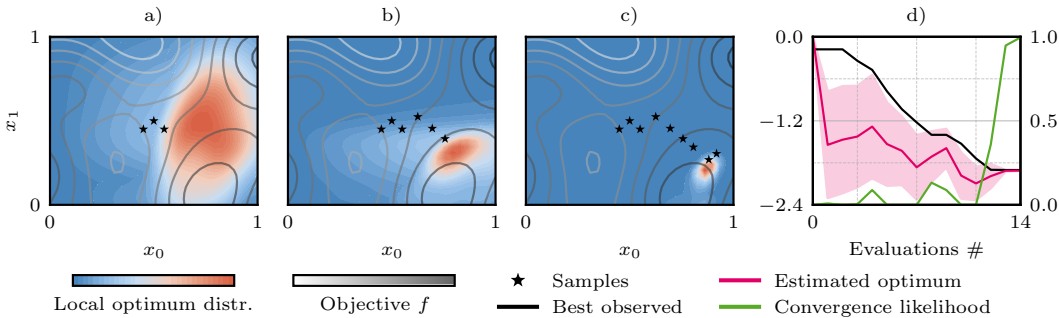

Figure 2: **Illustration of LES on a 2D example:** *a)* After three initial evaluations, the distribution over reachable local optima is wide. *b, c)* As LES selects new points, evaluations concentrate near the descent sequence, and the distribution of the local optimum narrows. *d)* Convergence behavior of LES. After 14 evaluations, the convergence criterion (see Appx. E.1) stops the optimization.

**Local Entropy Search with Gradient Descent Sequences.** Before we move on to the next section, we discuss the LES acquisition function for the gradient descent algorithm. The descent sequence is defined through the initial design $\boldsymbol{x}_0$ and the gradient of the function $\nabla f$ at the locations $\boldsymbol{z}_n$ as $\boldsymbol{z}_i = \boldsymbol{z}_{i-1} - \eta \nabla f(\boldsymbol{z}_{i-1})$ with $\boldsymbol{z}_0 = \boldsymbol{x}_0$. If $(\boldsymbol{z}_n)_{n \geq 0}$ converges, then $\boldsymbol{x}^* = \boldsymbol{z}_0 + \lim_{n \to \infty} \sum_{i=0}^{n} -\eta \nabla f(\boldsymbol{z}_i)$.

Since we can easily condition a GP on gradient information (Rasmussen & Williams, 2006) a LES acquisition function for local optimization via gradient descent is

$$\alpha_{\text{LES-GD}}(\boldsymbol{x}) = I\left((\boldsymbol{x}, y(\boldsymbol{x})); Q_{\boldsymbol{x}_0} \mid \mathcal{D}_t\right), \tag{12}$$

with $Q_{\boldsymbol{x}_0} = ((\boldsymbol{z}_0 = \boldsymbol{x}_0, \nabla f(\boldsymbol{z}_0)), (\boldsymbol{z}_1, \nabla f(\boldsymbol{z}_1)), \dots)$. In words: We are looking for the query whose outcome will reveal the most information about *gradients* of the function at the (distribution of) locations of the descent sequence. Generally, the design of the sequence $Q_{\boldsymbol{x}_0}$ is dependent on the optimizer choice $\mathcal{O}$ and the properties of the GP samples (see Appx. C.2). The performance and computational burden of LES depends on the design choices. We investigate alternative choices for gradient descent in Section 6.

*Remark* 2. For GPs with a squared exponential kernel the push-forward of gradient-descent sequences is dense in $\mathcal{X}$. This means, in principle, there can be sequences arbitrarily close to any point in the domain and no region is a priori "forbidden." In addition, all descent sequences are possible under this prior. For details, see Appx. G.2.

## 5 LOCAL ENTROPY SEARCH

This section casts the general local entropy search paradigm (Sec. 4) into a practical algorithm; see Fig. 2 for an illustrative example and Algorithm 1. We first derive the acquisition function (Sec. 5.1), afterwards we describe how to approximate the distribution of gradient descent sequences and how to condition on them (Sec. 5.2).

### 5.1 THE LES ACQUISITION FUNCTION

The LES acquisition function follows the entropy-search principle: ask where an observation $(\boldsymbol{x}, y(\boldsymbol{x}))$ would tell us most about the optimizer's descent sequence $Q_{\boldsymbol{x}_0}$ – their mutual information. In practice this means comparing two entropies: (i) the predictive entropy at $\boldsymbol{x}$, which measures the overall uncertainty about $y(\boldsymbol{x})$ under the GP, and (ii) the expected entropy that remains if we condition on how $\boldsymbol{x}$ would influence the descent sequences drawn from the GP posterior.

$$\alpha_{\text{LES}}(\boldsymbol{x}) = I\left((\boldsymbol{x}, y(\boldsymbol{x})); Q_{\boldsymbol{x}_0} \mid \mathcal{D}_t\right)$$
$$= \underbrace{\text{H}[y(\boldsymbol{x}) \mid \mathcal{D}_t]}_{\text{predictive entropy}} - \underbrace{\mathbb{E}_f\left[\text{H}\left[y(\boldsymbol{x}) \mid \mathcal{D}_t, Q_{\boldsymbol{x}_0}\right]\right]}_{\text{conditional entropy}}. \tag{13}$$

The predictive entropy (see Fig. 3, b)) can be calculated in closed form, as

$$\text{H}[y(\boldsymbol{x}) \mid \mathcal{D}_t] = \frac{1}{2} \log\left(2\pi \, e \, \sigma_y^2(\boldsymbol{x} \mid \mathcal{D}_t)\right). \tag{14}$$

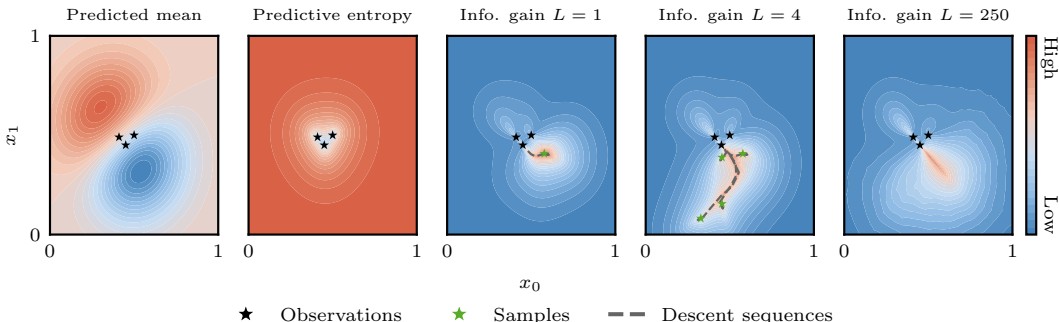

Figure 3: **Illustration of the LES acquisition function after three evaluations:** *Left:* The GP posterior mean after conditioning on the observations. *Second:* The predictive entropy is high in regions with large posterior variance. *Third to fifth:* The information gain between sampled descent sequences and query locations in $\mathcal{X}$ (the LES acquisition function) is high at points that are far from existing observations and aligned with likely descent sequences.

The expectation over the conditional entropy is more challenging. We do not have a closed form expression of the distribution of $Q_{\boldsymbol{x}_0}$ as this would mean to apply the optimization routine $\mathcal{O}$ to the distribution $p(f|\mathcal{D}_t)$. Thus, we cannot analytically calculate the expectation from it. Therefore, we approximate it by $L$ Monte-Carlo samples:

$$\mathbb{E}_f\left[\mathrm{H}\left[p\left(y(\boldsymbol{x}) \mid \mathcal{D}_t, Q_{\boldsymbol{x}_0}\right)\right]\right] \approx \frac{1}{L}\sum_{l=1}^{L}\mathrm{H}\left[p\left(y(\boldsymbol{x}) \mid \mathcal{D}_t, Q_{\boldsymbol{x}_0}^l\right)\right]. \tag{15}$$

To approximate $Q_{\boldsymbol{x}_0}$, we first draw $L$ sample paths from the posterior GP according to (4) to retrieve samples $f_t^1, ..., f_t^l, ..., f_t^L$. Then we apply the optimization routine $\mathcal{O}$ to each sample to retrieve the descent sequences $Q_{\boldsymbol{x}_0}^l = \left((\boldsymbol{z}_0^l, R_0^l), (\boldsymbol{z}_1^l, R_1^l), ...\right)$ and the local optima $\boldsymbol{x}^{l,*}$.

For example, in the case of gradient descent, we get $Q_{\boldsymbol{x}_0}^l = \left((\boldsymbol{z}_0^l = \boldsymbol{x}_0, R_0^l = \nabla f^l(\boldsymbol{x}_0)), (\boldsymbol{z}_1^l = \boldsymbol{x}_0 - \nabla f^l(\boldsymbol{x}_0), R_1^l = \nabla f^l(\boldsymbol{z}_1^l)), ...\right)$.

To condition on the parameter observation pairs $Q_{\boldsymbol{x}_0}^l$, we add them to the already existing observations and analytically compute the predictive posterior variance and entropy. Note that we approximate $Q_{\boldsymbol{x}_0}^l$ with a finite sequence to compute (15) (see Sec. 5.2). By inserting (14) and the approximation (15) into (13) we get the final LES acquisition function (see Fig. 3):

$$\alpha_{\mathrm{LES}}(\boldsymbol{x}) \approx \frac{1}{2}\log\left(2\pi e\,\sigma_y^2(\boldsymbol{x} \mid \mathcal{D}_t)\right) - \frac{1}{L}\sum_{l=1}^{L}\frac{1}{2}\log\left(2\pi e\,\sigma_y^2\left(\boldsymbol{x} \mid \mathcal{D}_t \cup Q_{\boldsymbol{x}_0}^l\right)\right). \tag{16}$$

Extension of the LES acquisition function to the batch case is straight forward (see Appx. C.3).

---

**Algorithm 1** Local Entropy Search with stopping rule

---

1: **Input**: initial design and corresponding observation $\mathcal{D}_1 = (\boldsymbol{x}_0, y_0)$, local optimizer $\mathcal{O}$
2: **for** $t \in 1, 2, ...$ **do**
3:     $\mathcal{GP}_t \leftarrow$ fit GP model of $f(\boldsymbol{x})$ using $\mathcal{D}_t$ with MAP hyperparameter optimization
4:     $\hat{\boldsymbol{x}}_t^*, \hat{f}_t^* \leftarrow$ identify current optimum from $\mathcal{D}_t$
5:     $f^1, ..., f^L \leftarrow$ draw $L$ samples from $\mathcal{GP}_t$                                                     ▷ cf. (4)
6:     **for** $l \in 1, ..., L$ **do** $Q^l \leftarrow$ apply $\mathcal{O}$ to $f^l$ starting from $\hat{\boldsymbol{x}}_t^*$
7:     $\boldsymbol{x}_{t+1} = \mathrm{argmax}_{\boldsymbol{x} \in Q^1, ..., Q^L}\,\alpha_{\mathrm{LES}}(\boldsymbol{x})$            ▷ Maximize LES acquisition function cf. Sec. 5.1
8:     **for** $l \in 1, ..., L$ **do** $r_t^l = f^l(\hat{\boldsymbol{x}}_t^*) - f^l(\boldsymbol{x}^{l,*})$                     ▷ Estimate local regret
9:     **if** $\sum_{l=1}^{L}\mathbb{1}\left(r_t^l \le \epsilon\right) \ge k_{\max}$ **do** stop       ▷ Stopping rule for probabilistic regret. See Appx. E.
10:    $y_{t+1} \leftarrow$ evaluate objective with $\boldsymbol{x}_{t+1}$
11:    $\mathcal{D}_{t+1} \leftarrow \mathcal{D}_t \cup \{(\boldsymbol{x}_{t+1}, y_{t+1})\}$
12: **end for**
13: **return** $\hat{\boldsymbol{x}}^*$

---

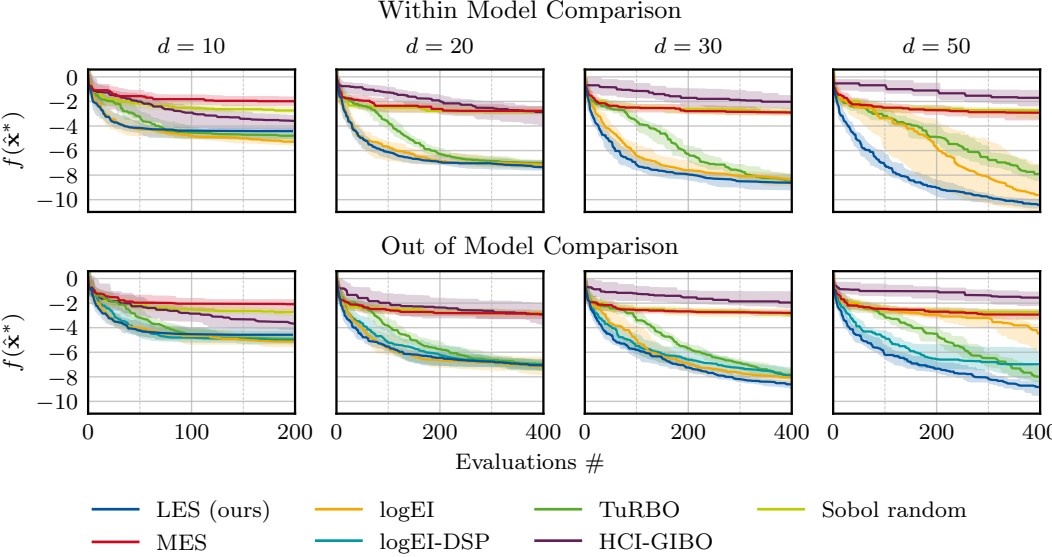

Figure 4: **Optimizing Gaussian Process Samples:** Median, 25-, and 75-percent quantiles for the best function values found for 20 sampled objective functions with medium complexity (see Tab. 1). LES outperforms baselines as dimensionality increases.

## 5.2 ADDITIONAL APPROXIMATIONS AND IMPLEMENTATION DETAILS

To make LES a practical algorithm we introduce additional approximations and design choices. For hyperparameter values, we refer to Appx. D.1. To solve the original problem (see Sec. 3) we need to apply the local optimizer from a fixed initial design $\boldsymbol{x}_0$. However, in practice, we always start the descent sequence from the current best guess $\hat{\boldsymbol{x}}_t^*$. Additionally, instead of ensuring convergence of the local optimizer, we simply stop all the iterative optimizers after finitely many steps. Still, conditioning on all elements in $Q_{\boldsymbol{x}_0}^l$ can be prohibitively expensive. Thus, we choose to only condition on $P$ equally spaced elements along the interpolated descent sequence. Additionally, we condition on function values instead of gradient observations which reduces runtime while achieving similar performance (see Appx. D.5). Optimizing any acquisition function is challenging in high-dimensions as it is non-convex (see Fig. 3). Fortunately, the approximation of $Q_{\boldsymbol{x}_0}$ gives us access to promising candidates. Thus we optimize the acquisition function under the finite candidate set $Q_{\boldsymbol{x}_0}^1, ..., Q_{\boldsymbol{x}_0}^L$ with $L$ times $P$ candidate points.

**Practical LES in one paragraph** (i) Draw $L$ posterior GP samples. (ii) For each sample, run $\mathcal{O}$ for a finite number of steps starting at the current incumbent $\hat{\boldsymbol{x}}_t^*$ to obtain a descent sequence; discretize each sequence to $P$ support points. (iii) Compute the predictive entropy at candidate $\boldsymbol{x}$ and the average conditional entropy after conditioning on those discretized sequences; their difference is the acquisition function in (16) (Fig. 2). (iv) To keep optimization tractable, maximize the acquisition over the finite candidate set given by the union of all discretized sequences. (v) Evaluate, update the GP, and repeat (Alg. 1).

## 6 EMPIRICAL RESULTS

In this section, we empirically evaluate LES and compare it against other local and global BO variants (additional results in Appx. D). As objectives, we use GP-samples with varying lengthscales to increase complexity as well as synthetic and application-oriented benchmarks.

### 6.1 ABLATIONS AND BENCHMARK ALGORITHMS

**LES Variants** We evaluate LES with three local optimization algorithms: gradient descent, ADAM (Kingma & Ba, 2015), and covariance matrix adaptive evolutionary search (CMA-ES) (Hansen

Table 1: **Within-Model Comparisons:** Median final objective for LES, logEI, TuRBO, and Sobol random sampling. Bold indicates performance not statistically significantly below the best. *Hyperprior as in (Hvarfner et al., 2024).

| Complexity | Method | $d=5$ | $d=10$ | $d=20$ | $d=30$ | $d=50$ |
|---|---|---|---|---|---|---|
| **high**: $\mathrm{p}(l;d) =$ $\mathrm{logn}(-2.5\sqrt{2}+\log\sqrt{d}, \sqrt{3}/5)$ $\mathrm{E}[\mathrm{p}(l;50)]=0.25$ | LES (ours) | $-2.8$ | $\mathbf{-4.8}$ | $\mathbf{-7.4}$ | $\mathbf{-9.0}$ | $\mathbf{-10.8}$ |
| | logEI | $\mathbf{-3.9}$ | $-4.4$ | $-2.9$ | $-3.2$ | $-2.9$ |
| | TuRBO | $-3.5$ | $\mathbf{-4.8}$ | $\mathbf{-7.2}$ | $-8.0$ | $-7.1$ |
| | Sobol random | $-2.4$ | $-2.7$ | $-2.8$ | $-3.2$ | $-3.0$ |
| **medium**: $\mathrm{p}(l;d) =$ $\mathrm{logn}(-2.0\sqrt{2}+\log\sqrt{d}, \sqrt{3}/4)$ $\mathrm{E}[\mathrm{p}(l;50)]=0.52$ | LES (ours) | $-2.9$ | $-4.4$ | $\mathbf{-7.3}$ | $\mathbf{-8.6}$ | $\mathbf{-10.4}$ |
| | logEI | $\mathbf{-3.6}$ | $\mathbf{-5.3}$ | $-7.1$ | $-8.4$ | $-9.6$ |
| | TuRBO | $-3.1$ | $-4.8$ | $\mathbf{-7.0}$ | $\mathbf{-8.5}$ | $-7.9$ |
| | Sobol random | $-2.5$ | $-2.7$ | $-2.8$ | $-2.9$ | $-2.7$ |
| **low**: $\mathrm{p}(l;d) =$ $\mathrm{logn}(-1.0\sqrt{2}+\log\sqrt{d}, \sqrt{3}/2)$ $\mathrm{E}[\mathrm{p}(l;50)]=2.65$ | LES (ours) | $-2.1$ | $-3.7$ | $-5.5$ | $\mathbf{-6.8}$ | $\mathbf{-8.8}$ |
| | logEI | $\mathbf{-2.9}$ | $\mathbf{-4.1}$ | $\mathbf{-5.8}$ | $-6.6$ | $-8.5$ |
| | TuRBO | $-2.4$ | $-3.6$ | $-5.5$ | $\mathbf{-6.5}$ | $-8.1$ |
| | Sobol random | $-2.0$ | $-2.3$ | $-2.9$ | $-2.8$ | $-3.0$ |
| **extremely low**\*: $\mathrm{p}(l;d) =$ $\mathrm{logn}(1.0\sqrt{2}+\log\sqrt{d}, \sqrt{3})$ $\mathrm{E}[\mathrm{p}(l;50)]=96.2$ | LES (ours) | $\mathbf{-0.6}$ | $-0.8$ | $\mathbf{-1.2}$ | $\mathbf{-1.5}$ | $\mathbf{-3.0}$ |
| | logEI | $\mathbf{-0.6}$ | $\mathbf{-0.9}$ | $\mathbf{-1.3}$ | $\mathbf{-1.5}$ | $\mathbf{-3.0}$ |
| | TuRBO | $-0.6$ | $-0.8$ | $-1.2$ | $-1.5$ | $-2.8$ |
| | Sobol random | $-0.5$ | $-0.7$ | $-1.0$ | $-1.0$ | $-1.5$ |

& Ostermeier, 2001). Results show that LES-ADAM and LES-GD perform best depending on problem complexity, while LES-CMA-ES shows a more global search behavior which is beneficial for low dimensional problems (see Appx. D.7). Additionally, we show that the more accurate the acquisition function approximation (higher $L$ and $P$), the better the performance at the cost of higher runtime. Below, we investigate LES-ADAM with a medium approximation accuracy resulting in an average runtime of $17.8\,\mathrm{sec}$ per iteration in 50 dimensions excluding hyperparameter optimization (see Appx. D.5 for runtime comparisons).

**Baselines** We compare LES-ADAM to two other local BO paradigms: TuRBO (Eriksson et al., 2019) is based on trust-regions and high-confidence improvement Bayesian optimization (HCI-GIBO) is a gradient-based approach. HCI-GIBO (He et al., 2025) is a recent GIBO (Müller et al., 2021) extension. As a global ES alternative we choose max-value entropy search (MES) (Wang & Jegelka, 2017). Additionally, we compare to logEI (Ament et al., 2023) and Sobol random sampling.

**Other Local Information-Theoretic Strategies** We propose and evaluate two other search strategies as special cases of the LES paradigm: local Thompson sampling and conditioning only on the final point of the descent sequence. They empirically perform worse confirming that conditioning on the whole sequences and Monte-Carlo sampling (see Appx. F.2).

## 6.2 GAUSSIAN PROCESS SAMPLES

**Model Complexity** Recent work by (Hvarfner et al., 2024) highlights model complexity as a key factor in high-dimensional BO performance. The assumed difficulty of the problem – encoded through the model complexity in the form of length scale priors – determines global BO performance. Smaller length scales yield more complex functions with more local optima (Adler, 2010, Chapt. 6). Building on this insight, we construct benchmark functions with varying problem difficulty by sampling functions from a GP with different model complexities. Specifically, we generate functions by scaling the log-normal length scale prior $p(l)$ proposed in (Hvarfner et al., 2024). We consider four complexity levels across 5 to 50 dimensions (see Tab.1) where the lowest model complexity corresponds to the one used in (Hvarfner et al., 2024).

**Within Model Comparison** To assess the performance of the proposed acquisition function independently of the effects of hyperparameter optimization, we first use known hyperparameters in all evaluated BO algorithms. Results on medium model complexity (Fig. 4) show that, after 400 evaluations, LES outperforms all baselines as dimensionality increases. The asymptotical

Table 2: **Out-Of-Model Comparisons:** Median final objective for LES, logEI-DSP, TuRBO, and Sobol random sampling. Bold indicates performance not statistically significantly below the best.

| Complexity | Method | $d = 5$ | $d = 10$ | $d = 20$ | $d = 30$ | $d = 50$ |
|---|---|---|---|---|---|---|
| **high**: $\mathrm{p}(l; d) =$ $\mathrm{logn}(-2.5\sqrt{2} + \log\sqrt{d}, \sqrt{3}/5)$ $\mathrm{E}[p(l; 50)] = 0.25$ | LES | $-2.9$ | $\mathbf{-5.0}$ | $\mathbf{-7.2}$ | $\mathbf{-8.5}$ | $\mathbf{-7.8}$ |
| | TuRBO | $-3.7$ | $\mathbf{-5.0}$ | $\mathbf{-7.1}$ | $-8.2$ | $-7.1$ |
| | logEI-DSP | $\mathbf{-3.7}$ | $-4.1$ | $-4.0$ | $-4.0$ | $-4.1$ |
| | Sobol | $-2.4$ | $-2.7$ | $-2.8$ | $-3.2$ | $-3.0$ |
| **medium**: $\mathrm{p}(l; d) =$ $\mathrm{logn}(-2.0\sqrt{2} + \log\sqrt{d}, \sqrt{3}/4)$ $\mathrm{E}[p(l; 50)] = 0.52$ | LES | $-3.0$ | $-4.6$ | $\mathbf{-7.1}$ | $\mathbf{-8.6}$ | $\mathbf{-8.8}$ |
| | TuRBO | $-3.1$ | $-4.9$ | $\mathbf{-7.0}$ | $-7.9$ | $-8.0$ |
| | logEI-DSP | $\mathbf{-3.6}$ | $\mathbf{-5.0}$ | $-7.0$ | $-7.9$ | $-7.0$ |
| | Sobol | $-2.5$ | $-2.7$ | $-2.8$ | $-2.9$ | $-2.7$ |
| **low**: $\mathrm{p}(l; d) =$ $\mathrm{logn}(-1.0\sqrt{2} + \log\sqrt{d}, \sqrt{3}/2)$ $\mathrm{E}[p(l; 50)] = 2.65$ | LES | $-2.1$ | $-3.7$ | $-5.2$ | $\mathbf{-6.6}$ | $\mathbf{-8.5}$ |
| | TuRBO | $\mathbf{-2.4}$ | $-3.7$ | $-5.4$ | $\mathbf{-6.4}$ | $-7.7$ |
| | logEI-DSP | $\mathbf{-2.9}$ | $\mathbf{-4.1}$ | $\mathbf{-5.7}$ | $\mathbf{-6.6}$ | $-8.1$ |
| | Sobol | $-2.0$ | $-2.3$ | $-2.9$ | $-2.8$ | $-3.0$ |
| **extremely low**: $\mathrm{p}(l; d) =$ $\mathrm{logn}(1.0\sqrt{2} + \log\sqrt{d}, \sqrt{3})$ $\mathrm{E}[p(l; 50)] = 96.2$ | LES | $\mathbf{-0.6}$ | $\mathbf{-0.8}$ | $\mathbf{-1.2}$ | $\mathbf{-1.5}$ | $\mathbf{-2.9}$ |
| | TuRBO | $\mathbf{-0.6}$ | $\mathbf{-0.8}$ | $\mathbf{-1.2}$ | $\mathbf{-1.5}$ | $\mathbf{-2.9}$ |
| | logEI-DSP | $\mathbf{-0.6}$ | $\mathbf{-0.9}$ | $\mathbf{-1.3}$ | $\mathbf{-1.5}$ | $\mathbf{-3.0}$ |
| | Sobol | $\mathbf{-0.5}$ | $\mathbf{-0.7}$ | $-1.0$ | $-1.0$ | $-1.5$ |

performance benefits of searching globally with logEI is only seen for $d = 10$. MES, the global entropy search benchmark is not competitive. For other problem complexities the same trends can be observed (see Tab.1 and Appx. D.3). Performance differences increase with increasing problem complexity with LES clearly outperforming other methods for high complexity tasks in higher dimensions (see Appx. D.3).

**Out-Of-Model Comparison**   The out-of-model setup is identical to the within-model one except that GP hyperparameters are now estimated via MAP. All methods use the hyperprior that is used to sample from the GP, except for logEI-DSP, which was recently proposed by Hvarfner et al. (2024) and assumes a hyperprior with longer length scales. Appx. D.3 reports additional detailed results.

The results mirror the within-model case. Interestingly, logEI performs better when using the wrong (less complex) hyperprior, as in logEI-DSP which supports the claims made in (Hvarfner et al., 2024).

**Cumulative Regret**   The local search behavior inherent to LES leads to a substantially lower empirical cumulative regret for all problems except for $d = 5$ and the lowest complexity (Appx. D.3).

**A Stopping Criterion for Local Entropy Search**   We incorporate a probabilistic stopping rule for LES that assesses whether the current solution is locally optimal. Following the Monte-Carlo test of Wilson (2024), we estimate the probability that the local simple regret falls below a user-specified tolerance and terminate once this criterion is met. The full formulation, assumptions, and theoretical guarantees, along with empirical stopping-time results, are provided in Appx. E. When compared to the results in (Wilson, 2024) these results show that the local optimization needs fewer samples before stopping, reinforcing the intuition that reaching a local optimum is easier

### 6.3   SYNTHETIC AND APPLICATION-ORIENTED OBJECTIVE FUNCTIONS

We evaluate LES on additional analytic benchmarks (Fig. 5) and application-oriented tasks, with further results in Appx. D.4. On single-optimum functions (square), all methods reliably identify the solution, though LES and TuRBO achieve lower cumulative regret, highlighting the conservative exploration behavior inherent to local search. On the 30-d Ackley function, LES and TuRBO outperform global methods but LES shows high run-to-run variance due to the many local optima. For the rover (Wang et al., 2018) and Mopta08 (Jones, 2008) tasks, LES, logEI, and TuRBO perform similarly, with LES best on rover and logEI best on Mopta08.

Additional experiments, including benchmarks designed to expose weaknesses of local search are presented in Appx. D.4. In these cases, LES frequently gets trapped in local optima, leading to high

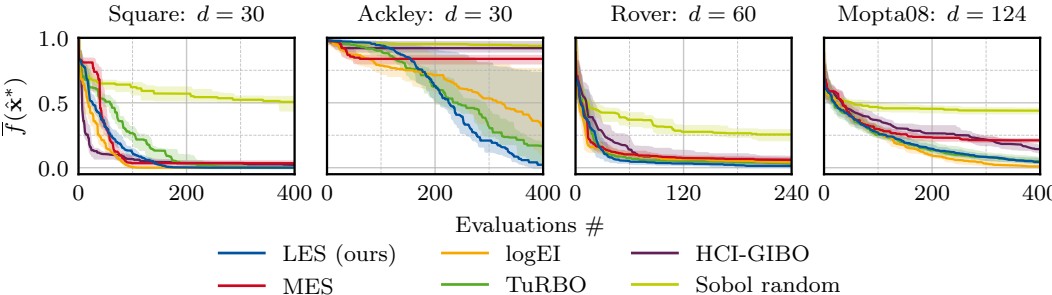

Figure 5: **Synthetic and Application-Oriented Objective Functions:** Median, 25-, and 75-percent quantiles for the best (normalized) function values found. Additional results in Appx. D.4.

run-to-run variance and overall poor performance relative to global methods. These results highlight the limitations of LES on highly multimodal landscapes with hard-to-escape local optima.

## 7  LIMITATIONS

We show that LES is beneficial especially in the high-dimensionality and high-complexity case. However, the focus on locality is also its most obvious constraint. Once the algorithm commits to a basin, it possesses no intrinsic mechanism for escape; achieving coverage of the full design space therefore requires globalization strategies. Furthermore, in real-world use cases it may be difficult to determine the problem complexity a-priori. Therefore, multi-starts, more advanced incumbent search, e.g., based on (Adebiyi et al., 2025), or switching heuristics between local and global optimization based on estimated objective complexity are interesting directions for future research. The stopping rule introduced in Appx. E.1 can be used to trigger a restart instead of stopping.

LES inherits the strengths and weaknesses of its surrogate. All acquisition decisions are driven by posterior samples; if the model is misspecified, the algorithm will exploit the wrong belief. Moreover, approximating the mutual information in (16) at each iteration requires drawing and optimizing posterior samples, with the number of samples increasing with dimensionality and model complexity.

The present formulation tackles unconstrained, single-objective optimization. LES is entropy-search based and therefore future work can naturally extend LES to more complex settings, such as multi-fidelity, constrained, batch (see Appx. C.3), or asynchronous optimization.

Finally, LES currently lacks finite-time regret guarantees of the GP-UCB type (Srinivas et al., 2012), and the current theoretical claims are limited to the high-probability certificate of local optimality offered by the stopping test. In addition, a bound on the difference between local and global regret for GP sample paths would bridge the current theoretical gap between local and global BO.

## 8  DISCUSSION AND CONCLUSION

This paper introduces LES, the first entropy-search paradigm tailored to local optimization. By propagating the GP posterior through the optimizer's descent sequence, LES selects each evaluation to maximize mutual information with that sequence, thereby reducing uncertainty over possible descent sequences.

Empirically, LES delivers strong sample efficiency. Across high-complexity GP samples and policy-search benchmarks, the ADAM-based variant consistently attains lower simple regret than global entropy-search baselines and other local BO strategies, especially as dimensionality and complexity grows. Additionally, we show in Appx. D.4 that local search has a more conservative exploration behavior than global BO. This can be a great asset when optimizing outside of simulated environments, as e.g., in real-world robot learning. A probabilistic stopping rule guarantees bounded local regret by detecting convergence to a local optimum without additional overhead by reusing the samples from the acquisition step (Appx. E).

## ACKNOWLEDGMENTS

We thank Johanna Menn, Paul Brunzema, and Friedrich Solowjow for the many helpful discussions. This work has been supported by the Robotics Institute Germany, funded by BMFTR grant 16ME0997K, and by the Deutsche Forschungsgemeinschaft (DFG, German Research Foundation) under Germany's Excellence Strategy - EXC-2023 Internet of Production - 390621612. Computations were performed with computing resources granted by RWTH Aachen University under project rwth1723.

## REPRODUCIBILITY STATEMENT

All code necessary to reproduce the results is available[2]. Additionally, we elaborate on approximations and implementation details in Sec. 5.2. We report hyperparameter values in Appx. D.

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

# Supplementary Material for "Local Entropy Search over Descent Sequences for Bayesian Optimization"

Following is the technical appendix. Note that all citations here are in the bibliography of the main document and similarly for many of the cross-references.

## A   Introduction to Bayesian Optimization with Entropy Search

Because evaluating $f(\boldsymbol{x})$ is costly, BO leverages all data obtained until iteration $t$ to choose the next parametrization $\boldsymbol{x}_{t+1}$ in Domain $\mathcal{X}$. After $t_{\text{init}}$ random samples are evaluated, BO takes two steps to maximize the utility of the next experiment. First, a GP is trained on all past observations to approximate $f(\boldsymbol{x})$. Second, this model is used in an acquisition function to balance exploration and exploitation. The acquisition function $\alpha$ uses the probabilistic GP predictions to calculate the utility of an experiment. It is maximized to find the next query:

$$\boldsymbol{x}_{t+1} = \text{argmax}_{\mathcal{X}} \alpha(\mathcal{GP}_k). \tag{17}$$

Approximately solving (17) is much easier than the original problem because only the fast-to-evaluate GP model needs to be evaluated. This new query is evaluated, new data is received, and the next iteration is started by again updating the GP model. This way, the GP model is iteratively refined in promising regions. We refer to (Garnett, 2023) for a detailed introduction to BO.

Global entropy search methods use an information-theoretic perspective to select where to evaluate. They find a query point that maximizes the expected information gain about the global optimum $\boldsymbol{x}_{g}^* = \arg\max \boldsymbol{x} \in \mathcal{X} f(\boldsymbol{x})$ whose value $f^* = f(\boldsymbol{x}^*)$ achieves the global maximum of the function $f$.

The original entropy search (ES) (Hennig & Schuler, 2012) and predictive entropy search (PES) (Hernández-Lobato et al., 2014) maximize the information gain about the *location* of the optimum:

$$\alpha_t(\boldsymbol{x}) = I\left(\{\boldsymbol{x}, y(\boldsymbol{x})\}; \boldsymbol{x}_g^* \mid D_t\right) \tag{18}$$

The random variable $y(\boldsymbol{x})$ denotes the predictive distribution of the noisy observation at the query location $\boldsymbol{x}$ and $\boldsymbol{x}_g^*$ denotes the estimated distribution of the global optimizer. The information gain can be expressed as the difference between predictive entropy of noisy observation at the query location, $\text{H}\left(p\left(\boldsymbol{x}_g^* \mid D_t\right)\right)$ and the expectation of the predictive conditioned on the distribution of minimizers $\mathbb{E}\left[\text{H}\left(p\left(\boldsymbol{x}_* \mid D_t \cup \{\boldsymbol{x}, y\}\right)\right)\right]$. Calculating those terms requires expensive approximations that do not scale well especially in high dimensions.

Max-value entropy search (MES) (Wang & Jegelka, 2017) maximize the information gain about the *function value* of the optimum:

$$\alpha_t(x) = I\left(\{\boldsymbol{x}, y\}; f_g^* \mid D_t\right) \tag{19}$$

This approach is computationally significantly more efficient than ES and PES, because the expectation and entropy need to be only calculated over the one dimensional distribution of optimum values.

Joint entropy search (JES) (Hvarfner et al., 2022; Tu et al., 2022) maximizes the information gain about the joint distribution of location *location* and *function value* of the optimizer

$$\alpha_{\text{JES}}(\boldsymbol{x}) = I\left((\boldsymbol{x}, y); \left(\boldsymbol{x}_g^*, f_g^*\right) \mid \mathcal{D}_n\right) \tag{20}$$

This paper applies the entropy search paradigm to local BO. Up to here, we only discussed entropy search for single objective optimization. It can be extended to other BO paradigms such as constrained (Perrone et al., 2019), multi-objective (Belakaria et al., 2020), and multi-fidelity (Marco et al., 2017) optimization.

# B    EXACT INFORMATION GAIN MAXIMIZATION OF THE LOCAL OPTIMIZER

In LES, we do not directly minimize the entropy of the solution of the local optimization algorithm but instead minimize the entropy of the descent sequence. This section shows why directly minimizing the entropy of the solution is not possible in general. Furthermore, we show that the entropy search version of GIBO is a special case for one step gradient descent, where it actually is possible.

## B.1    IMPRACTICABILITY IN THE GENERAL CASE

Suppose we want to directly minimize the entropy of the local optimizer:

$$\max_{x \in \mathcal{X}} I\left((\boldsymbol{x}, y(\boldsymbol{x})); O^*_{\boldsymbol{x}_0} \mid \mathcal{D}_t\right). \tag{21}$$

One way of going forward is to reformulate it in the standard entropy search way:

$$
I\left((\boldsymbol{x}, y(\boldsymbol{x})); O^*_{\boldsymbol{x}_0} \mid \mathcal{D}_t\right) \\
= \underbrace{\mathrm{H}[y(\boldsymbol{x}) \mid \mathcal{D}_t]}_{\text{predictive entropy}} - \mathbb{E}_f \underbrace{\left[\mathrm{H}\left[y(\boldsymbol{x}) \mid \mathcal{D}_t, O^*_{\boldsymbol{x}_0}\right]\right]}_{\text{conditional entropy}}. \tag{22}
$$

After Monte-Carlo approximation we get:

$$\mathbb{E}_f\left[\mathrm{H}\left[p\left(y(\boldsymbol{x}) \mid \mathcal{D}_t, O^*_{\boldsymbol{x}_0}\right)\right]\right] \approx \frac{1}{L} \sum_{l=1}^{L} \mathrm{H}\left[p\left(y(\boldsymbol{x}) \mid \mathcal{D}_t, O^{*,l}_{\boldsymbol{x}_0}\right)\right]. \tag{23}$$

Unfortunately, we can condition a GP efficiently only on point-wise observations of, for example, function values or gradients by adding them to the original data set as virtual points. It is not possible to condition a GP directly on $O^{*,l}_{\boldsymbol{x}_0}$ being a location that can be reached by gradient descent from $\boldsymbol{x}_0$. In Appendix F.2 we show how we can condition on $O^{*,l}_{\boldsymbol{x}_0}$ being a local optimum with zero gradient and positive Hessian. However, this loses information about the sequence to the local optimum. An alternative approach would be the following reformulation:

$$
\alpha(\boldsymbol{x}) = I\left((\boldsymbol{x}, y(\boldsymbol{x})); O^*_{\boldsymbol{x}_0} \mid \mathcal{D}_t\right) \\
= \underbrace{\mathrm{H}[O^*_{\boldsymbol{x}_0} \mid \mathcal{D}_t]}_{\text{predictive entropy}} - \mathbb{E}_{y(\boldsymbol{x})} \underbrace{\left[\mathrm{H}\left[O^*_{\boldsymbol{x}_0} \mid \mathcal{D}_t, y(\boldsymbol{x})\right]\right]}_{\text{conditional entropy}}. \tag{24}
$$

In principle, we can approximate the conditional entropy in (24) with Monte-Carlo approximation because we only have to condition a GP on realizations of $y(\boldsymbol{x})$. However, this is prohibitively expensive because it would require a new Monte-Carlo approximation of $O^*_{\boldsymbol{x}_0}$ to evaluate $\alpha$ at a new query location $\boldsymbol{x}$. Therefore, we do not consider this possibility any further.

## B.2    EXACT INFORMATION GAIN MAXIMIZATION IN ENTROPY-BASED GIBO

This paragraph shows that the information theoretic version CAGES (Tang & Paulson, 2024) of the GIBO (Müller et al., 2021) acquisition function can be seen as a special case of LES (11). This special case arises when considering a one step gradient descent algorithm that produces the following descent sequence

$$\mathrm{GS}_{\boldsymbol{x}_0}(f) := \left\{\boldsymbol{z}_0 = \boldsymbol{x}_0,\ \boldsymbol{z}_1 = \boldsymbol{z}_0 - \eta\nabla f(\boldsymbol{z}_0)\right\}. \tag{25}$$

Now suppose that the local optimum is the last element of that sequence $O^*_{\boldsymbol{x}_0} = \boldsymbol{x}_0 - \eta\nabla f(\boldsymbol{x}_0)$. Inserting in equation (24) yields:

$$
\alpha_{\mathrm{GES}}(\boldsymbol{x}) = \mathrm{H}[O^*_{\boldsymbol{x}_0} \mid \mathcal{D}_t] - \mathbb{E}_{y(\boldsymbol{x})}\left[\mathrm{H}\left[O^*_{\boldsymbol{x}_0} \mid \mathcal{D}_t, y(\boldsymbol{x})\right]\right] \\
= \mathrm{H}[\boldsymbol{x}_0 - \eta\nabla f(\boldsymbol{x}_0) \mid \mathcal{D}_t] - \mathbb{E}_{y(\boldsymbol{x})}\left[\mathrm{H}\left[\boldsymbol{x}_0 - \eta\nabla f(\boldsymbol{x}_0) \mid \mathcal{D}_t, y(\boldsymbol{x})\right]\right] \tag{26}
$$

Since $\boldsymbol{x}_0$ and $\eta$ are non-random variables, we get:

$$\alpha_{\mathrm{GES}}(\boldsymbol{x}) = \mathrm{H}[\nabla f(\boldsymbol{x}_0) \mid \mathcal{D}_t] - \mathbb{E}_{y(\boldsymbol{x})}\left[\mathrm{H}\left[\nabla f(\boldsymbol{x}_0) \mid \mathcal{D}_t \cup (\boldsymbol{x}, y(\boldsymbol{x}))\right]\right] \tag{27}$$

Inserting the entropy of a multivariate normal distribution, we get the original GES acquisition function (Tang & Paulson, 2024, Eq. (8)):

$$\alpha_{\mathrm{GES}}(\boldsymbol{x}) = \frac{1}{2}\log|\Sigma'(\boldsymbol{x}_0 \mid \mathcal{D}_t)| - \frac{1}{2}\log|\Sigma'(\boldsymbol{x}_0 \mid \mathcal{D}_t \cup (\boldsymbol{x}, y(\boldsymbol{x})))| \tag{28}$$

The covariance of the gradient at location $\boldsymbol{x}_0$ given data $\mathcal{D}_t$ is denoted as $\Sigma'\left(\boldsymbol{x}_0 \mid \mathcal{D}_t\right)$. Note that the expectation over $y(\boldsymbol{x})$ can be ignored because the predictive variance of the gradient is independent of $y(\boldsymbol{x})$ and only depends on $\boldsymbol{x}$. The small difference between the CAGES and the GIBO acquisition function is that the GIBO acquisition function uses the trace operator instead of the determinant operator $|\cdot|$ in (28).

This observation highlights that GIBO-style acquisition functions learn about one step of the gradient descent sequence, whereas LES maximizes the information gain about the whole descent sequence.

## C    ADDITIONAL DETAILS ON LES

### C.1    GP-SAMPLE APPROXIMATION STRATEGIES

In this work, we approximate GP sample paths using the decoupled method of (Wilson et al., 2020) (Eq. (4)), which we regard as state-of-the-art and computationally attractive. It has been shown to outperform alternatives, is readily available in TensorFlow and PyTorch, and is still recommended in recent tutorials on GP sampling (Do et al., 2025). We therefore do not empirically investigate alternative sampling strategies for LES.

By contrast, the GP-sample benchmarks in GIBO (Müller et al., 2021) and follow-up work (Nguyen et al., 2022) adopt a different approach: they sample the posterior at fixed locations and then re-interpolate these points with a GP. In our preliminary experiments, this led to overly smooth sample functions in low dimensions.

Exploring how different sampling strategies affect LES remains an interesting direction for future work, both to assess robustness and to better understand potential biases introduced by approximation schemes.

### C.2    RELATIONSHIP BETWEEN PROPERTIES OF TARGET FUNCTION, MODEL SAMPLES AND LOCAL OPTIMIZER

The choice of local optimizer $\mathcal{O}$ is constrained by the properties of the GP sample paths and practically by their approximations. For instance, if $\mathcal{O}$ is gradient-based, the sample paths $f^l$ must be at least once differentiable. More generally, the samples need to be differentiable to the same order required by $\mathcal{O}$. LES, however, is not limited to gradient-based methods: zeroth-order optimizers such as hill climbing or pattern search can be used when samples are non-differentiable. We illustrate this in Appx. D.7 with LES-CMAES, which employs a zeroth-order evolutionary optimizer.

Different kernels also practically affect optimizer choice. For example, Matérn-$1/2$ or Matérn-$3/2$ kernels often generate sample paths with many shallow local minima. In such cases, an optimizer that incorporates momentum (e.g., Adam) may help to avoid undesired convergence to these minima.

Crucially, these requirements apply to the GP sample paths, not to the true objective. Indeed, GPs can approximate sub-gradients of non-differentiable functions, as shown in (Wu et al., 2023).

### C.3    QLES: BATCHED LOCAL ENTROPY SEARCH

The LES paradigm can be straightforwardly extended to the batch case. This serves as an example of the versatility of entropy search methods and their potential for local optimization. We simply maximize the mutual information between the joint predictive distribution of multiple samples $y(\boldsymbol{x}_1), ..., y(\boldsymbol{x}_q)$ and the distribution of descent sequences. Equation (13) becomes:

$$
\begin{aligned}
\alpha_{\mathrm{qLES}}(\boldsymbol{x}_1, ..., \boldsymbol{x}_q) &= I\left(\boldsymbol{x}_1, ..., \boldsymbol{x}_q, y(\boldsymbol{x}_1), ..., y(\boldsymbol{x}_q)); Q_{\boldsymbol{x}_0} \mid \mathcal{D}_t\right) \\
&= \underbrace{\mathrm{H}[y(\boldsymbol{x}_1), ..., y(\boldsymbol{x}_q) \mid \mathcal{D}_t)]}_{\text{predictive entropy}} - \mathbb{E}_f \underbrace{\left[\mathrm{H}\left[y(\boldsymbol{x}_1), ..., y(\boldsymbol{x}_q) \mid \mathcal{D}_t, Q_{\boldsymbol{x}_0}\right]\right]}_{\text{conditional entropy}}.
\end{aligned} \tag{29}
$$

With this, the only change to the acquisition function calculation is the entropy calculation. The entropy of the predictive and conditional entropies can still be calculated in closed form and omitting the intermediate steps, the acquisition function (16) becomes:

$$
\begin{aligned}
\alpha_{\mathrm{qLES}}(\boldsymbol{x}_1, ..., \boldsymbol{x}_q) \approx &\frac{1}{2} \log \det\left(\Sigma_y(\boldsymbol{x}_1, ..., \boldsymbol{x}_q \mid \mathcal{D}_t)\right) \\
&- \frac{1}{L} \sum_{l=1}^{L} \frac{1}{2} \log \det\left(\Sigma_y\left(\boldsymbol{x}_1, ..., \boldsymbol{x}_q \mid \mathcal{D}_t \cup Q_{\boldsymbol{x}_0}^l\right)\right).
\end{aligned} \tag{30}
$$

The term $\Sigma_y(\boldsymbol{x}_1, ..., \boldsymbol{x}_q \mid \mathcal{D}_t)$ denotes the predictive covariance matrix of the posterior GP at the query locations $\boldsymbol{x}_1, ..., \boldsymbol{x}_q$ given data $\mathcal{D}_t$.

Optimizing (30) becomes more computationally expensive as $q$ increases. However, we expect this increase to be relatively minor. The most expensive parts of the LES formalisms, i.e., generating $L$

GP samples, locally optimizing them and then conditioning $L$ GPs on the descent sequences have to be done only once, independently of $q$.

# D EMPIRICAL EVALUATION

This section gives additional results and details on the empirical evaluation. Most notably, we show that the local exploration behavior results in reduced cumulative cost (D.3,D.4) and show extended results on GP-samples with varying problem complexity for the within and out of model comparison setting (D.3). Additional results are the impact of the discretizations on runtime and quality (D.5) and the impact of the local optimizer choice (D.7).

## D.1 LES ALGORITHM HYPERPARAMETER

Table 3 summarizes the main LES hyperparameter. In summary, LES has four types of hyper parameters: (a) GP-hyperparameter as any other BO algorithm (see Tab. 4, 10). (b) $L$ and $P$ that govern the accuracy of the acquisition function approximation. They should be chosen as large as computational resources permit (see Appx. D.5) (c) parameter of the stopping rule of the stopping rule (see Appx. E) (d) the local optimizer and its parameters (see Appx. D.7).

Table 3: Default hyperparameter of LES

| Name | Description | Value |
|------|-------------|-------|
| | Number of initial uniform random samples | 2 |
| $L$ | Monte-Carlo samples of gradient descent sequences | 250 |
| $P$ | discretizations of the gradient descent sequences | 8 |
| $M$ | prior features of the GP posterior sampling | 1024 |
| $\epsilon$ | stopping criterion optimality | $0.1, 0.01$ |
| $\delta$ | total risk | 0.05 |
| $\delta_{\text{est}}$ | estimation risk | 0.0025 |
| $T_{\text{dec}}$ | samples between each decision | 25 |
| | Local Optimizer | ADAM |
| | Number of Local Optimization Steps | 500 |
| | Learning Rate | 0.002 |
| | Momentum Parameters | Keras Default |

## D.2 BENCHMARK ALGORITHM HYPERPARAMETER

For HCI-GIBO we choose $\alpha = 0.9$ and perform hyperparameter optimization after each gradient step. We use the BoTorch (Balandat et al., 2020) implementations of MES, logEI, and TuRBO. MES uses a candidate set of 5000 points and both MES and TuRBO use default parameters of the BoTorch implementation. Again, all algorithm use identical GP parameters with the exception of logEI-DSP, where the hyperprior of (Hvarfner et al., 2024) is chosen.

## D.3 ADDITIONAL DETAILS AND RESULTS ON GP SAMPLES

**General Setup.** Table 4 summarizes the GP hyperparameters used in the GP-sample experiments. Following (Hvarfner et al., 2024), we employ a log-normal hyperprior $p(l)$ for the length scales and assume a constant, known measurement noise distribution. Test functions are generated by first sampling length scales from the hyperprior and then drawing functions according to (4). To vary problem difficulty, we scale the log-normal hyperpriors of (Hvarfner et al., 2024) (see Sec. 6.2); expected length scales are reported in Table 5. Note that the original hyperprior assumes very large average length scales, whereas in the high-complexity scenario the expected length scale at $d = 50$ is $\mathrm{E}[p(l)] = 0.25$, which is still reasonable.

Across all experiments, data is not standardized, each algorithm is evaluated on 20 random seeds, and the two initial points are chosen randomly.

Table 4: GP hyperparameters for out of model comparison on GP samples

| Name | Description | Value |
|---|---|---|
| $k(\cdot, \cdot)$ | kernel | SE-ARD |
| $p(l; d)$ | length-scale hyper prior | Log Normal (see Tab. 5) |
| $\sigma_{\mathrm{n}}$ | observation noise | fixed at 0.002 |
| $\sigma_{\mathrm{k}}$ | GP output scale | variable - init at 1 |
| $l_{\mathrm{init}}$ | length scale initialization | $\mathrm{E}[\mathrm{p}(l; d)]$ |
| | hyperparameter optimization frequency | after every sample[3] |
| | standardize data | yes |

Table 5: Expected length scales $\mathrm{E}[\mathrm{p}(l; d)]$ for the different hyperpriors.

| Complexity | $d = 5$ | $d = 10$ | $d = 20$ | $d = 30$ | $d = 50$ |
|---|---|---|---|---|---|
| **high**: $\mathrm{p}(l; d) = \mathrm{logn}(-2.5\sqrt{2} + \log\sqrt{d}, \sqrt{3}/5)$ | 0.08 | 0.11 | 0.15 | 0.19 | 0.25 |
| **medium**: $\mathrm{p}(l; d) = \mathrm{logn}(-2.0\sqrt{2} + \log\sqrt{d}, \sqrt{3}/4)$ | 0.16 | 0.23 | 0.33 | 0.4 | 0.52 |
| **low**: $\mathrm{p}; \mathrm{d}(l) = \mathrm{logn}(-1.0\sqrt{2} + \log\sqrt{d}, \sqrt{3}/2)$ | 0.83 | 1.19 | 1.67 | 2.05 | 2.65 |
| **extremely low** (Hvarfner et al., 2024): $\mathrm{p}(l; d) = \mathrm{logn}(1.0\sqrt{2} + \log\sqrt{d}, \sqrt{3})$ | 21.86 | 30.92 | 34.73 | 53.56 | 69.15 |

**Within-Model Comparison.** In this setting, all BO algorithms are given access to the sampled ground-truth hyperparameters (length scales, output scale, and noise variance). Tables 6 and 7 summarize the results, with statistical significance determined by the signed rank test. Entries not in bold are statistically significantly worse than the best-performing algorithm. Figures 6–13 show the convergence curves.

LES achieves statistically significant improvements in higher-dimensional, high-complexity settings, particularly in terms of cumulative regret. Global BO methods only outperform LES in high-complexity, low-dimensional cases.

---

[2]Except for the GIBO variants, where we optimize the hyperparameter only after each step.

Table 6: Best achieved function value after full budget for within model comparison on GP samples. Entries not in bold are statistically significantly worse than the best preforming algorithm. Smaller is better.

| Complexity | Method | $d=5$ | $d=10$ | $d=20$ | $d=30$ | $d=50$ |
|---|---|---|---|---|---|---|
| **high** | LES (ours) | $-2.8$ | $\mathbf{-4.8}$ | $\mathbf{-7.4}$ | $\mathbf{-9.0}$ | $\mathbf{-10.8}$ |
| | MES | $-2.4$ | $-2.6$ | $-3.0$ | $-2.9$ | $-2.8$ |
| | logEI | $\mathbf{-3.9}$ | $-4.4$ | $-2.9$ | $-3.2$ | $-2.9$ |
| | TuRBO | $-3.5$ | $\mathbf{-4.8}$ | $\mathbf{-7.2}$ | $-8.0$ | $-7.1$ |
| | HCI-GIBO | $-2.7$ | $-2.0$ | $-2.3$ | $-1.8$ | $-0.7$ |
| | Sobol random | $-2.4$ | $-2.7$ | $-2.8$ | $-3.2$ | $-3.0$ |
| **medium** | LES (ours) | $-2.9$ | $-4.4$ | $\mathbf{-7.3}$ | $-8.6$ | $\mathbf{-10.4}$ |
| | MES | $\mathbf{-3.4}$ | $-2.0$ | $-2.8$ | $-2.9$ | $-2.9$ |
| | logEI | $\mathbf{-3.6}$ | $\mathbf{-5.3}$ | $\mathbf{-7.1}$ | $\mathbf{-8.4}$ | $-9.6$ |
| | TuRBO | $-3.1$ | $-4.8$ | $\mathbf{-7.0}$ | $\mathbf{-8.5}$ | $-7.9$ |
| | HCI-GIBO | $-2.9$ | $-3.6$ | $-2.9$ | $-2.0$ | $-1.7$ |
| | Sobol random | $-2.5$ | $-2.7$ | $-2.8$ | $-2.9$ | $-2.7$ |
| **low** | LES (ours) | $-2.1$ | $-3.7$ | $-5.5$ | $\mathbf{-6.8}$ | $\mathbf{-8.8}$ |
| | MES | $\mathbf{-2.9}$ | $-4.0$ | $-4.9$ | $-4.6$ | $-3.9$ |
| | logEI | $\mathbf{-2.9}$ | $\mathbf{-4.1}$ | $\mathbf{-5.8}$ | $\mathbf{-6.6}$ | $\mathbf{-8.5}$ |
| | TuRBO | $-2.4$ | $-3.6$ | $-5.5$ | $\mathbf{-6.5}$ | $-8.1$ |
| | HCI-GIBO | $-2.1$ | $-3.4$ | $-5.0$ | $-5.9$ | $-7.3$ |
| | Sobol random | $-2.0$ | $-2.3$ | $-2.9$ | $-2.8$ | $-3.0$ |
| **extremely low** | LES (ours) | $\mathbf{-0.6}$ | $-0.8$ | $\mathbf{-1.2}$ | $\mathbf{-1.5}$ | $\mathbf{-3.0}$ |
| | MES | $\mathbf{-0.6}$ | $-0.9$ | $-1.3$ | $-1.5$ | $-2.7$ |
| | logEI | $\mathbf{-0.6}$ | $\mathbf{-0.9}$ | $\mathbf{-1.3}$ | $\mathbf{-1.5}$ | $\mathbf{-3.0}$ |
| | TuRBO | $-0.6$ | $-0.8$ | $-1.2$ | $-1.5$ | $-2.8$ |
| | HCI-GIBO | $\mathbf{-0.6}$ | $\mathbf{-0.8}$ | $-1.1$ | $-1.3$ | $-2.7$ |
| | Sobol random | $-0.5$ | $-0.7$ | $-1.0$ | $-1.0$ | $-1.5$ |

Table 7: Cumulative observed function values after full budget for within model comparison on GP samples. Entries not in bold are statistically significantly worse than the best preforming algorithm. Smaller is better.

| Complexity | Method | $d=5$ | $d=10$ | $d=20$ | $d=30$ | $d=50$ |
|---|---|---|---|---|---|---|
| **high** | LES (ours) | $\mathbf{-260.1}$ | $\mathbf{-842.1}$ | $\mathbf{-2515.1}$ | $\mathbf{-2938.4}$ | $\mathbf{-3214.3}$ |
| | MES | $35.8$ | $22.3$ | $0.1$ | $1.1$ | $4.0$ |
| | logEI | $-176.6$ | $-230.6$ | $-17.1$ | $-0.8$ | $6.5$ |
| | TuRBO | $\mathbf{-247.9}$ | $-672.5$ | $-1783.3$ | $-1503.0$ | $-977.9$ |
| | HCI-GIBO | $-59.5$ | $-28.3$ | $-16.6$ | $-19.8$ | $-24.7$ |
| | Sobol random | $1.5$ | $-1.7$ | $-7.2$ | $7.2$ | $3.7$ |
| **medium** | LES (ours) | $\mathbf{-272.8}$ | $\mathbf{-811.4}$ | $\mathbf{-2464.3}$ | $\mathbf{-2861.7}$ | $\mathbf{-3159.1}$ |
| | MES | $-97.1$ | $167.3$ | $288.4$ | $6.5$ | $-0.6$ |
| | logEI | $-163.7$ | $-645.5$ | $-2111.3$ | $-2071.3$ | $-1378.3$ |
| | TuRBO | $\mathbf{-246.7}$ | $-697.6$ | $-1876.0$ | $-1924.9$ | $-1375.1$ |
| | HCI-GIBO | $-114.5$ | $-121.3$ | $-51.4$ | $-27.4$ | $-4.2$ |
| | Sobol random | $6.4$ | $3.0$ | $0.6$ | $5.3$ | $1.3$ |
| **low** | LES (ours) | $\mathbf{-192.9}$ | $\mathbf{-685.7}$ | $\mathbf{-2059.5}$ | $\mathbf{-2415.1}$ | $\mathbf{-2969.3}$ |
| | MES | $-97.6$ | $-358.9$ | $-682.8$ | $-276.7$ | $337.6$ |
| | logEI | $-88.1$ | $-434.0$ | $-1679.0$ | $-2144.8$ | $-2795.6$ |
| | TuRBO | $\mathbf{-201.8}$ | $-637.6$ | $-1916.6$ | $-2048.7$ | $-2390.1$ |
| | HCI-GIBO | $-169.8$ | $-581.9$ | $-1638.9$ | $-1874.3$ | $-2224.6$ |
| | Sobol random | $7.1$ | $1.7$ | $-42.8$ | $-0.2$ | $-1.5$ |
| **extremely low** | LES (ours) | $\mathbf{-56.8}$ | $\mathbf{-160.5}$ | $\mathbf{-471.9}$ | $\mathbf{-591.0}$ | $\mathbf{-1142.4}$ |
| | MES | $-13.0$ | $-7.5$ | $-132.2$ | $-69.1$ | $-252.3$ |
| | logEI | $-51.0$ | $-122.5$ | $-357.4$ | $-320.6$ | $-581.0$ |
| | TuRBO | $-57.1$ | $-152.3$ | $-456.3$ | $-516.8$ | $-896.0$ |
| | HCI-GIBO | $-54.3$ | $-145.2$ | $-420.7$ | $-496.1$ | $-913.7$ |
| | Sobol random | $-1.9$ | $21.1$ | $-74.3$ | $61.9$ | $-88.9$ |

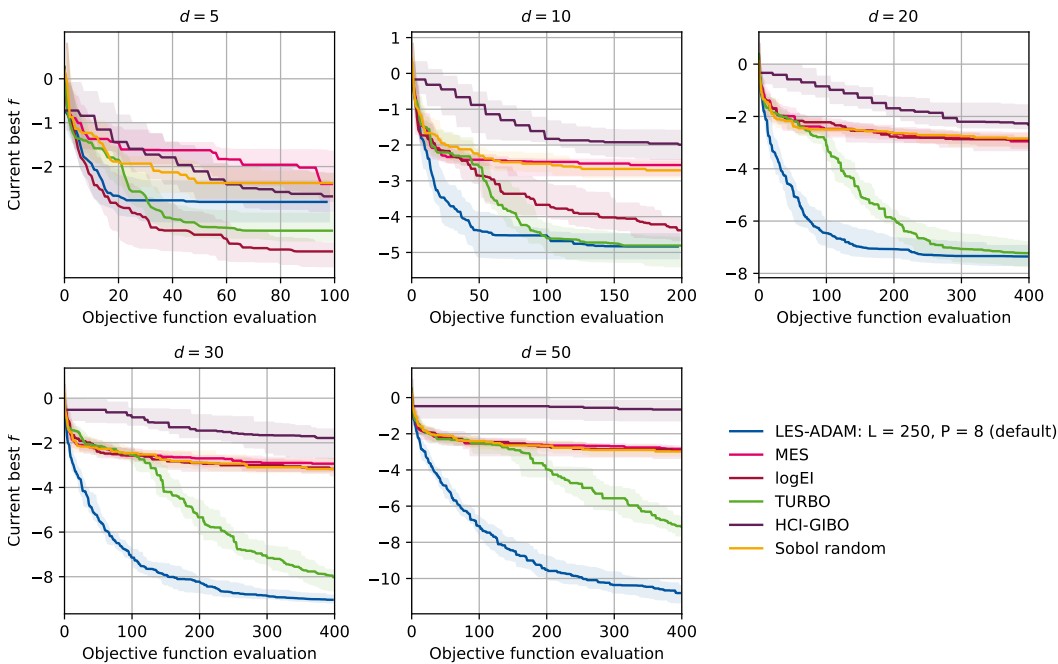

Figure 6: **Within Model Comparison, Complexity - high:** Median, 25-, and 75-percent quantiles - detailed results

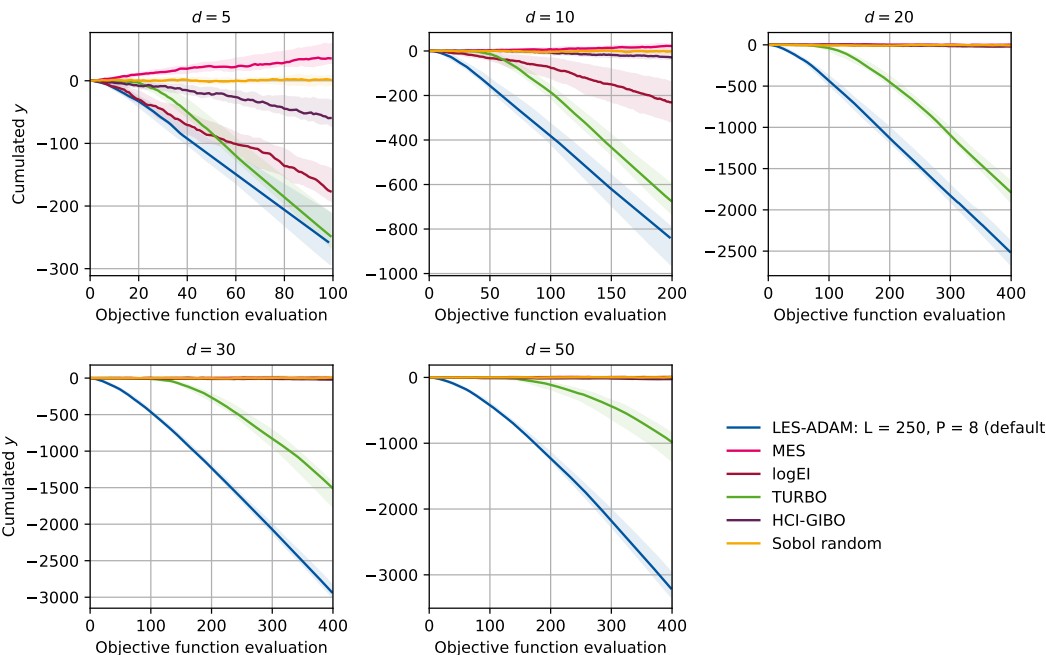

Figure 7: **Within Model Comparison, Complexity - high:** Median, 25-, and 75-percent quantiles - detailed results

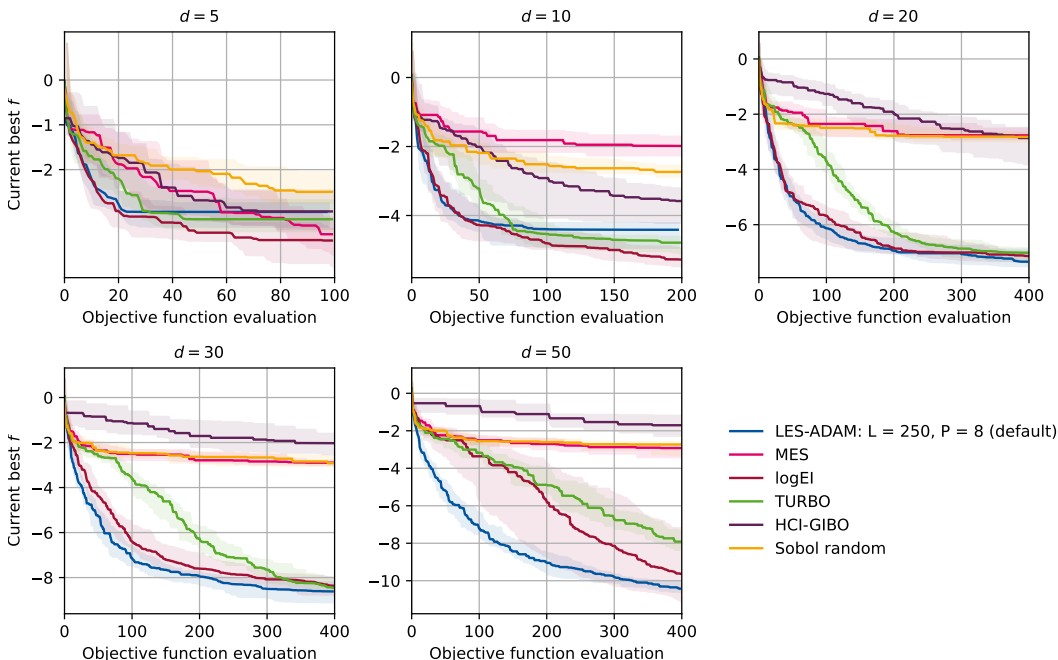

Figure 8: **Within Model Comparison, Complexity - medium:** Median, 25-, and 75-percent quantiles - detailed results

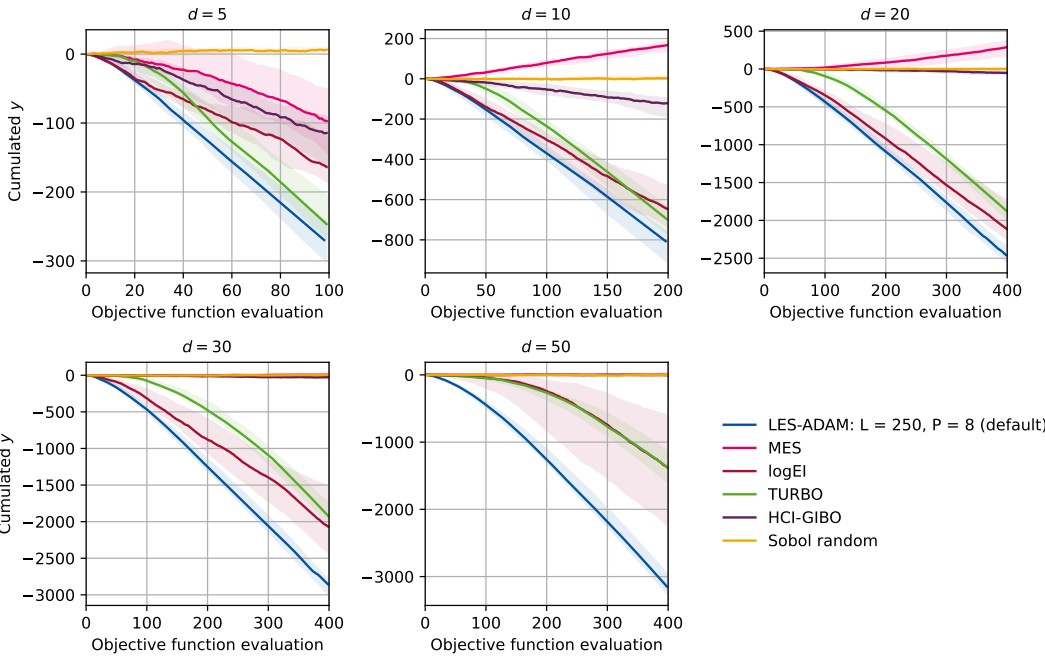

Figure 9: **Within Model Comparison, Complexity - medium:** Median, 25-, and 75-percent quantiles - detailed results

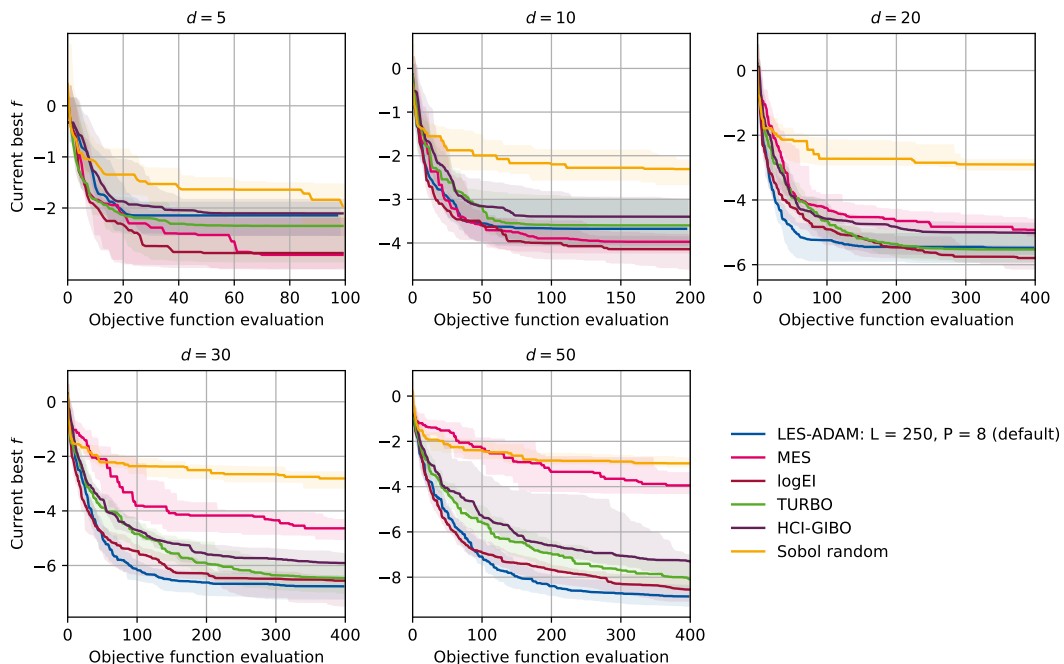

Figure 10: **Within Model Comparison, Complexity - low:** Median, 25-, and 75-percent quantiles - detailed results

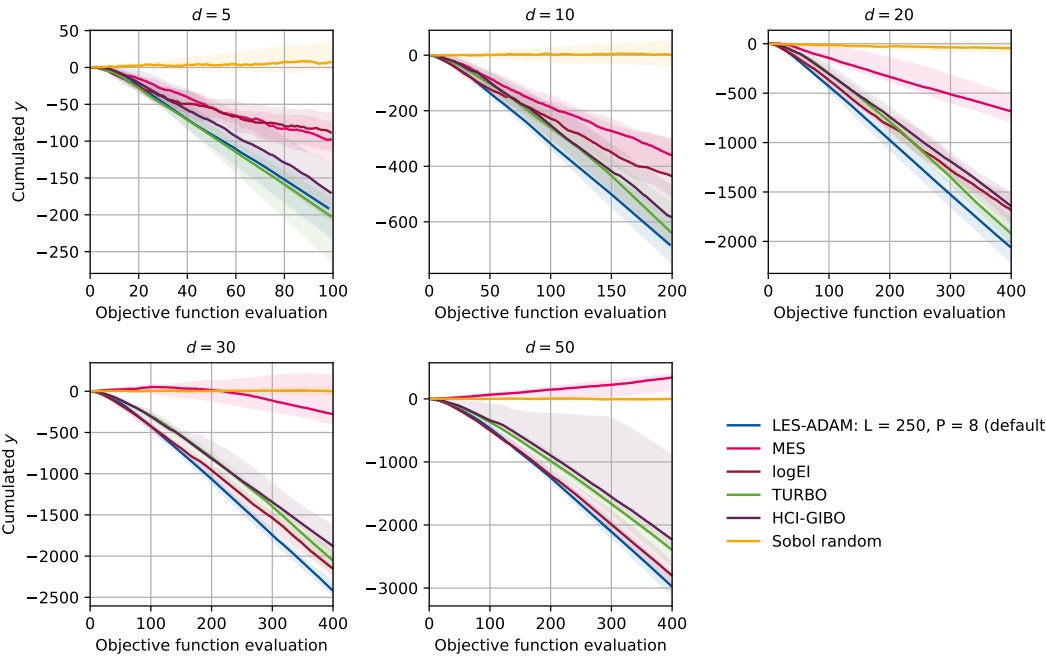

Figure 11: **Within Model Comparison, Complexity - low:** Median, 25-, and 75-percent quantiles - detailed results

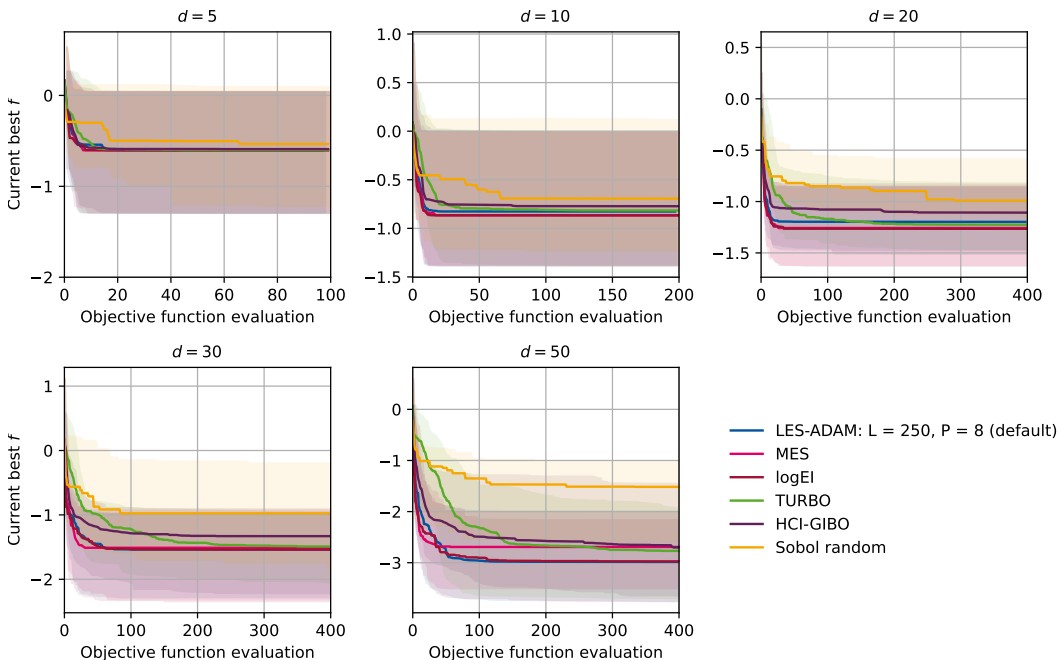

Figure 12: **Within Model Comparison, Complexity - extremely low:** Median, 25-, and 75-percent quantiles - detailed results

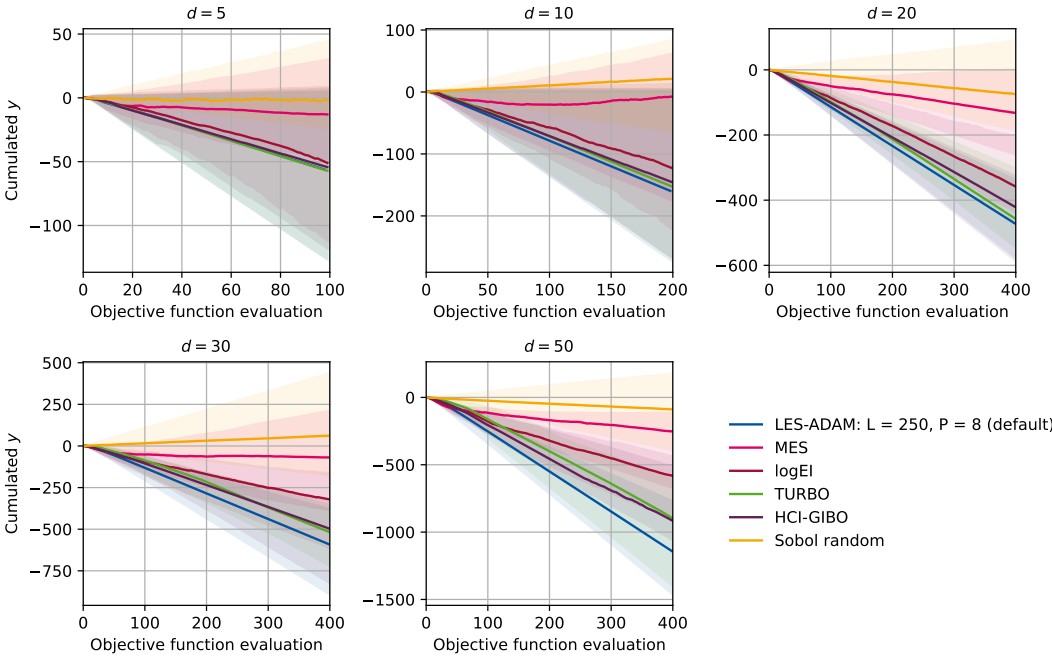

Figure 13: **Within Model Comparison, Complexity - extremely low:** Median, 25-, and 75-percent quantiles - detailed results

**Out of Model Comparison.** The out-of-model setup is identical to the within-model one except that GP hyperparameters are now estimated via MAP according to Table 4. Tables 8 and 9 report best and cumulative evaluations, and Figures 14–21 show the corresponding progress in optimization over the number of queries.

Table 8: Best achieved function value after full budget for out-of-model comparison on GP samples. Entries not in bold are statistically significantly worse than the best preforming algorithm. Smaller is better.

| Complexity | Method | $d = 5$ | $d = 10$ | $d = 20$ | $d = 30$ | $d = 50$ |
|---|---|---|---|---|---|---|
| **high** | LES (ours) | $-2.9$ | $\mathbf{-5.0}$ | $\mathbf{-7.2}$ | $\mathbf{-8.5}$ | $\mathbf{-7.8}$ |
| | MES | $-1.9$ | $-2.5$ | $-2.8$ | $-2.8$ | $-3.0$ |
| | logEI | $\mathbf{-4.0}$ | $-4.3$ | $-2.8$ | $-2.9$ | $-3.0$ |
| | logEI-DSP | $\mathbf{-3.7}$ | $-4.1$ | $-4.0$ | $-4.0$ | $-4.1$ |
| | TuRBO | $-3.7$ | $\mathbf{-5.0}$ | $\mathbf{-7.1}$ | $\mathbf{-8.2}$ | $-7.1$ |
| | HCI-GIBO | $-2.3$ | $-2.2$ | $-1.9$ | $-2.0$ | $-1.8$ |
| | Sobol random | $-2.4$ | $-2.7$ | $-2.8$ | $-3.2$ | $-3.0$ |
| **medium** | LES (ours) | $-3.0$ | $-4.6$ | $\mathbf{-7.1}$ | $\mathbf{-8.6}$ | $\mathbf{-8.8}$ |
| | MES | $\mathbf{-3.6}$ | $-2.1$ | $-2.9$ | $-2.8$ | $-2.9$ |
| | logEI | $\mathbf{-3.6}$ | $\mathbf{-5.1}$ | $\mathbf{-7.2}$ | $-8.1$ | $-4.5$ |
| | logEI-DSP | $\mathbf{-3.6}$ | $\mathbf{-5.0}$ | $\mathbf{-7.0}$ | $-7.9$ | $-7.0$ |
| | TuRBO | $-3.1$ | $-4.9$ | $\mathbf{-7.0}$ | $-7.9$ | $-8.0$ |
| | HCI-GIBO | $-2.9$ | $-3.7$ | $-2.9$ | $-2.0$ | $-1.6$ |
| | Sobol random | $-2.5$ | $-2.7$ | $-2.8$ | $-2.9$ | $-2.7$ |
| **low** | LES (ours) | $-2.1$ | $-3.7$ | $-5.2$ | $\mathbf{-6.6}$ | $\mathbf{-8.5}$ |
| | MES | $\mathbf{-2.9}$ | $\mathbf{-4.0}$ | $-5.1$ | $-5.4$ | $-3.7$ |
| | logEI | $\mathbf{-2.9}$ | $\mathbf{-4.1}$ | $\mathbf{-5.7}$ | $\mathbf{-6.4}$ | $\mathbf{-8.4}$ |
| | logEI-DSP | $\mathbf{-2.9}$ | $\mathbf{-4.1}$ | $\mathbf{-5.7}$ | $\mathbf{-6.6}$ | $-8.1$ |
| | TuRBO | $-2.4$ | $-3.7$ | $-5.4$ | $\mathbf{-6.4}$ | $-7.7$ |
| | HCI-GIBO | $-2.2$ | $-3.5$ | $-4.9$ | $-5.9$ | $-7.5$ |
| | Sobol random | $-2.0$ | $-2.3$ | $-2.9$ | $-2.8$ | $-3.0$ |
| **extremely low** | LES (ours) | $-0.6$ | $-0.8$ | $-1.2$ | $-1.5$ | $\mathbf{-2.9}$ |
| | MES | $-0.6$ | $-0.9$ | $-1.2$ | $-1.4$ | $-2.5$ |
| | logEI | $\mathbf{-0.6}$ | $\mathbf{-0.9}$ | $\mathbf{-1.3}$ | $\mathbf{-1.5}$ | $\mathbf{-3.0}$ |
| | logEI-DSP | $\mathbf{-0.6}$ | $\mathbf{-0.9}$ | $\mathbf{-1.3}$ | $-1.5$ | $\mathbf{-3.0}$ |
| | TuRBO | $-0.6$ | $-0.8$ | $-1.2$ | $-1.5$ | $-2.9$ |
| | HCI-GIBO | $-0.6$ | $-0.7$ | $\mathbf{-1.3}$ | $-1.5$ | $-2.5$ |
| | Sobol random | $-0.5$ | $-0.7$ | $-1.0$ | $-1.0$ | $-1.5$ |

Table 9: Cumulative observed function values after full budget for out of model comparison on GP samples. Entries not in bold are statistically significantly worse than the best preforming algorithm. Smaller is better.

| Complexity | Method | $d = 5$ | $d = 10$ | $d = 20$ | $d = 30$ | $d = 50$ |
|---|---|---|---|---|---|---|
| **high** | LES (ours) | **−259.0** | **−867.2** | **−2298.3** | **−2482.5** | **−2081.9** |
| | MES | 56.9 | 13.6 | 11.6 | −0.2 | 10.7 |
| | logEI | −172.9 | −197.7 | 0.4 | 3.4 | 2.8 |
| | logEI-DSP | −130.4 | −169.4 | −128.1 | −127.2 | −119.3 |
| | TuRBO | **−257.0** | −649.4 | −1789.6 | −1565.5 | −927.9 |
| | HCI-GIBO | −50.6 | −24.5 | −27.7 | −18.1 | −8.9 |
| | Sobol random | 1.5 | −1.7 | −7.2 | 7.2 | 3.7 |
| **medium** | LES (ours) | **−269.6** | **−819.0** | **−2267.5** | **−2494.5** | **−2427.1** |
| | MES | −111.1 | 176.1 | 34.3 | −3.5 | −5.4 |
| | logEI | −172.8 | −667.3 | −1949.7 | −1597.4 | −110.1 |
| | logEI-DSP | −150.1 | −524.4 | −1912.4 | −1644.4 | −652.1 |
| | TuRBO | **−250.9** | −673.5 | −1900.0 | −1737.7 | −1352.6 |
| | HCI-GIBO | −160.1 | −109.0 | −40.8 | −15.1 | −19.9 |
| | Sobol random | 6.4 | 3.0 | 0.6 | 5.3 | 1.3 |
| **low** | LES (ours) | **−182.3** | **−662.1** | **−1883.7** | **−2183.6** | **−2572.6** |
| | MES | −102.5 | −373.0 | −666.8 | −443.0 | 317.9 |
| | logEI | −101.7 | −448.3 | −1639.7 | **−2049.2** | **−2691.8** |
| | logEI-DSP | −88.9 | −380.8 | −1563.4 | −1966.1 | **−2604.0** |
| | TuRBO | **−204.1** | −620.1 | −1764.8 | −1848.2 | −2151.1 |
| | HCI-GIBO | −148.5 | −456.3 | −1454.9 | −1783.2 | −2128.8 |
| | Sobol random | 7.1 | 1.7 | −42.7 | −0.1 | −1.5 |
| **extremely low** | LES (ours) | **−54.2** | **−146.5** | **−475.7** | **−580.9** | **−1063.6** |
| | MES | −22.8 | −10.7 | −147.1 | −84.6 | −243.6 |
| | logEI | −43.6 | −102.3 | −338.4 | −318.6 | −629.5 |
| | logEI-DSP | −40.7 | −103.1 | −335.6 | −310.1 | −623.9 |
| | TuRBO | **−54.7** | **−155.0** | −466.0 | −554.3 | −1053.0 |
| | HCI-GIBO | −40.9 | −88.7 | −318.1 | −296.4 | −371.1 |
| | Sobol random | −1.9 | 21.1 | −74.2 | 61.9 | −88.8 |

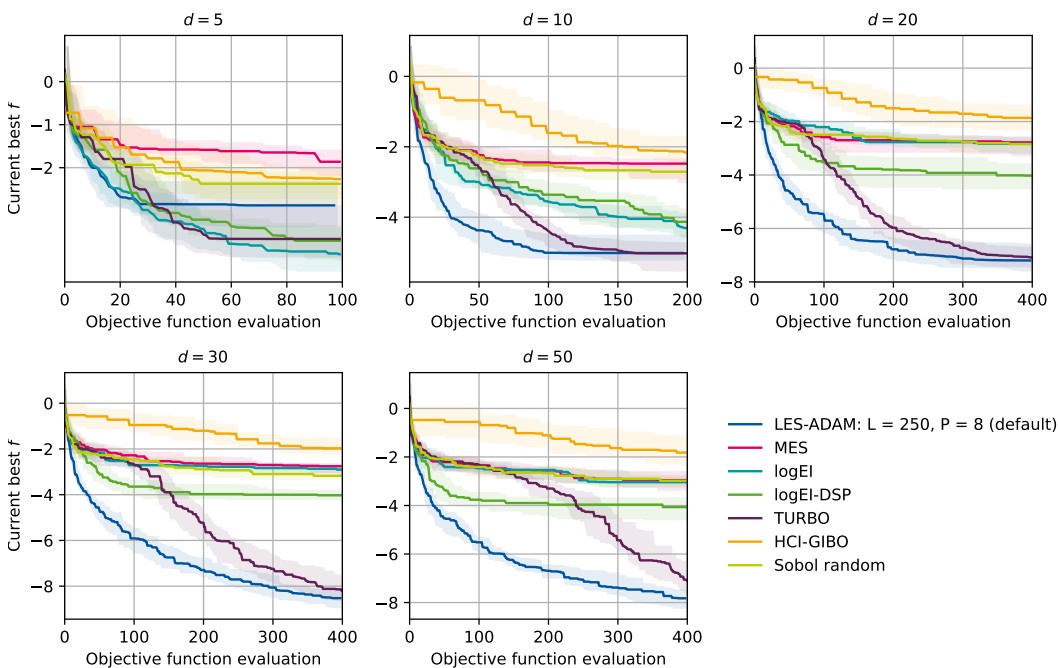

Figure 14: **Out of model comparison, complexity - high:** Median, 25-, and 75-percent quantiles - detailed results

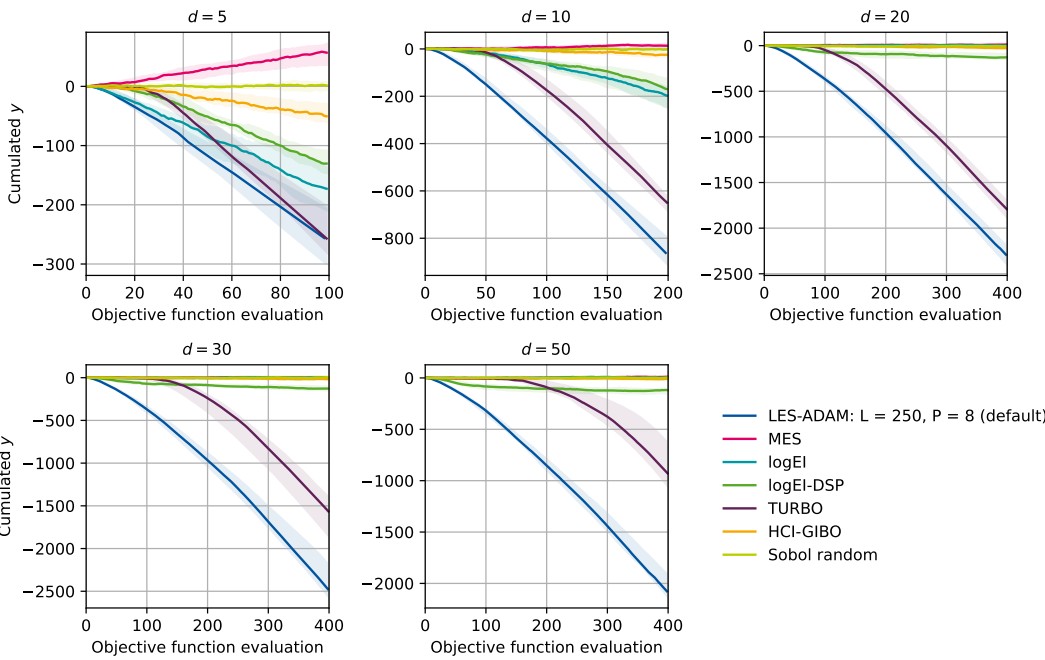

Figure 15: **Out of model comparison, complexity - high:** Median, 25-, and 75-percent quantiles - detailed results

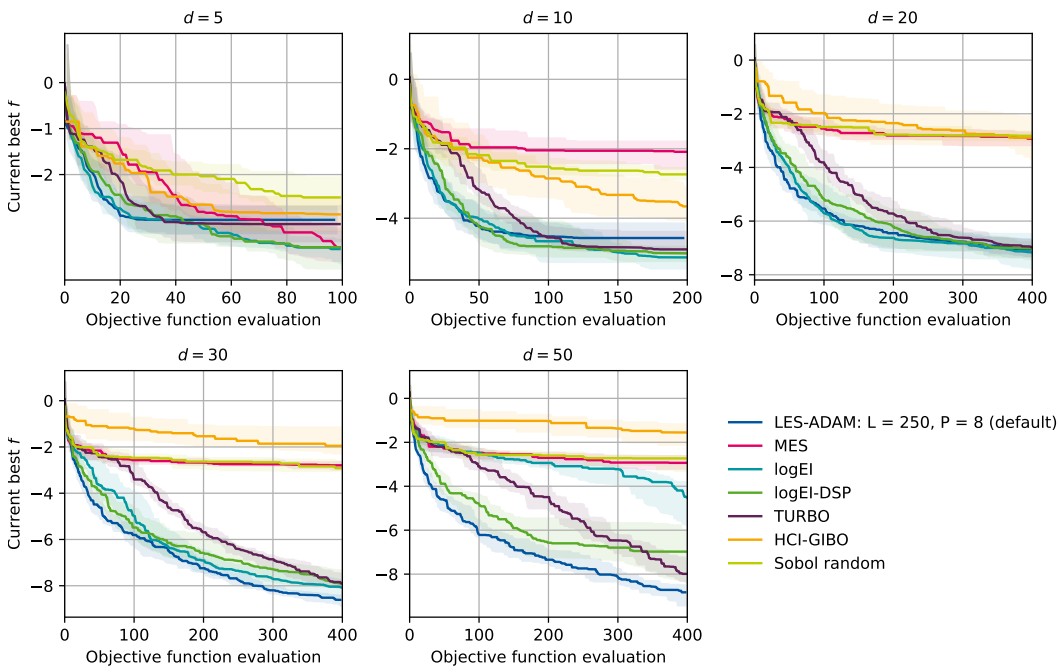

Figure 16: **Out of model comparison, complexity - medium:** Median, 25-, and 75-percent quantiles - detailed results

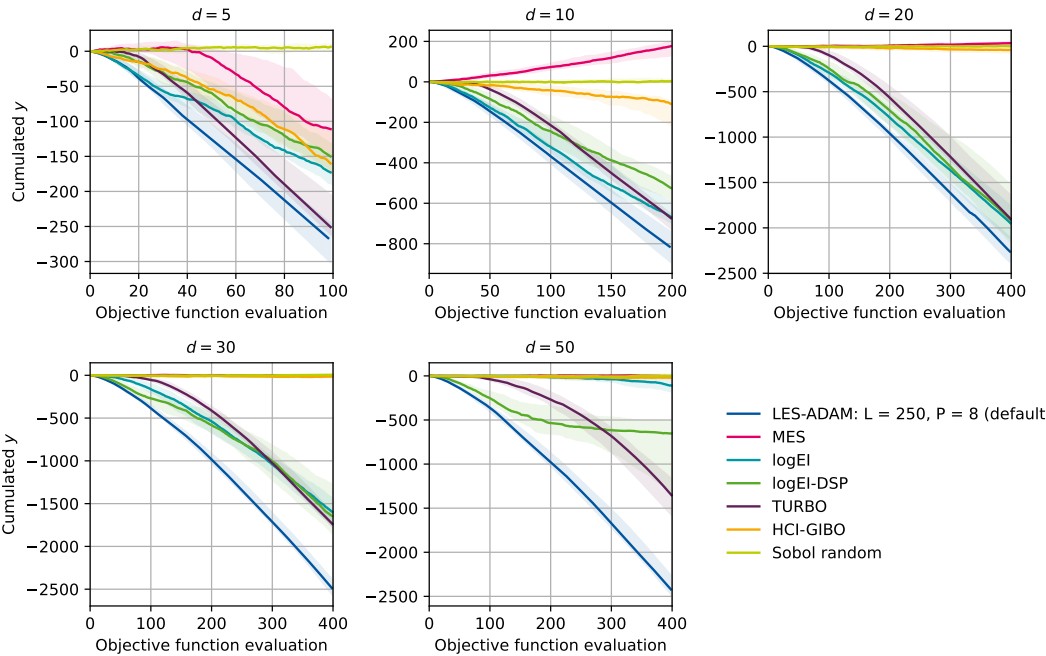

Figure 17: **Out of model comparison, complexity - medium:** Median, 25-, and 75-percent quantiles - detailed results

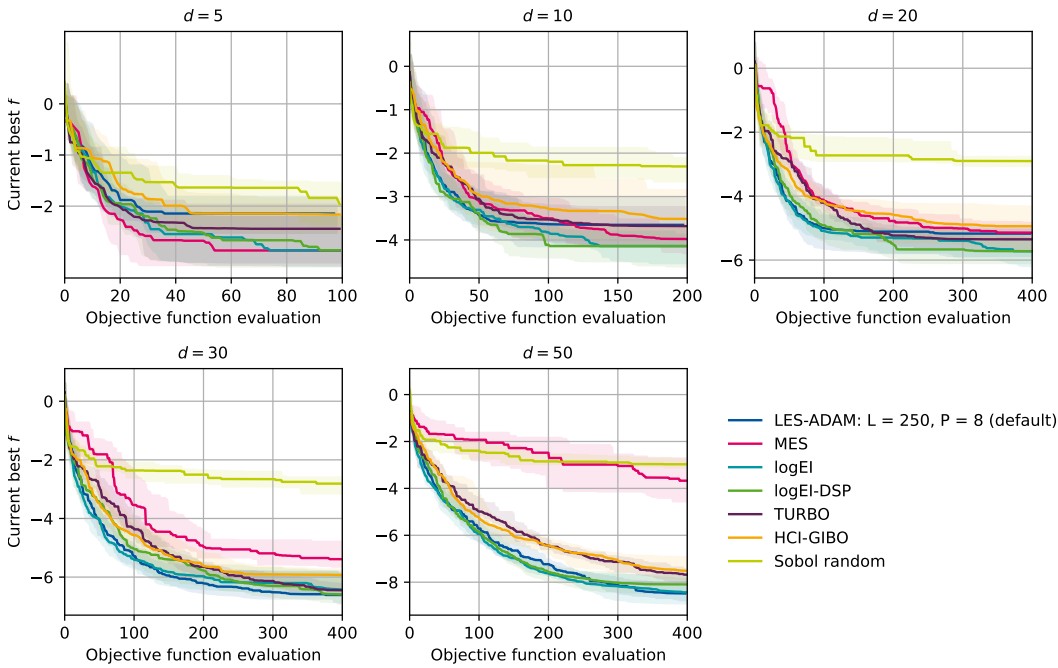

Figure 18: **Out of model comparison, complexity - low:** Median, 25-, and 75-percent quantiles - detailed results

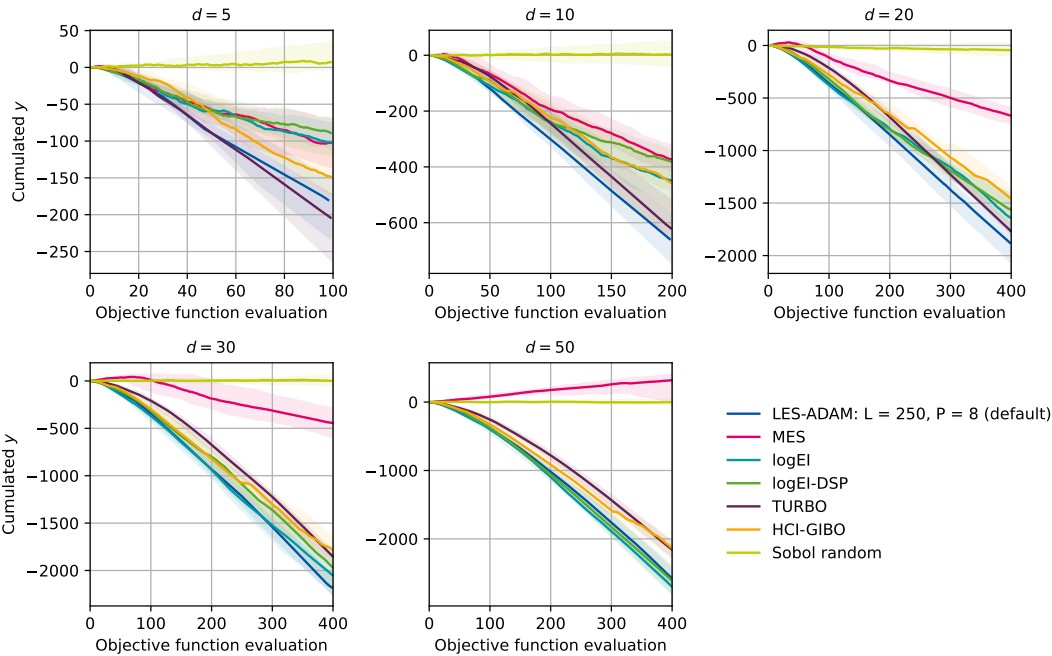

Figure 19: **Out of model comparison, complexity - low:** Median, 25-, and 75-percent quantiles - detailed results

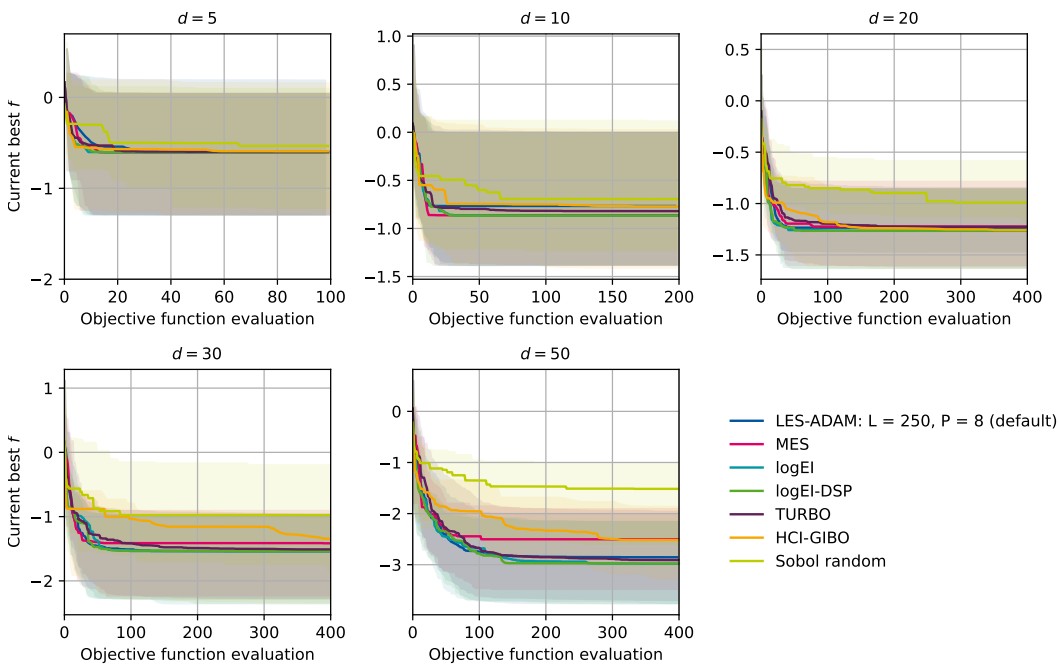

Figure 20: **Out of model comparison, complexity - extremely low:** Median, 25-, and 75-percent quantiles - detailed results

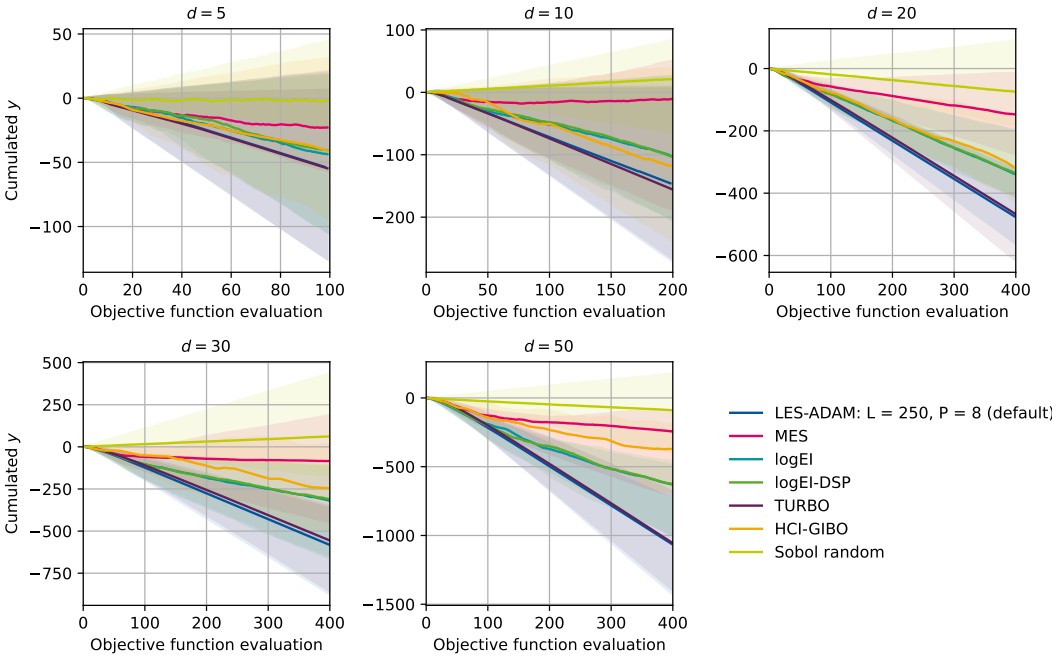

Figure 21: **Out of model comparison, complexity - extremely low:** Median, 25-, and 75-percent quantiles - detailed results

### D.4 ADDITIONAL DETAILS AND RESULTS ON SYNTHETIC AND APPLICATION-ORIENTED OBJECTIVE FUNCTIONS

In total, we evaluate LES on nine benchmark functions (Fig.23 and 24). On functions with a single local optimum (square function $f(x) = \boldsymbol{x}\boldsymbol{x}^\top$), all methods reliably identify the optimum, though LES and TuRBO achieve lower cumulative regret, underscoring the advantage of local search in this setting. In contrast, the 5-d Ackley function – designed as a failure case for LES – leads all methods, including LES and the global baselines, to perform poorly (Appx. D.4). Surprisingly, for 30-d Ackley, LES and TuRBO outperform global methods. However, LES has a high run-to-run variance which indicates that some runs get stuck in local optima.

In the rover (Wang et al., 2018) and Mopta08 (Jones, 2008) tasks, LES, logEI and TuRBO perform similarly with LES being best in the rover task and logEI being best in the Mopta08 task. In the lunar lander task (Brockman et al., 2016), LES is not competitive. The lunar lander task has multiple local optima where LES is getting stuck in some runs (see Fig. 22).

The lunar lander problem (Fig. 22) and the Ackley function both contain many local minima, which makes them particularly challenging for our local method. Interestingly, LES performs still best on the 30-dimensional Ackley function. Overall, LES achieves the lowest cumulative regret – sometimes tied with other algorithms – except on low-dimensional problems with many local optima (Ackley-$d = 5$, Lunar). For logEI, high exploration costs occur only in low dimensions, which may be explained by its tendency to repeatedly sample the same location once it has found a (local) optimum. In terms of simple regret, LES matches the baselines except on the low-dimensional, multi-modal benchmarks (5-d Ackley and Lunar).

All BO algorithms use the hyperparameter as presented in Table 10. Hyperparameters follow (Xu et al., 2025), using a box hyperprior and length scale initialization scaled by $\sqrt{d}$ to favor low model complexity. Observations are generated without noise. We use 20 seeds for the policy search tasks and 10 seeds for the synthetic functions.

Table 10: Model Hyperparameters for the synthetic and application-oriented objective functions

| Name | Description | Value |
|------|-------------|-------|
| $k(\cdot, \cdot)$ | kernel | SE-ARD |
| $p(l)$ | length-scale hyper prior | None |
| $\sigma_\mathrm{n}$ | observation noise | fixed at 0.001 |
| $\sigma_\mathrm{k}$ | GP output scale | variable |
| $l_{\max}$ | length scale upper bound | $\sqrt{d}$ |
| $l_{\min}$ | length scale lower bound | 0.05 |
| $l_{\mathrm{init}}$ | length scale initialization | $0.2\sqrt{d}$ |
| | hyperparameter optimization frequency | after every sample[4] |
| | standardize data | yes |

---

[4]Except for the GIBO variants, where we optimize the hyperparameter only after each step.

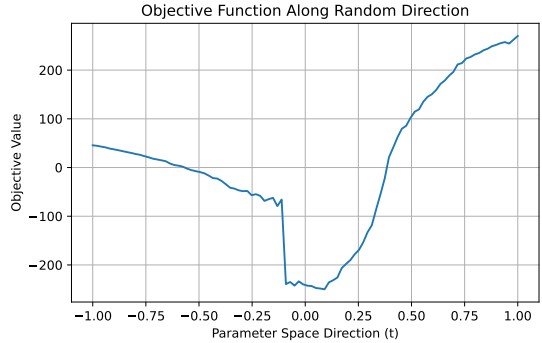

Figure 22: A random slice through the deterministic lunar lander objective function. Although the objective is deterministic, we see multiple noise-like local optima and a prominent step in the objective function landscape. Both properties are hard to model with a GP using an SE kernel with small observation noise.

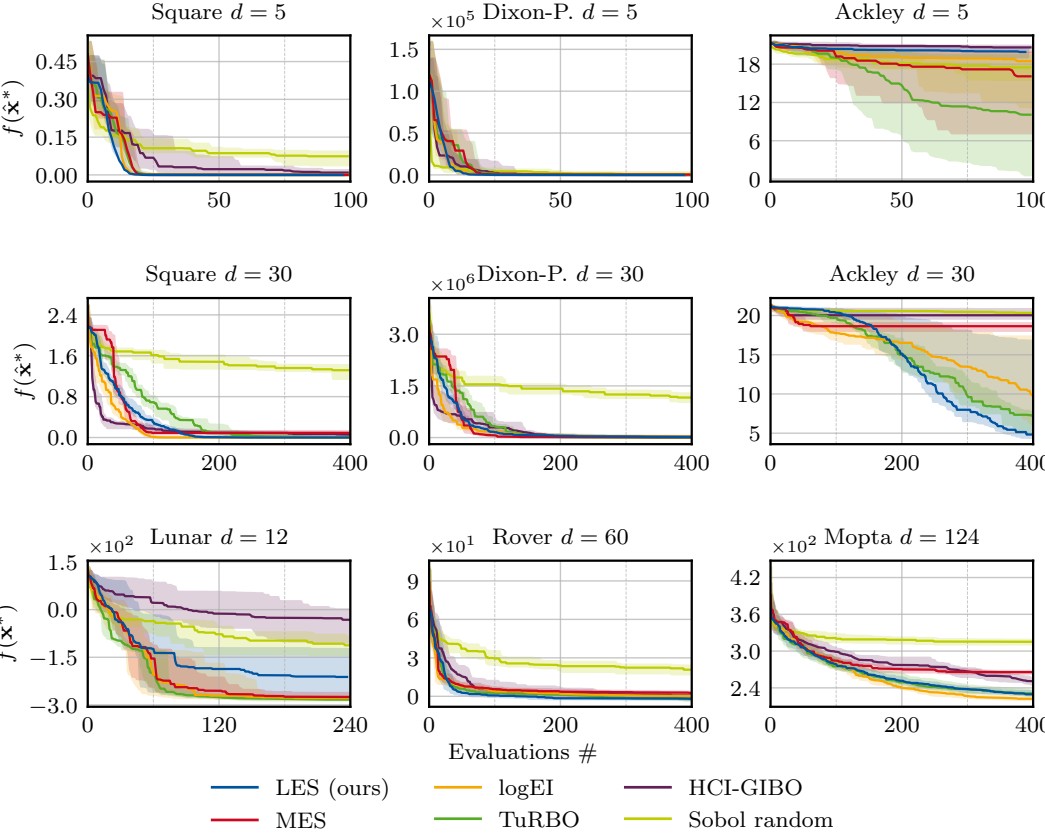

Figure 23: Median, 25-, and 75-percent quantiles - synthetic and application-oriented objective functions

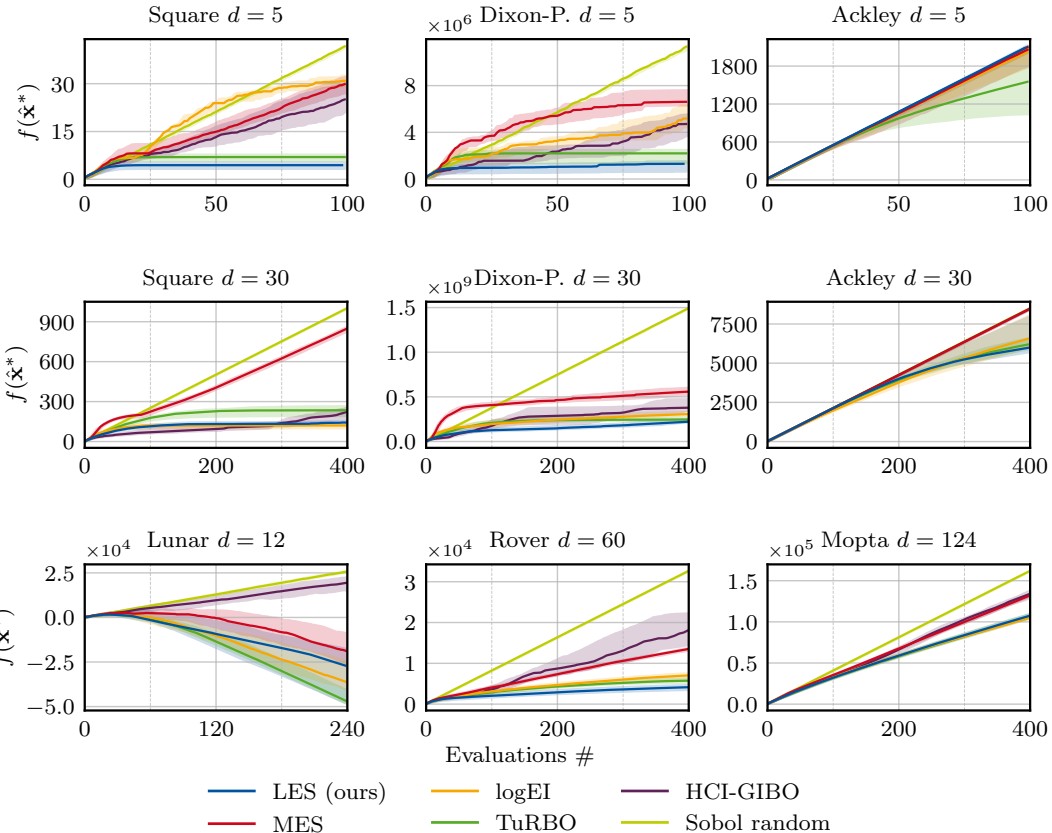

Figure 24: Median, 25-, and 75-percent quantiles of cumulative cost - synthetic and application-oriented objective functions

### D.5 ABLATIONS ON APPROXIMATION ACCURACIES AND RUNTIME

Figure 25 illustrates the effect of the number of Monte Carlo samples $L$ and the number of equally spaced points $P$ taken from each descent sequence. We evaluate this in the out-of-model GP sample scenario with medium complexity (see Sec. 6.2). As expected, more accurate approximations yield better performance, with the differences most pronounced in the $d = 50$ case. While $L = 250$ and $P = 16$ performs best, we adopt $L = 250$ and $P = 8$ in our experiments as a compromise between runtime and accuracy.

We further verify that conditioning on function values instead of gradients in the descent sequence does not substantially harm performance. For runtime and memory reasons, gradient conditioning (LES-ADAM-Grad.-Cond.) was only run with coarse discretizations, up to 300 samples and excluding $d = 50$.

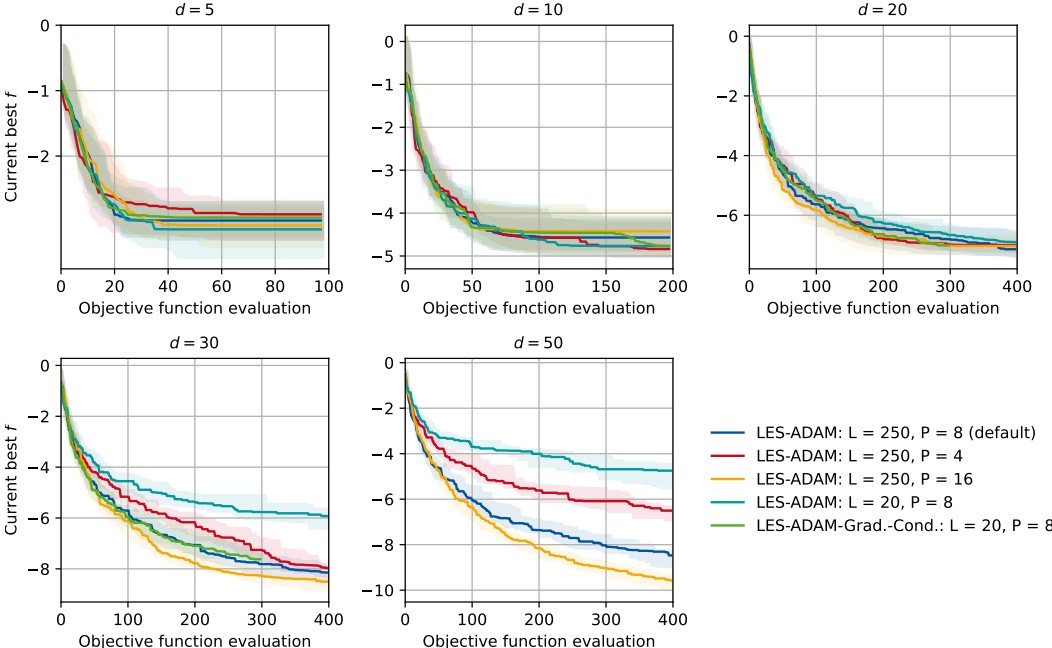

Figure 25: Comparing different acquisition function approximation accuracies - the better the approximation (larger value of $P$ and $L$) the better the results (Out-Of-Model Comparison on GP-Samples - medium complexity).

The computational cost of LES is closely related to the chosen discretization, i.e., the values of $L$ and $P$. To compare the influence of the acquisition function choice on overall runtime we evaluate it in the within-model comparison case. Table 11 shows the results for medium complexity. For comparison, we also include the runtime of our baselines. Additionally, Figure 26 shows the average runtime per iteration.

Our proposed approximation of LES requires roughly 10 times the wall-clock time compared to TuRBO, with an average of 17 seconds per iteration to select the next query. Notably, BoTorch's logEI has a similar runtime.

As a caveat, runtime depends strongly on several factors, including settings for acquisition function optimization. We used the default configurations from BoTorch tutorials for all baselines and did not optimize any baseline or our implementation for speed.

Table 11: Average computation time per iteration of LES-ADAM for the medium complexity within model comparison, i.e., without GP hyperparameter optimization. Results are given in seconds and are averaged over 20 seeds.

|  | $d = 5$ | $d = 10$ | $d = 20$ | $d = 30$ | $d = 50$ |
|---|---|---|---|---|---|
| LES: L = 250, P = 8 (default) | 11.4 | 12.6 | 16.5 | 17.1 | 17.6 |
| LES: L = 250, P = 4 | 10.6 | 11.7 | 14.0 | 14.3 | 15.4 |
| LES: L = 250, P = 16 | 13.8 | 16.6 | 24.5 | 24.1 | 25.3 |
| LES: L = 20, P = 8 | 3.5 | 3.8 | 4.0 | 4.2 | 4.3 |
| MES | 8.2 | 2.1 | 5.1 | 1.8 | 1.3 |
| TuRBO | 0.4 | 0.4 | 1.2 | 1.4 | 1.4 |
| logEI | 0.3 | 0.5 | 15.5 | 23.0 | 24.0 |

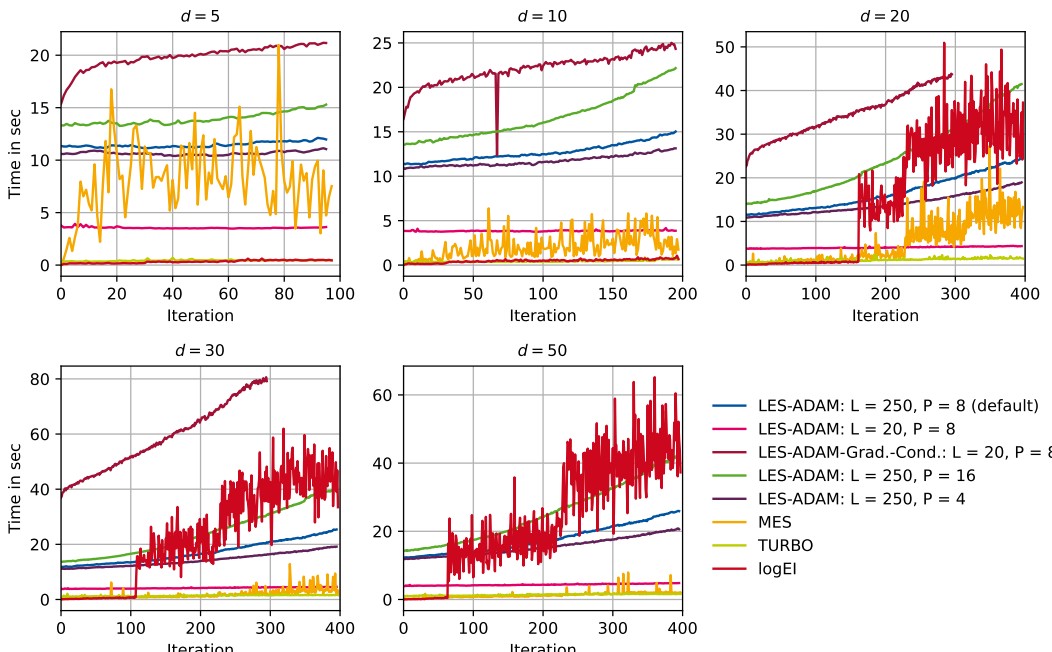

Figure 26: Evaluating LES with different local optimizers (Out-Of-Model Comparison on GP-Samples - low complexity)

## D.6 DESCENT SEQUENCE LENGTH AND STEP SIZE

To further evaluate the behavior of LES, we examine the average length of the descent sequences, i.e., $|z_0^l - z_P^l|$, with $z_0^l = \hat{x}_t^*$ and the step size (average distance between the new query and the current incumbent), i.e., $|x_t^* - x_{t+1}|$. Table 12 shows the medium-complexity within model-comparison case for LES-ADAM with default configuration. We aggregate results of the first and second half of the iterations.

Table 12: Average descent sequence length and step size (with % of seq. length in brackets) for LES-ADAM; L = 250, P = 8 (default): Comparison between first and second half of iterations.

| d | seq. length (1st half) | seq. length (2nd half) | step size (1st half) | step size (2nd half) |
|---|---|---|---|---|
| 5 | 0.0900 | 0.0051 | 0.0173 (19.2%) | 0.0026 (51.5%) |
| 10 | 0.1906 | 0.0200 | 0.0254 (13.3%) | 0.0068 (34.1%) |
| 20 | 0.3966 | 0.1206 | 0.0468 (11.8%) | 0.0737 (61.1%) |
| 30 | 0.6470 | 0.3597 | 0.0838 (12.9%) | 0.0627 (17.4%) |
| 50 | 0.9697 | 0.7946 | 0.1559 (16.1%) | 0.1274 (16.0%) |

We can observe that, the step size is always significantly smaller than the sequence length, so entropy reduction is not highest at the descent sequences endpoints. The step size and descent sequence length decrease in the second half of the optimization.

In addition, Fig. 27 shows the average step size and sequence length in the medium-complexity within model-comparison case for LES-ADAM with default configuration. We can see that both the sequence length and the step size decrease over time, which further indicates local search behavior.

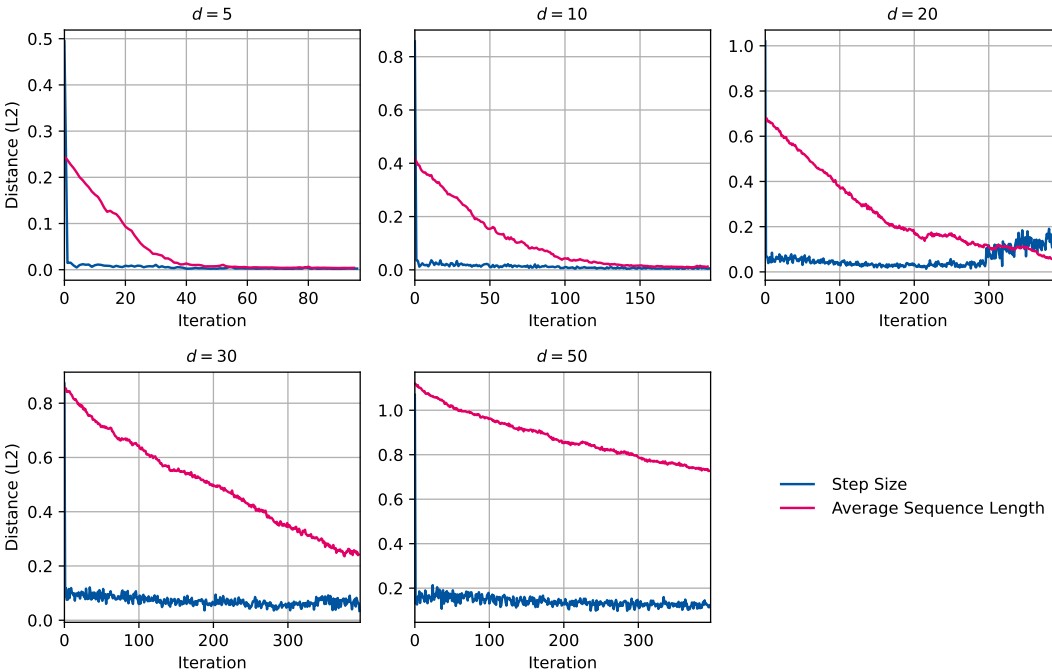

Figure 27: Average descent sequence length and step size for LES-ADAM; L = 250, P = 8 (default)

In summary, the LES framework automatically chooses the optimal (with respect to entropy reduction) exploration behavior/step size from the current model to discover the descent sequence.

### D.7 COMPARING DIFFERENT ITERATIVE OPTIMIZERS

If not stated otherwise, the ADAM optimizer was employed as the local optimization algorithm throughout this work. However, the general framework is applicable to any kind of iterative optimization. Therefore, we evaluated the impact of different iterative optimization schemes on the overall performance. We use the out of model comparison GP sample scenarios (see Sec. 6.2).

We expect that the performance of different inner optimizers also depends on the kernel. Sample paths from a GP with a Matérn 1/2 kernel are not continuously differentiable, making LES-ADAM and LES-GD less suitable choices. In such cases, LES with CMA-ES or other zeroth-order optimizers (e.g., hill climbing or pattern search) may perform better. We leave a more thorough evaluation of the benefits of different local optimizers in LES for future work.

**ADAM**   For Adam we use 500 local optimization steps a step size of 0.002, and default Keras momentum hyperparameter ($\beta_1 = 0.9$, $\beta_2 = 0.999$). As an alternative, we also evaluate a LES-ADAM variant with less aggressive gradient smoothing ($\beta_1 = 0.5$). Instead of conditioning on the gradient observations, we condition on function values. We show in Appx. D.5 that this simplification is justified empirically.

**Gradient Descent**   For Gradient Descent (GD) we also use 500 local optimization steps and a smaller learning rate of 0.0001. Preliminary results with a larger step size (the same as in ADAM) has produced oscillating behavior. Similarly to ADAM we condition on function values instead of gradient observations.

**CMAES**   As a third optimizer we choose CMAES and run it for 50 steps. We approximate the descent sequence by the mean of the parameter distribution. The $\sigma$ hyperparameter is set to 0.5 and similar to ADAM and GD we warm-start CMAES from the best solution found so far. Due to the high computational cost of running CMAES on multiple GP samples in each iteration, we evaluate it only in the 5-d and 10-d case and use a coarse discretization. Note that CMAES is a zeroth-order optimization algorithm and therefore does not require the GP-samples to be differentiable.

Figures 28 to 30 show the results for low, medium, and high complexity. LES-CMAES performs better than the other algorithms for $d = 5$ and low problem complexity. This may hint to a more global search behavior and may indicate that the optimizer's properties on the individual samples may carry over to the respective LES version. However, already at $d = 10$ or higher complexity LES-CMAES falls behind. This can be attributed to the worse performing global optimization or to the descent sequence approximation using the mean of the population not being accurate enough. Both LES-ADAM version perform similar in all complexities. LES-GD performs worse than LES-ADAM in the low-complexity case. However, in the medium and high complexity cases LES-GD performs best.

Overall, results highlight that investigating various local optimizers for different model properties is an interesting direction for future research.

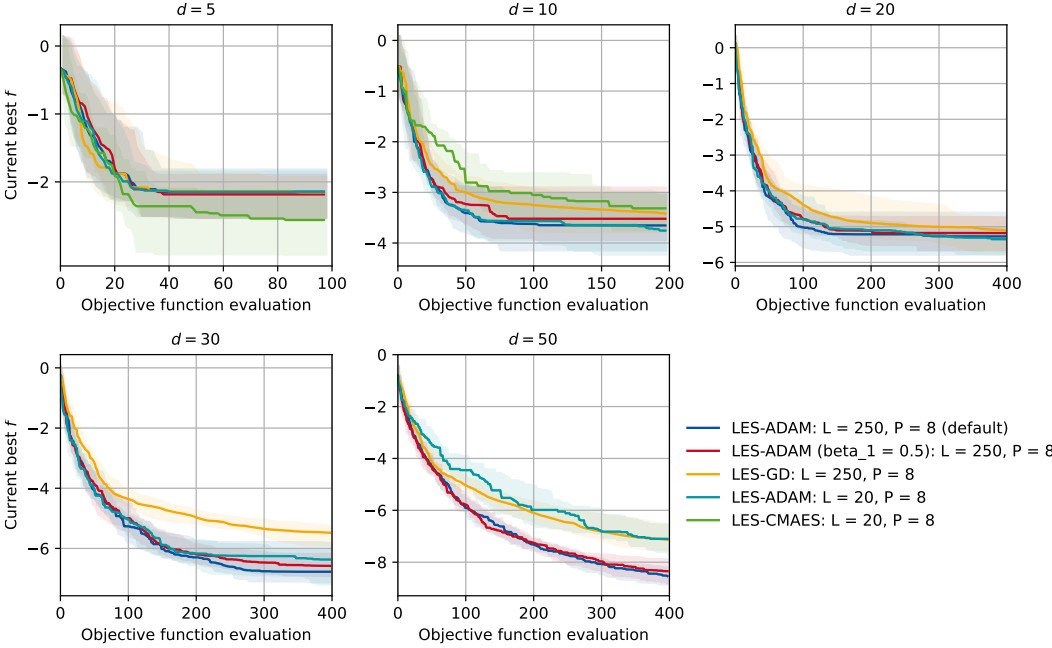

Figure 28: Evaluating LES with different local optimizers (Out-Of-Model Comparison on GP-Samples - low complexity)

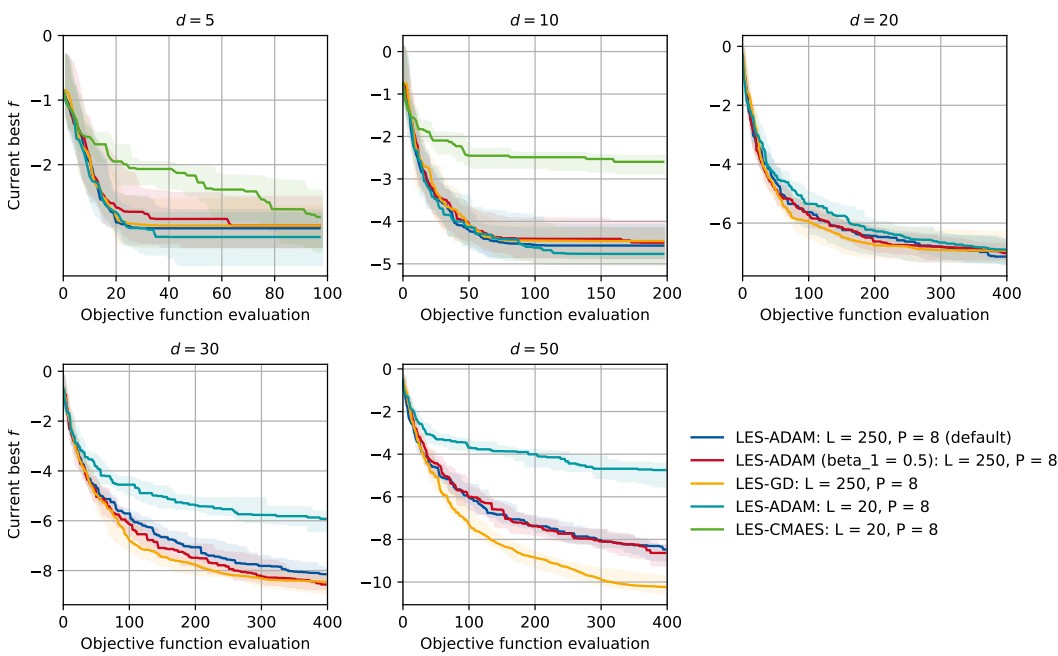

Figure 29: Evaluating LES with different local optimizers (Out-Of-Model Comparison on GP-Samples - medium complexity)

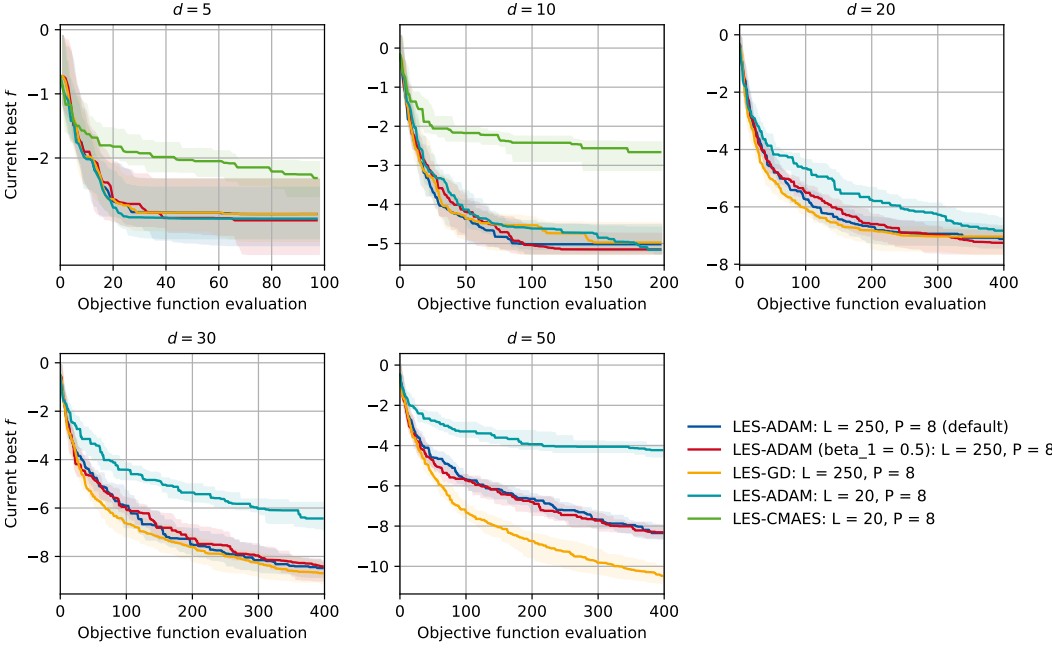

Figure 30: Evaluating LES with different local optimizers (Out-Of-Model Comparison on GP-Samples - high complexity)

# E  A STOPPING CRITERION FOR LOCAL ENTROPY SEARCH

## E.1  METHOD

In practical applications, it is essential to determine when to terminate the optimization. In the previous section, we performed local optimization on multiple posterior samples. This enables us to adopt the Monte-Carlo-based stopping criterion proposed by (Wilson, 2024) in a local setting without incurring significant additional cost, as we can directly reuse the samples generated during the acquisition step. To achieve this we define a new notion of local regret:

**Definition 1** (Local simple regret). *Given the posterior GP $f_t$ at BO step $t$, the local simple regret with respect to the model-based local optimum $f_t^\star = \sup_x f_t(x)$ of a candidate point $x \in \mathcal{X}$ is*

$$r_t^{\mathcal{O}}(x) \;=\; f_t^\star - f_t(x). \tag{31}$$

Local regret formalizes how close we are to the best value reachable from the user's initial guess by the optimizer $\mathcal{O}$. This is attractive when the global optimum is irrelevant or unattainable in practice. We stop the optimization if the current solution is within $\varepsilon$ of the optimum with probability $\delta$, i.e., when it is $(\varepsilon, \delta)$ locally optimal:

**Definition 2** ($(\varepsilon, \delta)$-local optimality). *Fix tolerances $\varepsilon > 0$ and $\delta \in (0,1)$. A point $x$ observed up to step $t$ is $(\varepsilon, \delta)$-locally-optimal (with respect to $x_0$ and optimizer $\mathcal{O}$) if*

$$\Pr\Big[r_t^{\mathcal{O}}(x) \leq \varepsilon\Big] \;\geq\; 1 - \delta. \tag{32}$$

We estimate the probability of the regret being smaller than epsilon using Monte-Carlo sampling:

$$\Pr\left(r_t \leq \varepsilon\right) \approx \frac{1}{L} \sum_{l=1}^{L} \mathbb{1}\left(r_t^l \leq \varepsilon\right) = \frac{1}{L} \sum_{l=1}^{L} \mathbb{1}\left(f^l(\hat{\boldsymbol{x}}_t^*) - f^l(\boldsymbol{x}^{l,*}) \leq \varepsilon\right). \tag{33}$$

We follow (Wilson, 2024) and design a probabilistic stopping rule that leads to bounded *local* regret (31) with high (within-model) probability. Next, we restate the formal results from (Wilson, 2024) for LES.

**Assumption 1.**

- The search space is the unit hyper-cube $\mathcal{X} = [0,1]^D$.
- There exists a constant $L_k > 0$ so that $\forall\, x, x' \in \mathcal{X}$, $\big|k(x,x) - k(x,x')\big| \;\leq\; L_k \left\|x - x'\right\|_\infty$.
- The sequence of query locations $(x_t)$ is almost surely dense in $\mathcal{X}$.

We show in Appx. G.1 that LES fulfills the third assumption for specific kernels.

**Theorem 1** (Proposition 2, (Wilson, 2024)). *Assume 1. Given a risk tolerance $\delta > 0$, define non-zero probabilities $\delta_{\mathrm{mod}}$ and $\delta_{\mathrm{est}}$ such that $\delta_{\mathrm{mod}} + \delta_{\mathrm{est}} \leq \delta$ and let $\left(\delta_{\mathrm{test}}^t\right)_{t \geq 0}$ be a positive sequence so that $\sum_{t=0}^{\infty} \delta_{\mathrm{test}}^t \leq \delta_{\mathrm{est}}$. For any regret bound $\varepsilon > 0$, if the Monte-Carlo test of (Wilson, 2024, Alg. 2) is run at each step $t \in \mathbb{N}_0$ with tolerance $\delta_{\mathrm{test}}^t$ to decide whether a point satisfies $\Pr\Big[r_t^{\mathcal{O}}(x) \leq \varepsilon\Big] \;\geq\; 1 - \delta_{\mathrm{mod}}$, then LES almost surely terminates and returns a solution with probabilistic regret that satisfies Definition 2.*

## E.2  RESULTS

The stopping times (Sec. E.1) for LES on the out-of-model comparison with low model complexity are in Tab. 13. For $d \geq 30$ fewer than half the runs stop within the budget of 400 evaluations. When compared to the results in (Wilson, 2024) these results show that the local optimization needs fewer samples before stopping. These results reinforce the intuition that reaching a local optimum is easier than reaching a global one – even in black-box optimization problems.

Table 13: Median stopping times with decision every 25 queries ($\delta = 0.05$) - out of model comparison.

| $\varepsilon$ | $d = 5$ | $d = 10$ | $d = 20$ |
|---|---|---|---|
| 0.1 | 50 | 150 | 325 |
| 0.01 | 50 | 175 | 350 |

Table 14 summarizes the results for the stopping rule in the within model comparison case with low problem complexity. For comparison, (Wilson, 2024) reported an average stopping time after 100 queries for the 4-d within-model case. Note that we choose the results of (Wilson, 2024) for the low noise case, since it is most fitting to our experiments. The parameters reported in Table 3 lead to a BO run being stopped after $k_{\max} = 248$ out of $L = 250$ samples show local regret smaller than $\varepsilon$.

Table 14: Number of queries until half of the runs are stopped ($\delta = 0.05$). Decision every 25 queries - within model comparison.

| $\varepsilon$ | $d = 5$ | $d = 10$ | $d = 20$ |
|---|---|---|---|
| 0.1 | 50 | 150 | 275 |
| 0.01 | 50 | 175 | 300 |

# F    ALTERNATIVE INFORMATION-THEORETIC LOCAL ACQUISITION FUNCTIONS

We propose two additional information-theoretic local acquisition functions that are closely related to the local entropy search paradigm: Local Thompson sampling and local-optimum LES. Both are conceptually more straight forward and easier to compute than LES but perform worse.

## F.1    LOCAL THOMPSON SAMPLING

In local Thompson sampling (L-TS) we sample only one path from the GP, which we then minimize locally using the ADAM optimizer and query at its minimum. This method proves to be significantly more computationally efficient than the entropy search approach, as it avoids the need for the relatively costly Monte Carlo approximation outlined in equation (16).

We assess local Thompson sampling to better understand the benefits of considering the distribution over descent sequences at each iteration, as implemented in LES. We expect that L-TS may not perform as well as LES, since L-TS optimizes a single descent sequence in a greedy manner. In contrast, LES recognizes that multiple descent sequences exist and seeks to maximize information gain across all of them.

## F.2    CONDITIONING ONLY ON THE LOCAL OPTIMUM

In Appx. B.1, we have shown that directly conditioning on the local optimum is not possible in general. That is, we cannot condition a GP on $O_{\boldsymbol{x}_0}^{*,l}$ in

$$\mathbb{E}_f \left[ \mathrm{H} \left[ p \left( y(\boldsymbol{x}) \mid \mathcal{D}_t, O_{\boldsymbol{x}_0}^* \right) \right] \right] \approx \frac{1}{L} \sum_{l=1}^{L} \mathrm{H} \left[ p \left( y(\boldsymbol{x}) \mid \mathcal{D}_t, O_{\boldsymbol{x}_0}^{*,l} \right) \right]. \tag{34}$$

We cannot encode that the local optimum at location $\boldsymbol{x}^{*,l}$ was reached through a local optimizer from point $\boldsymbol{x}_0$. What we can encode is that $\boldsymbol{x}^{*,l}$ is a local optimum defined by a gradient of zero and a positive (known) Hessian:

$$\begin{aligned}
&\frac{1}{L} \sum_{l=1}^{L} \mathrm{H} \left[ p \left( y(\boldsymbol{x}) \mid \mathcal{D}_t, O_{\boldsymbol{x}_0}^{l} \right) \right] \\
&\approx \frac{1}{L} \sum_{l=1}^{L} \mathrm{H} \left[ p \left( y(\boldsymbol{x}) \mid \mathcal{D}_t \cup (\boldsymbol{x}^{*,l}, f^l(\boldsymbol{x}^{*,l})), (\boldsymbol{x}^{*,l}, \nabla f^l(\boldsymbol{x}^{*,l})), (\boldsymbol{x}^{*,l}, \Delta f^l(\boldsymbol{x}^{*,l}))) \right) \right]
\end{aligned} \tag{35}$$

Note that the exact values of the observations again are irrelevant for the conditional entropy. This gives rise to the LES-ADAM Opt. Cond. acquisition function.

We evaluate LES-ADAM Opt. Cond. because we aim to demonstrate that the sequence leading to the local optima contains valuable information; merely conditioning on the local optimum—the final point of this descent sequence—is insufficient.

## F.3    RESULTS

Figure 31 shows that LES-ADAM outperforms the other information-theoretic approaches, with the performance gap widening in higher dimensions. The experiments are conducted in the out-of-model GP sample scenario with medium complexity (see Sec. 6.2). Overall, the results indicate that considering multiple descent sequences per iteration, as well as the entire descent sequence rather than only the distribution of local optima, improves performance.

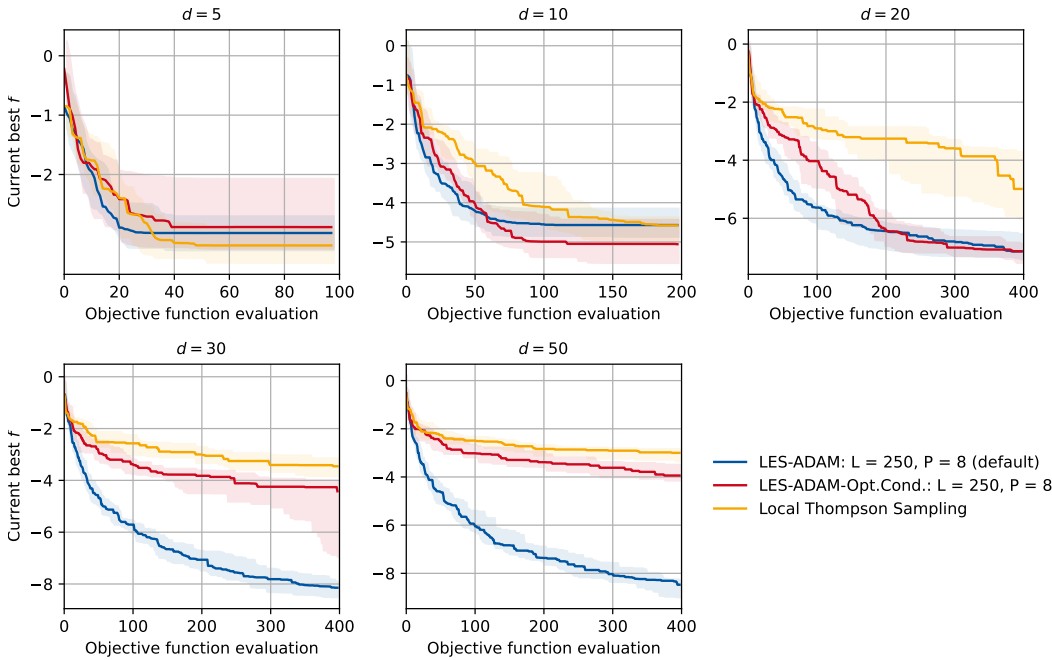

Figure 31: Median, 25-, and 75-percent quantiles, out-Of-model comparison on GP-Samples - medium complexity).

## G  THEORETICAL RESULTS

This section contains some theoretical results in support of the main claims of the paper. We use the notational shorthand $k_t(\cdot,\cdot) = k(\cdot,\cdot \mid \mathcal{D}_t)$.

### G.1  LES QUERIES ARE DENSE

**Lemma 1** (Density of LES maximizers). *Assume $\mathcal{X} = [0,1]^D$ and let $k_t : \mathcal{X} \times \mathcal{X} \to \mathbb{R}$ be a continuous, positive-definite kernel that admits the no-empty-ball property (Wilson, 2024, Definition 3), $\forall \boldsymbol{x}, \boldsymbol{x}' \notin \mathcal{D}_t\ k(\boldsymbol{x}, \boldsymbol{x}') > 0$, and the descent sequence $(\boldsymbol{z}_n)$ contains at least one $\boldsymbol{z} \notin \mathcal{D}_t$ almost surely. Fix a noise variance $\gamma^2 \geq 0$ and denote by $k_t$, $\sigma_t^2$ the posterior covariance and predictive variance after $t$ evaluations. Then, for every $t \in \mathbb{N}$ and all $\boldsymbol{x}, \boldsymbol{x}' \in \mathcal{X}$,*

$$k_t(\boldsymbol{x}, \boldsymbol{x}) >\ k_t(\boldsymbol{x}', \boldsymbol{x}') = 0 \implies \alpha_{\mathrm{LES},t}(\boldsymbol{x}) > \alpha_{\mathrm{LES},t}(\boldsymbol{x}') \tag{36}$$

*Consequently the sequence $(\boldsymbol{x}_t)$ of maximizers $\boldsymbol{x}_t \in \arg\max_{\boldsymbol{x} \in \mathcal{X}} \alpha_{\mathrm{LES},t}(\boldsymbol{x})$ form a dense sequence in $\mathcal{X}$ almost surely.*

*Proof.* If $k_t(\boldsymbol{x}', \boldsymbol{x}') = 0$, positive-semidefiniteness implies $k_t(\boldsymbol{x}', \boldsymbol{z}) = 0$ for every $\boldsymbol{z} \in \mathcal{X}$. Hence, $\sigma_t^2(\boldsymbol{x}') = \sigma_{t \mid X_{\mathrm{DS}}}^2(\boldsymbol{x}') = \gamma^2$ for every Monte-Carlo draw $X_{\mathrm{DS}}$, so $\alpha_{\mathrm{LES},t}(\boldsymbol{x}') = 0$.

Take $\boldsymbol{x}$ with $k_t(\boldsymbol{x}, \boldsymbol{x}) > 0$. Because the the descent sequence $(\boldsymbol{z}_n)$ contains at least one $\boldsymbol{z} \notin \mathcal{D}_t$ almost surely the posterior covariance is not zero the covariance vector $k_t(\boldsymbol{x}, X_{\mathrm{DS}})$ is non-zero. The GP variance update gives

$$\sigma_{t \mid X_{\mathrm{DS}}}^2(\boldsymbol{x}) = \sigma_t^2(\boldsymbol{x}) - k_t(\boldsymbol{x}, X_{\mathrm{DS}})\big[K_t(X_{\mathrm{DS}}, X_{\mathrm{DS}}) + \gamma^2 I\big]^{-1} k_t(X_{\mathrm{DS}}, \boldsymbol{x}), \tag{37}$$

and the quadratic form on the right is strictly positive, hence $\sigma_{t \mid X_{\mathrm{DS}}}^2(\boldsymbol{x}) < \sigma_t^2(\boldsymbol{x})$. Therefore each logarithm inside $\alpha_{\mathrm{LES},t}(\boldsymbol{x})$ is positive and their expectation is strictly positive $\alpha_{\mathrm{LES},t}(\boldsymbol{x}) > 0$.

Density of the query points now follow from (Wilson, 2024, Proposition 4). □

## G.2 GRADIENT DESCENT PATHS UNDER A GP PRIOR WITH A SQUARED EXPONENTIAL KERNEL

In this section we zoom in on a specific instantiation of LES that samples candidate points by running gradient-descent (8) on functions drawn from a squared-exponential (SE) GP prior. This section explains why that particular pairing is a sensible starting point. We show, with a suitably step size, a gradient descent sequence starting from an initial design can reach any subset of the domain with positive prior probability. In addition, for any finite horizon the distribution of such sequences has full support on the corresponding product space and can therefore realize any finite sequence. Importantly, these reachability and support guarantees ensure that no part of the search space is ruled out by construction in this setting.

**Assumption 2** (Design domain). The search space $\mathcal{X} \subset \mathbb{R}^d$ is non-empty, compact, convex, and has non-empty interior.

We formalize the assumptions as follows.

**Assumption 3** (Step size). Let $\mathcal{X} \subset \mathbb{R}^D$ be compact. For every realized objective $f \in \mathcal{C}^1(\mathcal{X})$ denote by $L(f)$ the global Lipschitz constant of its gradient. Choose a step size $\eta(f) > 0$ satisfying

$$\eta(f)\, L(f) \;<\; 1.$$

With this choice the gradient–descent map $\Phi_f(\boldsymbol{x}) \;=\; \boldsymbol{x} - \eta(f)\,\nabla f(\boldsymbol{x})$ is a strict contraction on $\mathcal{X}$.

**Assumption 4** (Squared–exponential prior). The objective is a random draw from $p(f) \sim GP\big(0, k_{\mathrm{SE}}\big)$, so that $f \in C^\infty(\mathcal{X})$ almost surely and the associated RKHS is dense in $C^\infty(\mathcal{X})$ with the $C^1$-norm.

**Lemma 2** (Open–set reachability of GD paths). *Let Assumption 2, 3, 4 hold. Define the gradient–descent iterates as in (8) so that*

$$\boldsymbol{z}_0 \in \mathcal{X}, \qquad \boldsymbol{z}_{n+1} = \boldsymbol{z}_n - \eta \nabla f(\boldsymbol{z}_n), \quad n \geq 0. \tag{38}$$

*Then for every non-empty open set $U \subset \mathcal{X}$*

$$\Pr_f\Big[(\boldsymbol{z}_n) \cap U \neq \varnothing\Big] \;>\; 0. \tag{39}$$

*Proof.* Fix $U \neq \varnothing$ open and choose $u \in U$ and a horizon $m \in \mathbb{N}$.

Since the map $\Phi_f(\boldsymbol{z}) = \boldsymbol{z} - \eta \nabla f(\boldsymbol{z})$ is a contraction (Assumption 3) one can constructs a smooth function $f^\star \in C^\infty(\mathcal{X})$ such that

$$\boldsymbol{z}_m(f^\star) = u, \qquad \|\nabla f^\star\|_{\mathrm{Lip}} \leq L_\star. \tag{40}$$

Since the RKHS $\mathcal{H}_{k_{\mathrm{SE}}}$ of the squared–exponential kernel is dense in $C^\infty(\mathcal{X})$, every $C^1$–ball $B_\varepsilon(f^\star) = \{f : \|f - f^\star\|_{C^1} < \varepsilon\}$ contains at least one element of $\mathcal{H}_{k_{\mathrm{SE}}}$. By the Gaussian–measure support theorem (Vaart, van der & Zanten, van, 2008, Lem. 5.1), every open set that intersects $\mathcal{H}_{k_{\mathrm{SE}}}$ has positive prior probability, hence

$$\Pr\big[\|f - f^\star\|_{C^1} < \varepsilon\big] \;>\; 0. \tag{41}$$

The mapping $C^1(\mathcal{X}) \ni g \longmapsto \Phi_g^{(m)}(\boldsymbol{x}_0)$ is continuous in the $C^1$-norm when $\eta L_\star < 1$. Hence there exists $\varepsilon > 0$ such that $\|f - f^\star\|_{C^1} < \varepsilon \implies \boldsymbol{x}_m(f) \in U$.

Combining this with (41) gives $\Pr\big[\boldsymbol{x}_m(f) \in U\big] > 0$, which implies (39). $\square$

**Corollary 1** (Full support of finite GD paths). *Fix an integer horizon $N \geq 0$ and define*

$$\mathcal{O} : C^1(\mathcal{X}) \longrightarrow \mathcal{X}^{N+1}, \qquad \mathcal{O}(f) := (\boldsymbol{x}_0, \boldsymbol{x}_1, \ldots, \boldsymbol{x}_N), \tag{42}$$

*where $(\boldsymbol{x}_t)$ are obtained by gradient descent. Under Assumptions 2–4, the push-forward measure $\mathcal{O}_\sharp\big[GP(0, k_{\mathrm{SE}})\big]$ has full support on $\mathcal{X}^{N+1}$; i.e., for every open cylinder set $U_0 \times \cdots \times U_N \subset \mathcal{X}^{N+1}$ with all $U_t$ open and non-empty,*

$$\Pr_f\Big[(\boldsymbol{x}_0, \ldots, \boldsymbol{x}_N) \in U_0 \times \cdots \times U_N\Big] \;>\; 0. \tag{43}$$

*Proof.* Proceed inductively on $N$.

*Base case $N = 0$:* trivial because $\boldsymbol{x}_0$ is fixed.

*Inductive step:* Assume the claim holds up to horizon $N - 1$. Given open $U_0, \ldots, U_N$, the induction hypothesis provides a function $f^\star$ and $\varepsilon > 0$ such that $\|f - f^\star\|_{C^1} < \varepsilon$ implies $(\boldsymbol{x}_0, \ldots, \boldsymbol{x}_{N-1}) \in U_0 \times \cdots \times U_{N-1}$. Apply Lemma 2 with starting point $\boldsymbol{x}_{N-1}(f^\star)$ and target open set $U_N$ to obtain a further refinement $\varepsilon'$. Choose $\delta = \min\{\varepsilon, \varepsilon'\}$ and use the support property (41) to conclude the probability is positive. $\qquad\square$

As long as $k_t(\boldsymbol{x}, \boldsymbol{x}) > 0 \ \ \forall \boldsymbol{x} \in \mathcal{X}$ the same applies to the posterior. The proofs are identical because conditioning on finitely many points does not change the RKHS nor the support of the measure; it only shifts the mean.

In summary, Lemmas 2 and 1 together show that, in the SE-prior/GD instantiation of LES, the candidate-generation mechanism is fully expressive: no open region is inaccessible and no finite GD sequence is excluded.

