# OpenReview forum: "Local Entropy Search over Descent Sequences for Bayesian Optimization"
_ICLR.cc/2026/Conference — ICLR 2026 Poster_

### Official Review · Reviewer_8PaV · 2025-10-27

**Soundness:** 2
**Presentation:** 3
**Contribution:** 2
**Rating:** 6
**Confidence:** 4

**Summary:**

The paper proposes a new Acquisition Function (ACF) for Bayesian Optimization (BO), which especially improves the local search using the idea from two other main papers (Hennig & Schuler, 2012) and (Muller et al., 2021). In general, replacing the conventional sampler of the ACF at one point with a trained Gaussian Process model (GP), then maximizing utilities, this method proposes a gradient descent-based ACF with finite sequences to determine the potential candidates. The gradients of the GP model are used to create the sequences from different samples, and the potential candidates should minimize the entropy of these sequences, based on the assumption that reaching local extrema will have a stable gradient path. Their main contribution is the improvement of the stability and robustness of the local search within the GP model. Experimentally, the algorithm delivers remarkable performance in synthetic benchmarks, while showing marginal improvement in real-world problems. In addition, the ACF-only benchmarks are impressive, and the computational cost is handled well. The discussion is well done, which explains clearly the working conditions and the limitations of the approach. Overall, this work proves a slight contribution, which can potentially improve the outputs of low-dimensional problems (under 100 dimensions).

**Strengths:**

This work demonstrates several notable strengths across originality, quality, clarity, and significance. First, it introduces a comprehensive and well-structured benchmarking framework, making the experimental evaluation both reproducible and conceptually clear. The presentation is straightforward, allowing readers to easily grasp the main ideas without being overwhelmed by technical complexity. Second, the paper’s core contribution, an efficient entropy-based search strategy for finding optimal values within GP models, is conceptually sound and addresses a gap in current Bayesian optimization approaches, where inner GP models' search efficiency has been underexplored. Third, the authors’ methodological choices, including their assumptions and theoretical lemmas, are particularly aligned with the challenges of local optimization under uncertainty, especially the computational cost problem. Finally, the synthetic benchmark results are consistently strong, showing that the proposed Local Entropy Search (LES) method outperforms existing baselines across multiple complexity levels, which is both empirically impressive and practically significant.

**Weaknesses:**

While the paper is well written and experimentally solid, several aspects could be strengthened to better support its claims. First, the motivation for emphasizing local search over global exploration is not clearly justified. The paper references use cases in neural network optimization and other machine learning settings, which differ substantially from the BO context. Although improving local search efficiency is meaningful, the trade-off between local exploitation and global convergence remains underexplored. As a result, the proposed method effectively bridges an inner-GP optimization gap but simultaneously introduces ambiguity regarding its global behavior.

Second, the experimental evaluation is limited primarily to synthetic benchmarks. While these results are strong, the real-world tasks show marginal improvement, and the performance quickly degrades when computational budgets are constrained. They did compare with the current state-of-the-art high-dimensional BO methods, but the benchmarks are set for low-dimensional problems. Besides, their synthetic benchmark is GP-based, which raises a question of the harmony between the target model and the optimizer GP model. Third, although the authors provide lemmas ensuring that the acquisition function (ACF) can cover the entire input space, they omit theoretical analysis on regret bounds and convergence guarantees. Aligned with the first problem, the convergence condition is properly needed in this case, despite the logical assumption that reaching local extrema will have a stable gradient path. Finally, while computational efficiency is discussed, the ACF evaluation may become a bottleneck in high-dimensional settings, as it dominates runtime when the dimension increases. Since improving acquisition-function efficiency is an active research direction, this approach may be less competitive in high-dimensional or highly multi-modal problems. Consequently, LES performs strongly in low-dimensional spaces with relatively few local optima but may struggle to maintain its advantages as dimensionality grows. This shows a narrow possible implementation range for LES, where it matches the ideal condition.

**Questions:**

There are some concerns that need to be clarified to strengthen the robustness of the work.

Is there any mathematical proof or empirical validation regarding the behaviour of the LES ACF, particularly concerning the issue of incumbent search? This could be clarified if the data-point maps indicate whether the samples are regionally concentrated. The incumbent search problem was raised by (Hvarfner et al., 2024). Although the authors demonstrate that the LES ACF can balance exploration and exploitation through entropy minimization and achieve full-space coverage, stabilizing sequential gradient chains may still induce incumbent search and reduce the ability to perform large exploratory jumps. Even though the use of optimizers such as ADAM supports escaping local optima, and the authors state that the global value is not the target, being easily trapped in local minima remains a major bottleneck. Furthermore, regionally clustered data points may cause the hyperparameters to reflect only the characteristics of nearby areas, making it difficult to escape those regions. Addressing this could lead to discovering higher-quality local optima or even the global optimum.
In the experimental setting, the number of initialization samples is only two. Why is this number so low compared with other works? Has any empirical study been conducted to justify this hyperparameter choice? At the beginning, with only two samples, BO behaviour is purely exploratory or even random, relying heavily on the prior hyperparameters. BO typically requires a sufficient number of observations to adequately represent the search space, whereas the LES ACF amplifies this early-stage phenomenon by over-exploiting incumbent values. Consequently, the exploration phase, crucial for finding global solutions, is further reduced during the stage that contributes most to global discovery. A minor question: since BoTorch supports standardization, what does the statement “data is not standardized” specifically mean?
(Hvarfner et al., 2024) claim their method improves performance in high-dimensional problems, most exceeding 100 dimensions. A benchmark with a higher dimensionality is therefore necessary to clearly reveal the limitations. Common synthetic benchmarks such as Levy or Hartmann should be included for more comprehensive comparison. Other local search methods have also shown strong results without neglecting global optimization, such as BAxUS (Papenmeier et al., 2022). The concern about BO-based benchmarks arises from the alignment between the target model and the surrogate model used by the optimizer. When both share similar characteristics and nearly identical uncertainty-quantification mechanisms, especially in within-model comparisons, the results may be biased. This study employs Matheron’s rule for sampling, which provides superior posterior draws, while other methods do not leverage it. Thus, much of the observed improvement might be attributed to the Matheron’s rule sampler rather than the LES ACF itself. Adding additional synthetic benchmarks would help clarify this effect. There is no need to test multiple levels of GP-based benchmarks.
There are further suggestions for improving the presentation:

Claiming that the global optimum is not the objective is acceptable; however, the ability to escape local optima deserves stronger emphasis. Completely disregarding global-optimum search is a significant drawback that limits the method’s applicability. If this design choice is justified more clearly and explained earlier in the paper, the overall persuasiveness would increase considerably.
For your consideration, it may be beneficial to adopt a multi-start ACF. Although this may slow down computation, it could prevent the method from getting trapped in a single region. Using only one current best value is risky; incorporating two or three additional starting points in different regions, while reducing the number of gradient chains, could achieve a better balance between efficiency and reliability. Increasing the number of initial points, as noted above, would also enhance robustness and exploration consistency. Combining an exploration-oriented ACF with the LES ACF could further expand the applicability beyond strictly local optimization tasks.
A brief note on structure: Section 4 contains only one subsection (4.1) and no 4.2, which seems redundant. Figure 4 appears in isolation and is discussed almost two pages later, perhaps due to layout constraints, but this should be improved. Early in the paper, when introducing the concept of gradient-based BO and entropy-based ACF, Figure 1 does not effectively illustrate the overall concept. A clearer overview diagram would help, ideally explaining the gradient-chain mechanism directly instead of using dense phrasing such as “The algorithm propagates the posterior belief over the objective through the optimizer, yielding a probability distribution over descent sequences.”

---

> ### Author Response · Authors · 2025-11-23
>
> ## Weaknesses
>
> >[...] First, the motivation for emphasizing local search over global exploration is not clearly justified. [...]
>
> LES is intentionally scoped as a local method. When the landscape is simple, global and local approaches reliably reach the global optimum. However, in high-complexity settings, global BO often fails to find good solutions, whereas consistently reaching a good local optimum remains feasible. In such cases, prioritizing local search leads to better practical performance.
>
> We added a sentence in the related-work section noting that several prior local BO approaches have been shown to outperform global BO on complex tasks, which is why local BO has become an active and growing research direction. An additional motivation for emphasizing local search is its more conservative exploration: as shown in Appendix D.3, LES’s localized behavior results in significantly reduced cumulative regret.
>
> >Second, the experimental evaluation is limited primarily to synthetic benchmarks. [...]
>
> Our primary contribution is the LES acquisition function, and our evaluation is designed to isolate and examine its behavior across controlled levels of complexity. As our results show, when the surrogate model fits the task well, LES delivers clear gains over baselines, particularly on high-complexity synthetic problems. In real-world tasks with limited budgets and potential model mismatch, the performance differences are naturally smaller. We explicitly acknowledge this in the limitations section. Addressing model–objective mismatch is important, but orthogonal to the acquisition-function contribution of this work.
>
> Regarding dimensionality, we note that evaluating entropy-search–style methods at 50 dimensions is already considered challenging. For example, Joint Entropy Search [1] is evaluated only up to 12 dimensions.
>
> [1] Hvarfner, Carl, Frank Hutter, and Luigi Nardi. “Joint entropy search for maximally-informed Bayesian optimization.” NeurIPS 2022.
>
> >Finally, while computational efficiency is discussed, the ACF evaluation may become a bottleneck in high-dimensional settings, as it dominates runtime when the dimension increases. [...]
>
> Our experiments show that LES performs particularly well on high-complexity landscapes, where both global and local BO baselines struggle. In these settings, the additional computational cost of the acquisition function can be offset by the gain in sample efficiency.
>
> ## Questions
> > Is there any mathematical proof or empirical validation regarding the behaviour of the LES ACF, particularly concerning the issue of incumbent search [...]?
>
> We rely on the standard choice of using the minimum posterior mean as the incumbent and did not explore alternative incumbent-search strategies in this work. Empirically, this simple rule performed well across all our experiments. As noted in our response to Reviewer o3dG, LES is compatible with more advanced incumbent-search (and multi-start) mechanisms.
>
>
> > [...] In the experimental setting, the number of initialization samples is only two. [...]
>
> The main focus of this paper is local search, so we intentionally keep the number of initial evaluations to a minimum. Using many randomly chosen initial points would implicitly introduce a form of global exploration. Our goal is to evaluate how well each method performs when starting from very limited initial information.
>
> > [...] what does the statement “data is not standardized” specifically mean?
>
> For the synthetic benchmarks where the objective is itself a GP sample path, we do not standardize the query responses before passing them to the optimizer because the values are already consistent with the GP prior and require no additional scaling. Standardization is applied only for objectives that are not GP sample paths.
>
>
> > This study employs Matheron’s rule for sampling, [...]
>
> Thank you for raising this point. LES uses Matheron’s rule because it provides efficient posterior draws for our acquisition function, but our baselines differ in how they generate samples. Importantly, two key baselines, logEI and HCI-GIBO, do not rely on posterior samples at all, so their performance is independent of our sampling strategy.
>
> > Claiming that the global optimum is not the objective is acceptable; however, the ability to escape local optima deserves stronger emphasis. [...]
>
> Thank you for these suggestions. LES is intentionally designed as a local BO method, and our experiments focus on understanding this local behavior. We agree that combining LES with multi-start strategies or additional exploratory components is an interesting direction for future work, and such extensions could broaden its applicability beyond the strictly local setting.
>
> > A brief note on structure: Section 4 contains only one subsection (4.1) and no 4.2, which seems redundant.[...]
>
> Thank you for this comment, we have revised the structure and layout in the updated paper.

---

### Official Review · Reviewer_gXxy · 2025-10-28

**Soundness:** 3
**Presentation:** 3
**Contribution:** 2
**Rating:** 4
**Confidence:** 3

**Summary:**

This paper proposes Local Entropy Search (LES) , a Bayesian optimization method for local optimization that aims to find the best solution reachable by an iterative optimizer starting from an initial design, rather than searching for the global optimum. The core idea is to reduce uncertainty about descent sequences by sampling GP posterior, running gradient optimizer on each sample to obtain finite step trajectories, then querying points that maximally reduce the average entropy of these trajectories. Empirically, LES outperforms global entropy search methods and existing local BO baselines on various benchmark problems up to 124 dimensions, though the method provides no theoretical guarantees and involves significant computational overhead compared to simpler local BO approaches.

**Strengths:**

- The acquistion function is very straightforward.
- The combination of entropy search with local search is freshing

**Weaknesses:**

- The empirical benchmark, although demonstrate up to 124 d, is on somwhat easy and not very representative BO benchmark, its unclear that the method will really be competitive in high input dimensionality or not. I think to demonstrate the real benifit of the acquisition function, which to me is the most promising point of this work, is to conduct more thorough empirical results to provide more compelling results.
- The algorithm is rather heuristic without too much theoretical insight. The existing theoretical analysis does not provide any insight.

**Questions:**

- Is there any intuition that the algorithm will work better in high dimension? Because this is slightly counterintuitive to me as I would imagine the local trajectory get randomly distributed in high dimenison, making an entropy reduction also very random.

---

> ### Author Response · Authors · 2025-11-23
>
> ## Weaknesses
>
> > The empirical benchmark, although demonstrate up to 124 d, is on somwhat easy and not very representative BO benchmark, its unclear that the method will really be competitive in high input dimensionality or not. I think to demonstrate the real benifit of the acquisition function, which to me is the most promising point of this work, is to conduct more thorough empirical results to provide more compelling results.
>
> We thank the reviewer for the comment. As discussed in the introduction, LES is designed for high-complexity settings, where small length scales and higher dimensionality make global BO challenging. Our empirical results, including Table 1, support this: LES consistently performs best or near-best as complexity and dimensionality increase. The benchmark we employ is sufficiently challenging to reveal meaningful performance differences across BO variants, and our evaluation spans 49 distinct settings.
>
> We agree that exploring even higher-dimensional cases is interesting future work, though we note that approximation accuracy and computational cost (see response to Question 1 and Appendix D) become limiting factors in very high dimensions. Furthermore, for entropy-search–style methods in particular, 50 dimensions is already considered a challenging regime; for example, Joint Entropy Search [1] is evaluated in up to 12 dimensions.
>
> [1] Hvarfner, Carl, Frank Hutter, and Luigi Nardi. “Joint entropy search for maximally-informed Bayesian optimization.” NeurIPS 2022.
>
> > The algorithm is rather heuristic without too much theoretical insight. The existing theoretical analysis does not provide any insight.
>
> We respectfully disagree with the assessment that LES is a heuristic. LES is a principled entropy-search formulation for local optimization, obtained by applying the ES mutual-information objective to the optimizer-induced random variable (the descent sequence).
>
> LES is derived directly and systematically from the entropy-search (ES) principle: define a random variable representing the quantity of interest and choose evaluations that maximally reduce its entropy via mutual information. In our case, the quantity of interest is the descent sequence of iterative optimizer. This is a natural local analogue of classical ES, where the quantity of interest is the global argmin. The construction is therefore fully aligned with prior information-theoretic BO work. Additionally, by adapting the stopping rule of Wilson (2024), LES provides a probabilistic local-optimality guarantee (Appendix F).
>
> ## Questions
>
> > Is there any intuition that the algorithm will work better in high dimension? Because this is slightly counterintuitive to me as I would imagine the local trajectory get randomly distributed in high dimenison, making an entropy reduction also very random.
>
> The core intuition behind LES in high dimensions is that reliably reaching a good local optimum typically requires much less information about the objective than identifying the global optimum. Even though early descent trajectories can be dispersed, LES only needs to model the function well enough along a single trajectory induced by the optimizer, not the entire high-dimensional landscape.
>
> In short, global BO must resolve uncertainty everywhere; LES must only resolve uncertainty along the much lower-dimensional descent manifold. This fundamental difference is why LES remains effective when dimensionality grows.
>
> More generally, as discussed in the related work, prior works have shown that for this reason local BO methods can outperform global BO in high-complexity problems.
>
> However, we agree that estimating the entropy of descent sequences becomes more demanding in higher dimensions, as more samples are required for accurate approximation. Appendix E.5 (Fig. 25) highlights this effect. Nonetheless, our results show that learning a single descent sequence is still markedly more sample-efficient than attempting global optimization, and LES benefits from this structural advantage.

---

> ### Comment · Reviewer_gXxy · 2025-11-25
>
> Thank the authors for their rebuttal. I think this is a nice paper combining entropy search with local search. There are some important factors, especially when connecting with the high-dimensional perspective, that I want to ensure understand correctly before modifying my evaluation.
>
> **On why entropy search works in high dimensions when the GP model becomes less reliable**: One fundamental motivation of GIBO (as well as TuRBO) is that in high dimensions, GP models are less accurate globally, so local search succeeds by only requiring local model accuracy. Entropy search, however, reduces uncertainty over entire descent sequence distributions. While this formulation is elegant, I cannot immediately appreciate why it suits high-dimensional settings where GP reliability is a concern. The authors argued that ``LES only needs to model the function well enough along a single trajectory,'' but empirical support would help: specifically, showing trajectory length distributions(e.g., $\|z_T - z_0\|$) across iterations. If trajectories remain locally concentrated, then LES naturally inherits TuRBO-like locality and this explains why it work.
>
> **On where information gain concentrates along trajectories**: Following the above, if LES reduces entropy over the whole sequence, I would hypothesis trajectory endpoints to dominate the acquisition since that is where uncertainty is highest. Could the authors decompose the entropy reduction along trajectory positions at each position $i$ (e.g., by exhaustive Monte Carlo approx)? This would clarify whether information gain concentrates near $x_0$ or toward endpoints.

---

> ### Author Response · Authors · 2025-11-28
>
> Dear reviewer, thank you for your answer and raising those points. We are aware, that you cannot answer or change your score after recent events but still want to answer these questions even though it may not impact the final score of this paper.
>
> In the tables below you can find the average length of the descent sequences, i.e.,
>
> $$\| z_0^{l} - z_T^{l}\|,$$
>
>  with $z_0^{l}=\hat{x}_t^*$ and the step size (average distance between the new query and the current incumbent), i.e.,
>
> $$\|x_t^* - x_{t+1}\|.$$
>
> The tables show the medium-complexity within model-comparison case for LES-ADAM with default configuration. We aggregate results of the first and second half of the iterations.
>
> ### First half of iterations
> | d | avg. sequence length | avg. step size (% of seq. length) |
> |---|----------------------|------------------|
> | 5 | 0.0900 | 0.0173 (19.2%) |
> | 10 | 0.1906 | 0.0254 (13.3%) |
> | 20 | 0.3966 | 0.0468 (11.8%) |
> | 30 | 0.6470 | 0.0838 (12.9%) |
> | 50 | 0.9697 | 0.1559 (16.1%) |
>
> ### Second half of iterations
> | d | avg. sequence length | avg. step size  (% of seq. length) |
> |---|----------------------|------------------|
> | 5 | 0.0051 | 0.0026 (51.5%) |
> | 10 | 0.0200 | 0.0068 (34.1%) |
> | 20 | 0.1206 | 0.0737 (61.1%) |
> | 30 | 0.3597 | 0.0627 (17.4%) |
> | 50 | 0.7946 | 0.1274 (16.0%) |
>
> We can observe that
> - the step size is always significantly smaller than the sequence length, so entropy reduction is not highest at the descent sequences endpoints,
> - and the step size and descent sequence length decrease in the second half of the optimization as convergence progresses.
>
> In addition, we plotted the sequence length and the step size during an optimization run and see that both the sequence length and the step size decrease over time, which further indicates local search behavior.
>
> In summary, the LES framework automatically chooses the optimal (with respect to entropy reduction) exploration behavior/step size from the current model to discover the descent sequence. We will make sure to add a short section in the appendix of the camera-ready paper to highlight these results.

---

### Official Review · Reviewer_yURN · 2025-10-30

**Soundness:** 3
**Presentation:** 3
**Contribution:** 3
**Rating:** 8
**Confidence:** 3

**Summary:**

This paper considers the problem of black-box optimization, in which one is trying to optimize a function while having access only to noisy function calls: no gradient, convexity, or other information is available.
In this setting, Bayesian Optimization (BO) is the state-of-the-art approach. BO learns a statistical surrogate for the objective and plugs it into an _acquisition function_ (AF), whose maximization yields the next design to evaluate, in an exploration-exploitation balancing way.
The authors contribute to this field with the introduction of a novel acquisition function, Local Entropy Search (LES), which belongs to the realm of information-theoretic AFs. Such AFs aim to find designs that maximize the _information gain_ about a random quantity, usually of interest for BO purposes, like the optimum $f^\star$, its location $x^\star$, or even $(x^\star,f^\star)$. The proposed LES selects a design that maximizes the information gain about a _local optimum_, here characterized as the last iterate of an arbitrary optimizer like gradient descent. After introducing their theoretical framework, the authors derive an efficient algorithm for computing LES and perform an extensive set of experiments, comparing LES against other state-of-the-art baselines on multiple synthetic and real-world examples, jointly with numerous ablation studies. LES emerges as a strong competitor by delivering the best overall performance, thus providing empirical evidence about the relevance of focusing on local optima rather than global optima when searching large, complex design spaces.

**Strengths:**

- The proposed AF belongs to the class of information-theoretic AFs, a class of theoretically grounded strategies. Framing the problem as trying to gain information over local optima rather than global optima is novel and well-motivated, specifically for high-dimensional settings, and the practical execution into a computationally reasonable algorithm is also a contribution in itself.

- The experiments section is quite extensive, to say the least. I have to say that I rarely encounter BO papers involving this level of thoroughness at the experiment section level. This was quite pleasant.

- ``Despite'' such a large breadth of experiments, LES does not seem to suffer any major flaw, compared to other baselines, and achieved the overall best performance. While one can always pretend that problems were cherry-picked to favor LES, I consider that the range of experiments here is wide enough to provide a fair representation of the test cases usually considered in the BO literature.

- Besides comparing to other baselines, the authors carried out several ablation studies to assess the impact of modifying individual blocks, which did not reveal any major flaw either.

**Weaknesses:**

I cannot pinpoint concrete weaknesses. The limitations of the approach are clearly stated, albeit briefly in the conclusion, but are presented in more detail in the appendix. I agree with such limitations; they do not represent grounds for rejection in my opinion.

**Questions:**

Since I don't think the paper suffers from any major weaknesses, my questions stem from genuine curiosity rather than a desire to challenge the method.

- Throughout the paper, an RBF kernel is employed for the GP surrogate. Draws from that surrogate are used to compute the acquisition function, where a finite number of optimization steps are performed for each draw, leading to a discretized descent sequence, again for each draw.
Given that the RBF kernel produces infinitely differentiable function draws, these should be quite amenable to gradient descent /ADAM, putting aside convexity. If one knows that the black-box function is not as regular as draws from an RBF kernel, one could opt for, say, a Matern 1/2 GP surrogate, whose draws should be more difficult to optimize. Have you tried to see how the performance of the AF changes as the kernel changes? In particular, I would assume that the results reported in Appendix E.6 (comparing different iterative optimizers) might vary conditionally on the kernel?

- It might be interesting to discuss the case of latent space BO, where one learns a continuous latent representation $\mathcal{Z}$ of a structured space $\mathcal{X}$ (e.g., JT-VAE [1,2]) and performs BO within that latent space, decoding any selected design $x=h(z)$ and then evaluating it. Without any incentive, there is little reason to think that the function $z \mapsto f(h(z))$ is smooth, and hence an AF based on maximizing the information about sequences produced by a gradient descent optimizer might not perform as well. As in the previous question, perhaps then a zero-order optimizer like CMAES might outperform ADAM/GD.

[1] Junction Tree Variational Autoencoder for Molecular Graph Generation, ICML 2018
[2] Sample-Efficient Optimization in the Latent Space of Deep Generative Models via Weighted Retraining, NeurIPS 2020

Some comments:
- Appendix E.6: performing a fair comparison with CMA-ES is always tricky, in my mind, given the number of hyperparameters it involves. You mentioned $\sigma$, but there is also the popsize, the evolution strategy conducted every generation (e.g., bipop), to only mention two.
- noise is kept pretty low throughout the paper, between $0.001$ and $0.02$ for standardized data. I know this is not the point of the paper, but varying the noise to see the impact would have been interesting as well.
- page 21, a superscript "2" points to a footnote "3".
- appendix F, I would also write the definition $f_t^\star = \text{sup}_x f_t(x)$
- Defintiion 2: use $\varepsilon$ or $\epsilon$ not both ?
- Algorithm 1 L9 involves $k_{test}$, shouldn't it be $\delta_{test}$?

---

> ### Author Response · Authors · 2025-11-23
>
> ## Questions:
>
> > Throughout the paper, an RBF kernel is employed for the GP surrogate. Draws from that surrogate are used to compute the acquisition function, where a finite number of optimization steps are performed for each draw, leading to a discretized descent sequence, again for each draw. Given that the RBF kernel produces infinitely differentiable function draws, these should be quite amenable to gradient descent /ADAM, putting aside convexity. If one knows that the black-box function is not as regular as draws from an RBF kernel, one could opt for, say, a Matern 1/2 GP surrogate, whose draws should be more difficult to optimize. Have you tried to see how the performance of the AF changes as the kernel changes? In particular, I would assume that the results reported in Appendix E.6 (comparing different iterative optimizers) might vary conditionally on the kernel?
>
> We agree with this observation and expect the kernel choice to influence the suitability of different inner optimizers. Sample paths from a Matérn-1/2 GP are not continuously differentiable, making LES-ADAM and LES-GD less suitable choices in that setting. In such cases, zeroth-order optimizers (e.g., CMA-ES, hill climbing, pattern search) may be more appropriate. We added a short discussion of this point in Appendix D.6 (prev. E.6). However, we did not conduct experiments with kernels other than the RBF kernel.
>
> > It might be interesting to discuss the case of latent space BO, where one learns a continuous latent representation $\mathcal{Z}$ of a structured space $\mathcal{X}$(e.g., JT-VAE [1,2]) and performs BO within that latent space, decoding any selected design $x=h(x)$ and then evaluating it. Without any incentive, there is little reason to think that the function $z \mapsto f(h(z))$ is smooth, and hence an AF based on maximizing the information about sequences produced by a gradient descent optimizer might not perform as well. As in the previous question, perhaps then a zero-order optimizer like CMAES might outperform ADAM/GD.
>
>  We agree that it is interesting to consider LES in latent-space BO. As LES inherits the properties of the surrogate model, a decoded objective that is not smooth may indeed make gradient-based LES variants less suitable, in which case using a zeroth-order inner optimizer would be the natural choice. We also note that smoothness in latent representations can be explicitly encouraged [R1].
>
> [R1] Siddharth Ramchandran, Manuel Haussmann, and Harri Lähdesmäki. High-dimensional bayesian optimisation with gaussian process prior variational autoencoders. In International Conference on Learning Representations (ICLR), 2025.
>
> ## Comments
>
> > Comment 1: Appendix E.6: performing a fair comparison with CMA-ES is always tricky, in my mind, given the number of hyperparameters it involves. You mentioned $\sigma$, but there is also the popsize, the evolution strategy conducted every generation (e.g., bipop), to only mention two.
>
> We agree that tuning CMA-ES involves many hyperparameters and that performing a fully fair comparison is challenging. Our main goal in Appendix E.6 is to illustrate that LES also works when paired with non–gradient-based inner optimizers.
>
> > Comments 3 to 6
> > - page 21, a superscript "2" points to a footnote "3".
> > - appendix F, I would also write the definition $f^*_z=\sup_xf_t(x)$
> > - Defintion 2: use $\epsilon$ or $\varepsilon$ not both ?
> > - Algorithm 1 L9 involves, $k_{test}$ shouldn't it be $\delta_{test}$?
>
> Thank you for reviewing the paper in such detail and pointing out these issues. We have corrected all of them in the revised manuscript.

---

> > ### Comment · Reviewer_yURN · 2025-11-27
> >
> > Thank you for your rebuttal! I will keep my score as is.

---

### Official Review · Reviewer_o3dG · 2025-11-01

**Soundness:** 4
**Presentation:** 4
**Contribution:** 4
**Rating:** 10
**Confidence:** 4

**Summary:**

Local Entropy Search (LES) is presented as a local, information-theoretic Bayesian optimization framework aimed at finding better solutions in complex, large design spaces. LES combines iterative local optimizers with the entropy-search principle to identify the best reachable solution from a given initial design $x_0$ for an expensive-to-evaluate black-box function. The key idea is to target the solution produced by a chosen local optimizer by focusing on its descent sequence, and to propagate the uncertainty of a Gaussian process (GP) surrogate through that optimizer to induce a distribution over possible descent paths starting from the initial design. This leads to an LES acquisition that selects the next candidate by minimizing the entropy of the descent sequence, equivalently maximizing the mutual information between the new observation and the descent sequence conditioned on current data. The derivation uses a closed-form expression for predictive entropy and a Monte Carlo estimate of the conditional entropy over sampled sequences. With these components in place, the paper presents the LES algorithm. For practicality, conditional entropy is evaluated at discretized support points along the descent sequences, and an early-stopping criterion is employed. Empirically, LES shows strong performance on high-complexity settings and remains competitive with TuRBO.

**Strengths:**

1. The paper shows real depth, both theoretically and empirically. The authors clearly know the literature, handle the fine details, and take a principled path to build LES.

2. The empirical study is broad, with enough variants to support strong claims about LES performance across complex settings, different distribution regimes (in-model and out-of-model), and a thoughtful set of baselines.

3. The paper gives a robust treatment of edge cases. It shows the one-step equivalence to GIBO, explains why certain alternative formulations are intractable, introduces an early stopping rule based on probabilistic regret, and provides demonstrations across multiple levels of complexity.

4. LES works across optimizers. The paper backs this with results for Adam, CMA ES, and others, which makes the robustness claim of LES credible.

5. The authors paid careful attention to scalability, including a batched variant (QLES) for practical deployment and efficient sampling via decoupled GP.

6. The experimental setup is clean and complete, which makes the work reproducible.

7. The authors also provide a clear and sufficient exposition of all components of the LES algorithm, with careful pointers to the appendix for detailed explanations, which helps both interested and unfamiliar readers.

**Weaknesses:**

1. The most prominent weakness, which the authors already acknowledge, is the tendency to get trapped in a local minimum basin, especially in complex settings with a highly multimodal posterior. The paper offers a careful discussion of this issue and introduces a stopping rule to trigger restarts, but the risk remains in challenging landscapes.

2. LES performance depends on several core components, including the chosen optimizer, the surrogate model specification, and the underlying problem complexity. All of these are however not uncommon in BO settings, and the authors provided a clear account of this.

**Questions:**

1. On local minima traps: the paper mentions multistart or switching between local and global algorithms. A recent ICLR paper by Adebiyi et al. (2025) proposes optimizing posterior samples via root-finding to identify starting points that lie near the global optimum of a posterior sample. Can the authors comment on the possibility of integrating such sorting algorithm in that paper to guarantee the selection of the ideal starting point to further extend the applicability of LES in multimodal scenarios?

2. From the results, it appears LES and TuRBO remains competitive in several complex settings, can the authors comment on the incentives of LES over TuRBO?

3. Although minor, could the authors comment on the selection of learning rate for LES-Adam? This can materially affect performance, especially in GP based settings.

---

> ### Author Response · Authors · 2025-11-23
>
> > Question 1. On local minima traps: the paper mentions multistart or switching between local and global algorithms. A recent ICLR paper by Adebiyi et al. (2025) proposes optimizing posterior samples via root-finding to identify starting points that lie near the global optimum of a posterior sample. Can the authors comment on the possibility of integrating such sorting algorithm in that paper to guarantee the selection of the ideal starting point to further extend the applicability of LES in multimodal scenarios?
>
> Thank you for highlighting this relevant line of work. We agree that advanced strategies for selecting or refining the incumbent $x_0$, including multi-start mechanisms or global-to-local transitions, are promising extensions of LES, especially in multimodal settings. In this work, we intentionally focused on evaluating LES as a pure local search paradigm to clearly isolate its behavior and performance. As a result, we did not explore more sophisticated incumbent-selection strategies.
>
> That said, we fully agree that LES can benefit from such approaches. The root-finding–based posterior-sample optimization proposed by Adebiyi et al. (2025) is compatible with LES and could be used to propose strong candidate starting points or restart locations. This has the potential to further improve performance in highly multimodal settings by guiding LES toward regions that are more likely to contain high-quality solutions.
>
> We added a sentence to the discussion section noting that such strategies,including the method of Adebiyi et al. (2025), are natural extensions and can be integrated into LES to enhance its global exploration capabilities.
>
> > Question 2. From the results, it appears LES and TuRBO remains competitive in several complex settings, can the authors comment on the incentives of LES over TuRBO?
>
> Thank you for the question. Our results show that LES consistently outperforms TuRBO on the most difficult problems and almost never performs worse on GP-samples. As expected for Bayesian methods, performance differences between all BO strategies decrease when the prior becomes less informative.
>
> In practice, we would therefore recommend LES over TuRBO when (i) reasonably good prior knowledge is available or (ii) the objective is particularly complex. We also expect LES to benefit more strongly from improved modeling choices (e.g., full Bayesian treatment or automatic kernel selection).
>
> Moreover, LES achieves substantially lower cumulative regret than TuRBO (Appendix D.3) due to its localized exploration behavior. Whenever cumulative regret is the relevant metric, LES appears advantageous.
>
> The main trade-off is computational cost: LES is more expensive per iteration than TuRBO (Appendix D.5). For relatively cheap-to-evaluate objectives where wall-clock time dominates, TuRBO can therefore remain the preferred choice.
>
> Finally, LES provides an entropy-search–based framework that naturally extends to settings such as constrained, multi-fidelity, or batch BO.
>
> > Question 3. Although minor, could the authors comment on the selection of learning rate for LES-Adam? This can materially affect performance, especially in GP based settings.
>
> We agree that the optimizer hyperparameters, including the learning rate in ADAM, can influence the descent sequence and in some cases even change which local optimum the optimizer converges to. In those situations, the outcome of LES will naturally differ as well, since LES is designed to model and learn from the optimizer's behavior.
>
> However, when different learning-rate settings lead the optimizer to the same local optimum, their effect on LES is minor. LES conditions on the full (discretized) descent sequence, so variations in convergence speed or intermediate trajectory geometry have much less impact than changes in the final point of convergence.
>
> We conducted some preliminary testing with different ADAM momentum settings and observed virtually no performance differences. Systematically exploring learning-rate schemes is an interesting direction for future work, for example, adapting the step size using the GP’s learned length scales, as suggested by Muller et al. (2021).

---

### Author Response · Authors · 2025-11-23

We thank all reviewers for their detailed assessments, constructive feedback, and insightful questions. We are pleased that the contribution and empirical depth of the paper were received positively. In the revised manuscript, we addressed the main points raised in the reviews and used the additional page to bring several important results and the limitations discussion from the appendix into the main paper.

 - Several reviewers asked for a clearer high-level motivation for local Bayesian optimization: in high-complexity and high-dimensional settings, global BO must reduce uncertainty across the entire domain, which becomes increasingly intractable. In contrast, reliably reaching a good local optimum requires learning the objective only along a single optimizer-induced trajectory, which is far more sample-efficient. This is precisely the regime LES is designed for.

 - LES is not a heuristic but a direct and principled extension of the entropy-search framework: it applies the mutual-information objective to the random descent sequence induced by a local optimizer. This construction is fully aligned with classical ES and reduces to known local BO methods in appropriate special cases.

 - LES is intentionally a local algorithm, and can be combined with global initialization or multi-start strategies when broader exploration is desired. Our experiments isolate the local-search behavior to evaluate the acquisition function itself, but LES remains compatible with such extensions.

Below we provide detailed responses to all reviewer comments. We are happy to further discuss any remaining questions or concerns.

---

> ### Author Response · Authors · 2025-12-02
>
> In light of recent events and the changes to the rebuttal process, we want to provide a short summary of the current state of the discussion to potentially help the new AC. We are aware that the reviewers can no longer participate in the discussion and acknowledge that we, as authors, have a conflict of interest when summarizing it and apologize in advance if we inadvertently misrepresent a reviewer's point.
>
> In the discussion phase, Reviewer gXxy raised interesting additional questions about the high-dimensional behavior of LES, in particular (i) if LES remains local, and (ii) where along the descent sequences the information gain tends to concentrate.
>
> In response, we ran an additional analysis, reporting (across dimensions and across the first and second half of optimization) the average total length of the descent sequences and the average step size between the current incumbent and the next query. Empirically, we observed that (a) the step size is consistently much smaller than the total sequence length, so entropy reduction is not concentrated solely at the endpoints, and (b) both sequence length and step size decrease over the course of optimization, indicating increasingly local search behavior. We plan to include these new results in the appendix summarizing these statistics and plotting the exploration behavior over time.
>
> Although the discussion phase was cut short, we believe we could address the reviewer's questions and comments. The reviews and discussions has led us to run further analysis and refine the presentation, and we believe the paper has benefited from this feedback. We thank the reviewers for their valuable input.

---

### Meta-Review · Area_Chair_r8fq · 2026-01-09

**Summary:**

The reviewers raised a variety of comments, the only valid one I am going to deal with here is the fact that the experimental results are many, yet mostly pretty toy/synthetic. A few reviewers raised this point in varying wording.

With that said, honestly the method is just too cute to not like. As a researcher who often rolls eyes at "yet another entropy search for this twist" paper, I have to say instead: what a delightful little spin on local search, and a much more clever distribution to target the entropy of.

**Reviewer Concerns:**

The authors response to the one specific criticism I want to talk about above is effectively that they acknowledge that their method loses some of its edge when the function is not in the model family considered.

But: how do you know that? It sounds like you've run on some of those more "real" problems that the reviewers were asking for if you know that is true. If so, include the results. If not: run on more interesting functions! This almost by definition can't be a good response to that criticism, since in order to know it you have to have run it and just not included it in the paper, or you haven't run it yet and are just assuming your method won't work on those benchmarks. They really aren't more expensive than the Ackley's of the world.

**Reviewer Scores:**

Most of the reviewers were pretty enthusiastic from the get go about the paper, and the negative review didn't really add any criticisms that I felt were more important than the one above, which the other reviewers were excited about the paper in spite of.

Ultimately I think it's fine, but run on more interesting functions!

---

### Decision · Program_Chairs · 2026-01-26

Accept (Poster)